# The scaffolding function of LSD1 controls DNA methylation in mouse ESCs

Sandhya Malla[1,2], Kanchan Kumari[1,2], Carlos A. García-Prieto [3,4], Jonatan Caroli[5], Anna Nordin [6,7], Trinh T. T. Phan [8], Devi Prasad Bhattarai[1,2], Carlos Martinez-Gamero[1,2], Eshagh Dorafshan[1,2], Stephanie Stransky [9], Damiana Álvarez-Errico [3], Paulina Avovome Saiki[1,2], Weiyi Lai[10], Cong Lyu[10], Ludvig Lizana [11], Jonathan D. Gilthorpe [12], Hailin Wang[10], Simone Sidoli [9], Andre Mateus [13,14], Dung-Fang Lee [8,15,16,17], Claudio Cantù [6,7], Manel Esteller [3,18,19,20], Andrea Mattevi [5], Angel-Carlos Roman[21] & Francesca Aguilo [1,2] ✉

Lysine-specific histone demethylase 1 (LSD1), which demethylates mono- or di-methylated histone H3 on lysine 4 (H3K4me1/2), is essential for early embryogenesis and development. Here we show that LSD1 is dispensable for mouse embryonic stem cell (ESC) self-renewal but is required for mouse ESC growth and differentiation. Reintroduction of a catalytically-impaired LSD1 (LSD1[MUT]) recovers the proliferation capability of mouse ESCs, yet the enzymatic activity of LSD1 is essential to ensure proper differentiation. Indeed, increased H3K4me1 in *Lsd1* knockout (KO) mouse ESCs does not lead to major changes in global gene expression programs related to stemness. However, ablation of LSD1 but not LSD1[MUT] results in decreased DNMT1 and UHRF1 proteins coupled to global hypomethylation. We show that both LSD1 and LSD1[MUT] control protein stability of UHRF1 and DNMT1 through interaction with HDAC1 and the ubiquitin-specific peptidase 7 (USP7), consequently, facilitating the deacetylation and deubiquitination of DNMT1 and UHRF1. Our studies elucidate a mechanism by which LSD1 controls DNA methylation in mouse ESCs, independently of its lysine demethylase activity.

Embryonic stem cells (ESCs) represent the immortal in vitro capture of a very early stage of the developing embryo. They maintain the potential to proliferate indefinitely and differentiate into all three embryonic germ layers, providing material for cell-based therapies and holding out the promise to transform the next generation of medicine. In contrast to somatic cells, for which the transcriptional status of most genes is epigenetically fixed, ESCs globally possess decondensed chromatin, which can constantly be remodeled during developmental specification[1,2]. This is sustained by the coordination of transcription factors, chromatin regulators, DNA and histone marks, and RNA modifiers[1,3].

Lysine-specific histone demethylase 1 (LSD; also known as KDM1A/AOF2/BHC110) is a histone modifying enzyme that demethylates the mono-and di-methyl moieties of histone H3 lysine 4 (H3K4me1/2)[4,5]. Although LSD1 has been shown to be involved in early embryogenesis[6,7], our comprehension of its function in ESC self-renewal and differentiation is still evolving[8]. For example, in human ESCs, LSD1 has been shown to maintain self-renewal by silencing developmental genes[9], and loss of LSD1 promotes neural lineage differentiation[10]. Moreover, it has been reported that mouse ESCs depleted of LSD1 retain stem cell characteristics, suggesting that LSD1 timely regulates the expression of key developmental regulators during early embryonic development[11]. In addition, other studies have also shown that LSD1 is not essential for the maintenance of ESC identity but it is required for the differentiation of multiple cell types in vitro

and for the late cell-lineage determination and differentiation during pituitary organogenesis in vivo[7,12–15]. Mechanistically, it has been proposed that LSD1 is poised at active pluripotency enhancers in mouse ESCs to rapidly silence the pluripotency program during lineage commitment allowing for proper differentiation[16].

Non-histone substrates of LSD1, including key regulators of DNA methylation maintenance such as DNA (cytosine-5)-methyltransferase 1 (DNMT1) and Ubiquitin-like, with PHD and RING finger domains 1 (UHRF1), have also been demonstrated[6,17–19]. Indeed, deletion of Lsd1/ LSD1 in mouse ESCs and cancer cells have been shown to induce progressive global DNA hypomethylation[6,20]. LSD1-mediated demethylation of DNMT1 increases DNMT1 protein stability by preventing its degradation by the proteasome[6,21,22]. Similarly, LSD1-mediated demethylation of UHRF1 in the G2/M phase prevents its ubiquitination and thus UHRF1 degradation[20], and diminishes the interaction between UHRF1 and PCNA, which is essential for DNA repair[17]. Yet another study proposes an indirect mechanism by which H3K4me1 demethylation by LSD1 is required to maintain DNA methylation levels at pluripotency genes leading to enhancer silencing during ESC differentiation[23].

Several lines of evidence demonstrate that demethylase-independent functions of LSD1 orchestrate tumorigenesis[24]. For instance, it has been shown that the interaction with LSD1 can lead either to degradation (e.g., p62) or stabilization (e.g., ERRα) of the interacting protein independently of LSD1 catalytic activity[25]. In addition, LSD1 functions as a pseudosubstrate of the E3 ubiquitin ligase FBXW7 triggering its self-ubiquitination and rapid degradation[26]. More recently, the histone demethylase activity of LSD1 has been reported to be dispensable for endocrine-resistant breast tumorigenesis[27]. Whether LSD1 non-canonical mechanisms also operate in pluripotency is yet to be explored.

Here, we show that ablation of LSD1 in mouse ESCs does not affect ESC self-renewal but impairs cellular growth. Albeit Lsd1 KO cells with reintroduction of a catalytically-impaired LSD1 (LSD1MUT) recover the proliferation capability, both Lsd1 KO and LSD1MUT mouse ESCs undergo defective differentiation. In mouse ESCs, loss of LSD1 results in a gain of H3K4me1 in the promoters and distal intergenic regions of a subset of genes without affecting global gene expression programs related with stemness. Moreover, in the ESC state, deletion of Lsd1 results in global DNA hypomethylation coupled with decreased DNMT1 and UHRF1 protein levels. Notably, Lsd1 KO with reintroduction of wild-type LSD1 (LSD1WT) or LSD1MUT mouse ESCs recover the protein levels of DNMT1 and UHRF1, and thus, DNA methylation levels. Strikingly, recovery of DNA methylation in LSD1MUT is not sufficient to allow for normal differentiation. Mechanistically, LSD1WT and LSD1MUT can associate with the deubiquitinase USP7 (ubiquitin-specific protease 7, also known as HAUSP) and protect DNMT1 and UHRF1 from proteasomal degradation. Additionally, HDAC1-mediated deacetylation of DNMT1 and UHRF1 also plays a key role in maintaining their stability. Our results prompt a re-evaluation of the proposed mechanism of action for LSD1 in demethylating non-histone substrates, especially DNMT1 and UHRF1, to increase their stability. They also bring light to LSD1-HDAC1-USP7 axis to coordinate DNA methylation maintenance in mouse ESCs.

## Results

### Lsd1 is dispensable for mouse ESC self-renewal

To understand the regulatory function of LSD1 in pluripotency, we first assessed the expression of Lsd1 in retinoic acid (RA)-induced neuronal differentiation and in embryoid bodies (EBs), comprising representatives of all three embryonic layers. Quantitative-reverse transcription PCR (RT-qPCR) revealed a moderate decrease in Lsd1 mRNA during RA-induced differentiation whereas Lsd1 mRNA expression was significantly higher at days 2, 4 and 6, but returned to control levels at day 8 of EB differentiation (Fig. 1A, B). LSD1 protein levels decreased at day

2 upon RA treatment but not during EB differentiation (Fig. 1C, D), arguing for a specific post-transcriptionally regulation of LSD1 occurring during neural differentiation. OCT4 expression, alongside the expression of the neuronal marker Nestin and the endodermal marker Sox17, were used to monitor proper mouse ESC differentiation (Fig. 1A–D). We next employed CRISPR/Cas9 to generate Lsd1 knockout (KO) mouse ESCs. To this aim, we used two different strategies: (i) a combination of two single guide RNAs (sgRNAs #1 and #2) targeting exon 1 (namely KO1); and (ii) one sgRNA (#3) targeting exon 6 which included the SWIRM domain of LSD1 (thereafter referred to as KO2; Supplementary Fig. 1A). After picking and expanding individual clones, targeted disruption of Lsd1 was confirmed by PCR and Sanger sequencing (Supplementary Fig. 1B, C). Consistently, three bands were detected in KO1 whereas KO2 displayed two bands after T7 endonuclease I digestion, indicative of biallelic heterozygosity and homozygosity, respectively (Supplementary Fig. 1D). Lsd1 KO mouse ESCs were further validated by western blotting and immunofluorescence analysis (Supplementary Fig. 1E, F).

In order to characterize Lsd1 KO clones, we performed proliferation, apoptosis, and cell cycle assays. Loss of LSD1 resulted in severely compromised cell growth and a marked increase in apoptosis, suggesting that LSD1 is required for the normal proliferative function of mouse ESCs (Fig. 1E, F). However, no significant difference in the cell cycle profile was observed in Lsd1 KO compared to WT mouse ESCs (Supplementary Fig. 2A). In addition, Lsd1 KO mouse ESCs retained ESC morphology and were alkaline phosphatase (AP)-positive, indicative of maintenance of pluripotency (Fig. 1G). Of note, Lsd1 deletion increased the number of partially differentiated colonies, whereas the number of undifferentiated colonies significantly decreased (Fig. 1H). Notably, ablation of Lsd1 did not affect the expression of the pluripotency factors OCT4 and NANOG (Fig. 1I). Moreover, the immunostaining assay of OCT4 and SSEA1 did not display any difference in the expression levels between Lsd1 KO and WT mouse ESCs (Fig. 1J). Hence, loss of Lsd1 in mouse ESCs appears to impair its basic proliferative functions while preserving the pluripotent potential.

To examine the role of LSD1 in the naïve ground state, mouse ESCs were cultured in 2iL medium, consisting of kinase inhibitors targeting MAP kinase (MEK) and glycogen synthase kinase 3β (GSK-3β; known as "2i"), along with leukemia inhibitory factor (LIF)[28]. Similar to the observed phenotype when growing the cells in medium containing serum and LIF, loss of LSD1 resulted in decreased cellular viability and increased apoptosis (Supplementary Fig. 2B, C). However, despite these effects, AP staining revealed consistent counts of both undifferentiated and partially differentiated colonies across the different cell lines (Supplementary Fig. 2D, E). This data aligns with the notion that while LSD1 may not be essential for maintaining pluripotency, its absence triggers significant proliferation deficiencies in both the metastable (LIF + serum) and in the naïve ground states (2iL).

To identify the transcriptional program of these Lsd1 KO mouse ESCs, we performed RNA-sequencing (RNA-seq) analysis. Gene expression profiles of both clones were slightly distinct due to normal stochastic heterogeneity between isolated clones (Fig. 1K, L; Supplementary Fig. 2F, G; Supplementary Data 1), yet, KO transcriptomes clustered together and displayed more similarity among them than with the RNA-seq signals retrieved from WT mouse ESCs (Fig. 1M). Gene ontology (GO) analysis of biological processes of common upregulated genes revealed categories related to generic functions, including negative regulation of cell differentiation, apoptosis, and endodermal differentiation (Fig. 1N). Common downregulated genes in KO1 and KO2 showed enrichment for nucleosome assembly, DNA repair and metabolic processes, among others (Fig. 1O). GO analysis did not reveal global alterations in categories associated with pluripotency and germ layer-specific markers between WT and Lsd1 KO mouse ESCs. Nevertheless, we did detect a specific upregulation of the endoderm genes Sox17 and Gata6, which were used to validate the

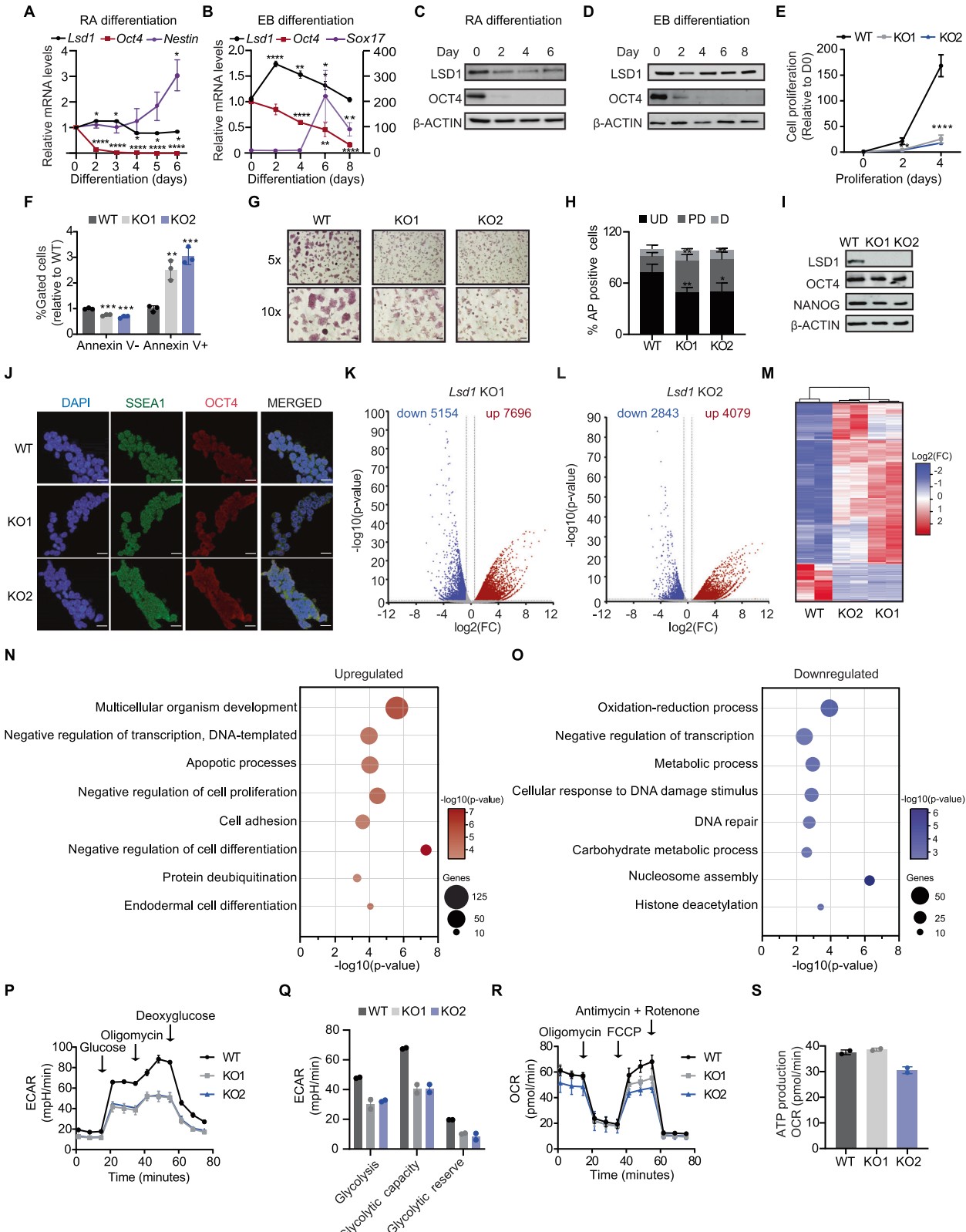

RNA-seq data by RT-qPCR (Supplementary Fig. 2H). Additionally, we validated the defective metabolic phenotype of *Lsd1* KO mouse ESCs by a flux analyzer. Loss of *Lsd1* led to decreased basal glycolysis and KOs were not able to respond to increased energetic demand even when oxidative phosphorylation was shut down (Fig. 1P, Q). To assess whether defective glycolysis results in the shift to oxidative metabolism, we performed a mito stress test. However, we did not observe

differences in mitochondrial function and ATP production between WT and *Lsd1* KOs, indicating that *Lsd1* KO mouse ESCs might use alternative fuels such as fatty acids to drive OXPHOS- ATP production (Fig. 1R, S). Taken together, RNA-seq showed that loss of LSD1 does not have a profound effect on the expression of pluripotency-associated genes and thereby, we were able to maintain *Lsd1* KO mouse ESCs in culture over long periods of time without loss of self-renewal.

**Fig. 1 | Loss of *Lsd1* does not lead to transcriptional deregulation of pluripotency genes.** RT-qPCR analysis of *Lsd1* in mouse ESCs along the course of (**A**) retinoic acid (RA)-mediated and (**B**) embryoid body (EB) differentiation. *Oct4* was used as a marker for pluripotency. **A** *Nestin* and (**B**) *Sox17* were employed as markers for neuronal and EB differentiation, respectively. In (**B**), the relative *Sox17* mRNA levels are represented on the right Y-axis. The mRNA levels are relative to the expression at day 0. Western blot of LSD1 and OCT4 on the whole-cell extracts (WCE) of mouse ESC subjected to (**C**) RA or (**D**) EB differentiation. β-ACTIN is used as the loading control. **E** Relative cell proliferation rate of WT and *Lsd1* KO mouse ESCs. **F** Percentage of live (Annexin V-) and apoptotic cells (Annexin V + ) in *Lsd1* KO mouse ESCs relative to WT. **G** AP staining images and (**H**) quantification of colonies in WT and *Lsd1* KO mouse ESCs. Undifferentiated (UD), partially differentiated (PD), and differentiated (D). Scale bars, 50 µm. **I** Western blot of LSD1, OCT4, and NANOG on WCE of WT and *Lsd1* KO mouse ESCs. β-ACTIN is used as the loading control. **J** Immunofluorescence images of SSEA1 and OCT4 in WT and *Lsd1* KO mouse ESCs. DAPI was used as the nuclear marker. Scale bars, 20 µm. Volcano plots of differentially expressed transcripts in (**K**) *Lsd1* KO1 and (**L**) *Lsd1* KO2 mouse ESCs in comparison to WT mouse ESCs. Significantly upregulated and downregulated transcripts are represented in red and blue, respectively ($p < 0.05$ and Fold change (FC) > 1.5). Non-significant hits are shown in gray dots. FDR value was calculated with the Benjamini−Hochberg correction. **M** Heatmap of differentially expressed genes in WT and *Lsd1* KO mouse ESCs. The upregulated and downregulated genes are indicated in red and blue, respectively. Gene ontology (GO) analysis of biological processes related to the common (**N**) upregulated and (**O**) downregulated genes of *Lsd1* KO mouse ESCs compared to WT mouse ESCs ($p < 0.05$ and FC > 1.5). *P*-values were adjusted with the Benjamini−Hochberg correction. **P** Measurement of extracellular acidification rate (ECAR) at indicated time points to determine glycolysis stress and (**Q**) glycolytic metabolic parameters in WT and *Lsd1* KO mouse ESCs. **R, S** Quantification of Oxygen consumption rate (OCR) over time and (S) ATP production in *Lsd1* KO mouse ESCs compared to WT mouse ESCs using the Seahorse mito stress test. Statistical analysis: Two-tailed unpaired t-test (**A**, **B**, **H**, and **K**−**L**), and ordinary one-way ANOVA (**E**, and **F**). $*p < 0.05$, $**p < 0.01$, $***p < 0.001$, and $****p < 0.0001$. Error bars denote mean ± SD; $n = 3$ (**A**, **B** (except for *Sox17* D2), and **E**); n = 4 (**H**); and n = 2 (**P**−**S**). Each dot in the bar graphs represents independent biological replicates (**F**). The exact *P*-values for panels (**A**, **B**, **E**, **F** and **H**) are represented in the source data. Results are one representative of n = 3 independent biological experiments (**C**, **D**, **G**, **I** and **J**). Uncropped blots are represented in the source data.

## Ablation of *Lsd1* leads to defective differentiation

To test the role of LSD1 in differentiation, WT and *Lsd1* KO mouse ESCs were differentiated to EBs for 8 days. *Lsd1* KO showed a significant reduction in the size of EBs compared to EBs derived from WT mouse ESCs (Supplementary Fig. 3A, B), suggesting a defective differentiation phenotype upon loss of *Lsd1*. In addition, *Lsd1* KO mouse ESCs were not able to suppress the expression of the core pluripotency factors *Oct4* and *Nanog* during the course of differentiation (Supplementary Fig. 3C). Moreover, EBs generated from both *Lsd1* KO mouse ESCs failed to express lineage-specific markers such as *Sox17* and *Foxa2* (endodermal) (Supplementary Fig. 3D), *Brachyury* (*T*) and *Msx1* (mesodermal) (Supplementary Fig. 3E), and *Sox11* (ectodermal) (Supplementary Fig. 3F). In order to exclude potential off-target effects, we next engineered a cell line in which *Lsd1* KO2 mouse ESCs were stably expressing a MYC-tagged WT LSD1 (LSD1[WT]) or a MYC-tagged mutated LSD1 (LSD1[MUT]) (Supplementary Fig. 3G). The residues required for demethylating histone H3K4 have not been described yet for the mouse LSD1 ortholog. However, previous studies have shown that double point mutation of human LSD1 at the residues Alanine 539 and Lysine 661 (A539/K661) is required to abolish its demethylase activity[29]. Therefore, we performed multi-species LSD1 protein alignment and we found that A539 and K661 are highly conserved among the represented species (Fig. 2A). Based on this alignment, we engineered and purified a double point mutated (A540E/K662A) mouse LSD1 protein (LSD1[MUT]) and assayed its activity together with a WT LSD1 (LSD1[WT]). The demethylase activity of LSD1[WT] ($K_{cat}$) was $1.74 \pm 0.023\,min^{-1}$ whereas the enzymatic activity of LSD1[MUT] was undetectable, indicating that A540E/K662A resulted in a catalytically-impaired LSD1 (Fig. 2B, left and right panel). Strikingly, both LSD1[WT] and LSD1[MUT] cell lines rescued the proliferation defect of *Lsd1* KO mouse ESCs and showed colony morphology and AP staining comparable to WT mouse ESCs (Supplementary Fig. 3H−J). In LSD1[MUT], RNA-seq analysis unveiled 685 downregulated genes and 1115 upregulated genes compared to their wild-type counterparts (Supplementary Fig. 3K; Supplementary Data 1). Among the 1115 upregulated genes in LSD1[MUT] mouse ESCs, 668 genes were similarly upregulated due to LSD1 deletion (Supplementary Fig. 3L). Further insight into the distinction between the impacts of LSD1 deletion and catalytic inactivation on the mouse ESC transcriptome was underscored through visualization of RNA-seq data (Supplementary Fig. 3M). Notably, LSD1[WT] cells were able to rescue the transcriptomic defects as only 390 genes were dysregulated, potentially reflecting clonal variability (Supplementary Fig. 3N). GO analysis of the biological processes associated with upregulated genes in LSD1[MUT] mouse ESCs identified categories related to regulation of transcription, chromatin organization, DNA repair, and neuron

projection development, amongst others (Supplementary Fig. 3O). Conversely, downregulated genes in LSD[MUT] exhibited enrichment in generic categories such as cell differentiation, cell cycle, phosphorylation, and chromatin organization, among others (Supplementary Fig. 3P). Once again, despite these alterations in gene expression, there were no observable differences in the expression of pluripotency-related markers between the wild-type and LSD[MUT] mouse ESCs. However, the defective size of *Lsd1* KO EBs was only partially recovered in LSD1[MUT] (Fig. 2C, D), as LSD1[MUT] EBs were not able to silence the core pluripotency factors and to enhance the expression of lineage markers similar to *Lsd1* KO EBs (Fig. 2E−I). Overall, these results indicate that albeit the enzymatic activity of LSD1 is not required to maintain the proliferation capacity of mouse ESCs, it is essential to trigger proper differentiation.

Studies have highlighted that the A540E mutation disrupts substrate binding in LSD1 by introducing a positive charge, while the K662A allele primarily impacts the FAD-dependent oxidase function of LSD1[29,30]. Therefore, our aim was to investigate the impact of substrate binding or catalytic function on both mouse ESC growth and EB formation by using cell lines individually carrying each of these single mutants (LSD1[A540E] and LSD1[K622A]). In line with our findings in the double mutant cell line (LSD1[MUT]), both LSD1[A540E] and LSD1[K622A] mouse ESCs exhibited a proliferation rate similar to that of WT mouse ESCs (Supplementary Fig. 4A). Our analysis further revealed that the EBs derived from LSD1[A540E] and LSD1[K622A] were significantly smaller than WT EBs, albeit partially rescuing the size of EBs derived from *Lsd1* KO mouse ESCs, which was only 50% of their wild-type counterparts (Supplementary Fig. 4B, C, 3A, B). LSD1[K622A] EBs demonstrated a more prominent defect in suppressing the expression of *Oct4* compared to LSD1[A540E] EBs, while no significant differences were observed between LSD1[A540E] and WT-derived EBs for *Nanog* and *Sox2* on day 8 of differentiation (Supplementary Fig. 4D). Additionally, although both mutants failed to promote developmental gene expression at WT levels, the impact was more pronounced in LSD1[K662A]-derived EBs (Supplementary Fig. 4E−G). This data suggests that the catalytic function of LSD1 holds more importance in EB differentiation than its substrate binding function.

To validate our findings that loss of LSD1 impairs mouse ESC differentiation, we tested the role of LSD1 in gastruloid generation. Gastruloids are small aggregates of ESCs that, under appropriate culture conditions, recapitulate the axial organization of post-implantation embryos and mimic major aspects of gastrulation[31]. *Lsd1*/LSD1 was uniformly expressed, implying that LSD1 has functional importance during gastrulation, whereas the expression of *Oct4*/OCT4 was decreased after 120 h of gastruloid formation (Supplementary

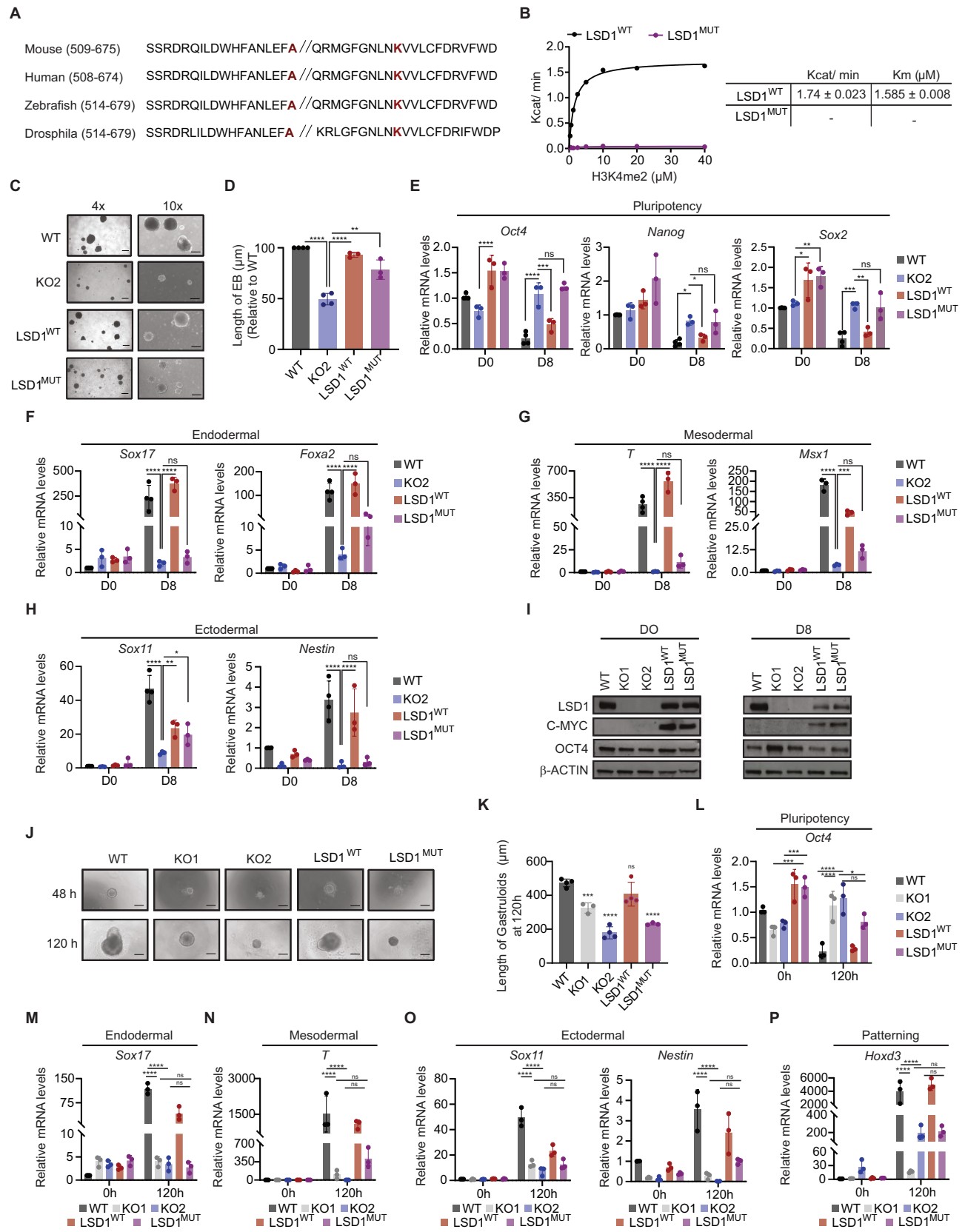

Fig. 4H, I). We next generated gastruloids from WT, *Lsd1* KO, LSD1^WT and LSD1^MUT mouse ESCs. *Lsd1* KO mouse ESCs failed to give rise to elongated gastruloid-like structures, being these aggregates significantly smaller than in WT mouse ESCs (Fig. 2J, K). Noteworthy, the morphologic phenotype was rescued in LSD1^WT but not in LSD1^MUT mouse ESCs (Fig. 2J, K). Similar to what we observed during EB formation, gastruloids derived from *Lsd1* KO and LSD1^MUT mouse ESCs retained higher levels of *Oct4* compared with WT, and the expression

**Fig. 2 | LSD1 is essential for mouse ESC differentiation. A** Sequence alignment of LSD1 in different species, with mutated amino acid residues highlighted in red. **B** Saturation curves of histone demethylase activity of purified WT and mutant LSD1 (LSD1[MUT]) proteins with increasing concentrations of H3K4me2 as a substrate. Each enzymatic curve was derived from the Michaelis-Menten equation (left panel) and their reaction kinetics (right panel). **C** Representative bright field images at (4x (left) and 10x (right)) magnification (**D**) quantification of the size of EB derived from WT, *Lsd1* KO2, LSD1[WT,] and LSD1[MUT] mouse ESCs at day 8 of differentiation. Scale bars, 200 μm. RT-qPCR analysis of (**E**) pluripotency (*Oct4, Nanog,* and *Sox2*), (**F**) endodermal (*Sox17* and *Foxa2*), (**G**) mesodermal (*T* and *Msx1*), and (**H**) ectodermal (*Sox11* and *Nestin*) markers in WT, *Lsd1* KO2, LSD1[WT,] and LSD1[MUT] mouse ESCs on day 0 and day 8 of EB differentiation. mRNA levels are relative to the expression of WT at day 0. *β-actin* is used as an internal control. **I** Western blot of LSD1, C-MYC, and OCT4 on whole cell extract of EBs of WT, *Lsd1* KO1, *Lsd1* KO2, LSD1[WT] and LSD1[MUT] mouse ESCs at day 0 and day 8 of EB differentiation. β-ACTIN is used as the loading control. **J** Morphological representation (10x magnification) and (**K**) measurement of the length of gastruloids derived from WT, *Lsd1* KO1, *Lsd1* KO2, LSD1[WT], and LSD1[MUT] mouse ESCs at indicated time points. Scale bar, 200 μm. Bar graph depicting the RT-qPCR analysis of (**L**) pluripotency (*Oct4*), (**M**) endodermal (*Sox17*), (**N**) mesodermal (*T*), (**O**) ectodermal (*Sox11* and *Nestin*), and (**P**) patterning (*Hoxd3*) markers in the gastruloids generated from WT, *Lsd1* KO1, *Lsd1* KO2, LSD1[WT,] and LSD1[MUT] mouse ESCs at 0 h and 120 h of gastrulation. mRNA levels are relative to the expression of WT at 0 h. *β-actin* is used as an internal control. Statistical analysis: Two-tailed unpaired t-test (**D, K**) and two-way ANOVA (**E–H** and **L–P**). $*p < 0.05$, $**p < 0.01$, $***p < 0.001$, and $****p < 0.0001$. Error bars denote mean ± SD; n ≥ 3 (**D–H** and **K–P**). Each dot in the bar graphs represents independent biological replicates (**D–H, K** and **L–P**). Results are one representative of n = 3 independent biological experiments (**C, I** and **J**). Uncropped blots are represented in the source data.

of the differentiation markers and the axial patterning marker *Hoxd3* failed to be upregulated (Fig. 2L–P). Collectively, these results show that loss of LSD1 abrogates proper gastruloid formation and revealed that the catalytic activity of LSD1 is required for proper mouse ESC differentiation.

## Loss of LSD1 results in accumulation of H3K4me1

To gain a better overview of the LSD1-mediated regulation of the histone modification landscape in mouse ESCs, we performed nanoscale liquid chromatography coupled to tandem mass spectrometry (nano LC-MS/MS), a robust quantitative method for the characterization of post-translational modifications (PTMs) of histones. Because a given PTM is commonly present in different peptides due to combinatorial PTM possibilities, we focused on the analysis of deconvoluted single PTMs. Hence, to retrieve the relative abundance of histone PTMs, the sum of all modified forms in their respective histone peptides was considered as 100% (Supplementary Data 2). After calculating the co-existence of the individual histone marks in WT mouse ESCs, our analysis revealed that H3K27me2 and H3K36me2 were the two most abundant methylation marks as previously reported, whereas most of K4, K14, K18 and K23 were not modified on histone H3 (Supplementary Fig. 5A)[32]. Next, we compared the relative abundance of histone marks in WT and *Lsd1* KO mouse ESCs (Supplementary Fig. 5B). H3_3_8K4me1 was the most abundant common peptide in both KOs compared to WT mouse ESCs (Supplementary Fig. 5C, D). Moreover, our analysis showed that H3K4me1 was significantly increased upon loss of LSD1, whereas no differences were observed in H3K4me2 and H3K4me3 between WT and *Lsd1* KO mouse ESCs (Fig. 3A). This data indicates that, in mouse ESCs, LSD1 demethylates the active histone modification mark H3K4me1 thereby acting as a transcriptional repressor.

To further investigate H3K4me1 levels regulated by LSD1, we performed chromatin immunoprecipitation-coupled with high-throughput DNA massively parallel sequencing experiments (ChIP-seq) with antibodies against either H3K4me1 or LSD1. We identified a total of 9,740 LSD1 peaks throughout the genome in WT mouse ESCs whereas only a negligible number of peaks (140) were found in *Lsd1* KO mouse ESCs (Fig. 3B; Supplementary Data 3). The vast majority of LSD1 peaks were promoter-distal (7662; >3 kb from TSS), which is consistent with prior observations[16]. LSD1 peaks overlapped with binding sites for transcription factors with a reported stem cell function such as MAZ, NR5A2, and Sox2/4, and other transcriptional factors involved in the differentiation of ESCs (Fig. 3C)[16,33,34]. Integration of RNA-seq in WT and *Lsd1* KO mouse ESCs with ChIP-seq data showed that the majority of genes near to LSD1 peaks are not differentially expressed (~70%; Fig. 3D). In WT mouse ESCs, we identified 4262 genes marked with H3K4me1, while in *Lsd1* KO mouse ESCs, the count was 9965 genes, with a significant 63% of the genes in WT mouse ESCs overlapping with *Lsd1* KO mouse ESCs (Fig. 3E; Supplementary Data 4). This is consistent with the observation that loss of LSD1 leads to an accumulation of

H3K4me1 in mouse ESCs. However, we found that such global increase of H3K4me1 in *Lsd1* KO mouse ESCs did not correlate with major changes in gene expression (Supplementary Fig. 5E). Particularly, 1414 genomic regions gaining H3K4me1 showed a significant upregulation of the expression of transcripts whereas only a minority (328) displayed downregulation (Supplementary Fig. 5E). GO analysis for biological processes showed that LSD1 binds preferentially to genes related to 'regulation of transcription', 'multicellular organism development' and 'cell differentiation' (Fig. 3F). Likewise, H3K4me1 peaks that were gained in both WT and *Lsd1* KO mouse ESCs were enriched in similar categories (Fig. 3F). We also found binding of LSD1 at genes important for 'Stem cell population maintenance' although to a lesser extent (Fig. 3F). We next identified genes that are repressed (1588) or overexpressed (462) during RA-mediated differentiation (FC > 1.5) by using publicly available data[35]. When overlapping the aforementioned genes with LSD1 ChIP-seq, we found that both overexpressed and repressed genes were bound by LSD1 (Supplementary Fig. 5F), suggesting that LSD1 does not promote or inhibit stem cell fate but instructs a more general role in gene expression of mouse ESCs[36]. Hence, enhancers controlling the expression of both pluripotency (e.g., *Nanog, Prdm14, Oct4*) and developmental genes (e.g., *Sox11* and *Sox17*) gained H3K4me1 upon loss of LSD1 (Fig. 3G, H; Supplementary Fig. 5G–I).

Following the adapted Cleavage Under Targets and Release Using Nuclease (CUT&RUN) protocol CUT&RUN-LoV-U (low volume and urea), we mapped H3K4me1 in WT, KO, LSD1[WT], and LSD1[MUT] mouse ESCs[37–39]. Principal component analysis unveiled a clear distinction between WT and *Lsd1* KO mouse ESCs, while LSD1[WT] and LSD1[MUT] clustered closely with WT mouse ESCs (Supplementary Fig. 5J). We identified 5288 target genes in WT, 8540 in KO, 5391 in LSD1[WT], and 7230 in LSD1[MUT], consistent with the demethylating role of LSD1 on H3K4me1 in vivo (Fig. 3I; Supplementary Data 5). Likewise, LSD1 deletion and inactivation led to elevated H3K4me1 levels at enhancers and superenhancer regions (Fig. 3J, K), and co-occupied LSD1 binding sites (Fig. 3L). Genes shared among all lines, as well as those from the LSD1[MUT] cell line, showed enrichment in generic functions such as regulation of transcription, cell cycle and RNA processing, amongst others (Fig. 3M; Supplementary Fig. 5K). However, H3K4me1 peaks exclusively identified in *Lsd1* KO mouse ESCs (1699; Fig. 3I) were associated with genes involved in neurogenesis-related categories (Fig. 3N). These findings underscore the distinctions in the H3K4me1 landscape mediated by LSD1 deletion or enzymatic activity of LSD1.

## Loss of LSD1 leads to global DNA hypomethylation

It has been shown that targeted deletion of LSD1 leads to progressive loss of global DNA methylation[6,20,23]. Hence, to further explore the role of LSD1 on DNA methylation, we employed liquid chromatography-tandem mass spectrometry (LC-MS/MS) and measured the abundance of 5-methylcytosine (5mC) and the oxidized 5mC derivative,

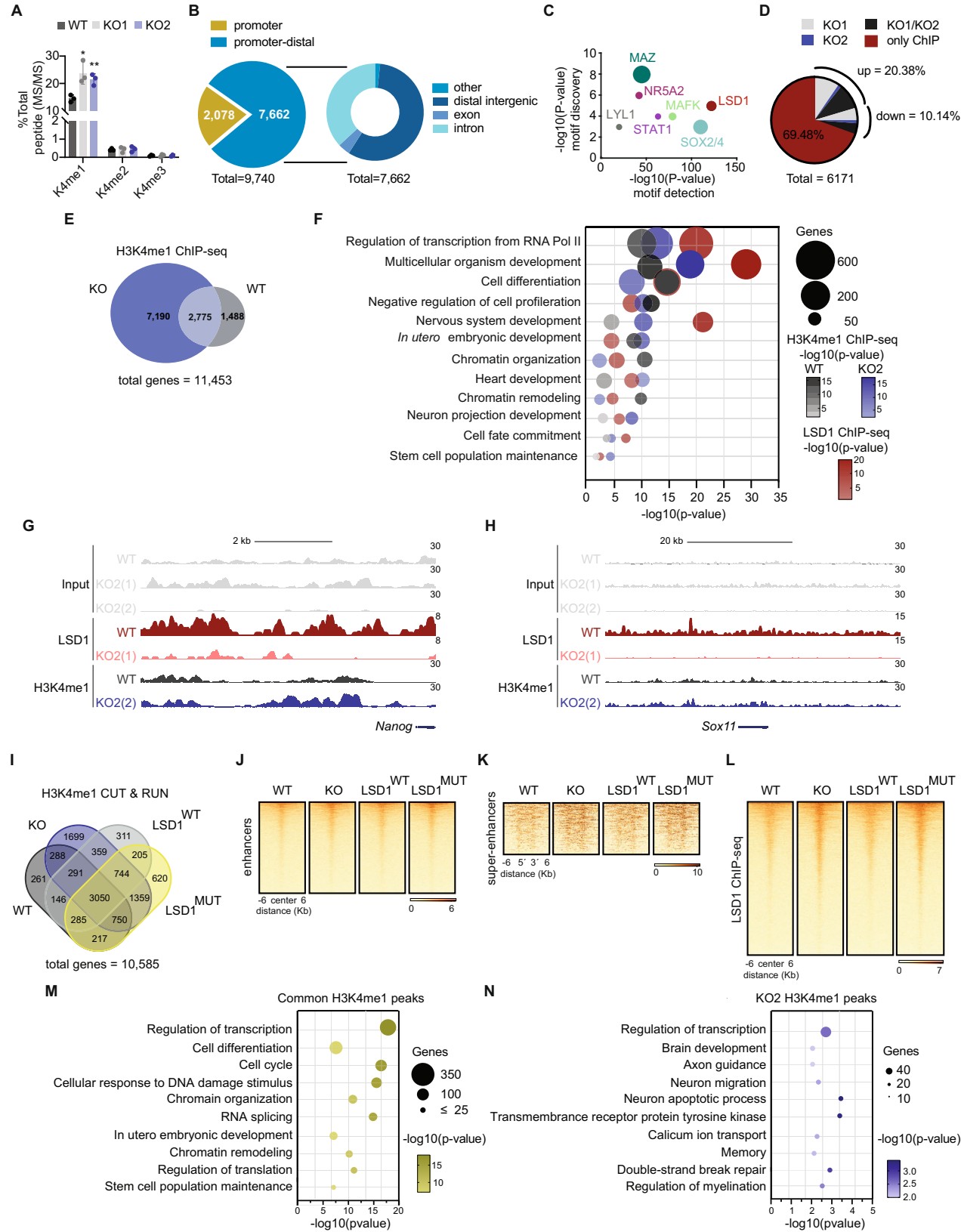

5-hydroxymethylcytosine (5hmC), in genomic DNA of WT and *Lsd1* KO mouse ESCs. Our analysis revealed that both 5mC and 5hmC levels were remarkably decreased upon loss of *Lsd1* (Fig. 4A; Supplementary Fig. 6A). Dot blot on isolated genomic DNA and immunofluorescence analysis with specific antibodies against 5mC and 5hmC confirmed the same result (Fig. 4B; Supplementary Fig. 6B, C). The 5hmC/5mC ratio

remained similar between WT and *Lsd1* KO mouse ESCs, suggesting that lower levels of 5hmC are a consequence of the reduced amount of available 5mC substrate rather than active 5mC hydroxylation (Supplementary Fig. 6D). Next, in order to exclude potential off-target effects, we assessed global DNA methylation levels in LSD1^WT mouse ESCs. LSD1^WT mouse ESCs exhibited 5mC levels similar to WT mouse

**Fig. 3 | Ablation of LSD1 affects global H3K4me1 levels. A** Quantification of H3K4 methylation (K4me1, K4me2, and K4me3) in WT and *Lsd1* KO ESCs. Each modification represents the percentage of one modified peptide among the total modified peptides observed in independent replicates. **B** Genomic distribution of LSD1 binding in the promoter (within 5 kb upstream of TSS) and promoter-distal regions (left) and the fraction of different regions such as distal intergenic, exon, and intron within promoter-distal regions (right) in WT mouse ESCs. **C** Enriched transcription factor motifs at LSD1 peaks in WT mouse ESCs. Homer algorithm was used to estimate the *P*-values of each motif discovery (the enrichment of the sequence in the dataset) as well as the *P*-values of each motif discovery (the similarity to a known motif). The size of the blobs represents the percentage of sequences with the motif in the dataset. **D** Pie chart representing the percentage of LSD1-only bound genes, bound and activated in *Lsd1* KO1, KO2 or KO1/KO2 mouse ESCs, and bound and repressed in *Lsd1* KO1, KO2 or KO1/KO2 mouse ESCs. **E** Venn diagram of overlapped genes in H3K4me1 ChIP-seq between WT and *Lsd1* KO2

mouse ESCs. **F** GO analysis of biological processes of genes associated with LSD1 peaks in WT mouse ESCs and H3K4me1 peaks in WT and *Lsd1* KO2 mouse ESCs. *P*-values were adjusted with the Benjamini–Hochberg correction. LSD1 and H3K4me1 ChIP-seq signals in WT and *Lsd1* KO2 mouse ESCs at the (**G**) *Nanog* and (**H**) *Sox11* enhancers. Respective inputs are depicted in gray. **I** Venn diagram of genes retrieved from H3K4me1 CUT & RUN in WT, *Lsd1* KO2, LSD1^WT and LSD1^MUT mouse ESCs. Density plots of H3K4me1 signals at (**J**) enhancers and (**K**) super-enhancers regions in WT, *Lsd1* KO2, LSD1^WT and LSD1^MUT mouse ESCs. **L** Density plots of regions co-occupied by LSD1 and H3K4me1 in WT, *Lsd1* KO2, LSD1^WT and LSD1^MUT mouse ESCs. Gene ontology analysis of (**M**) common genes and (**N**) genes exclusive to KO-associated H3K4me1 peaks. *P*-values were adjusted with the Benjamini–Hochberg correction. Statistical analysis: Two-tailed unpaired t-test (**A**). *p < 0.05, **p < 0.01. Error bars denote mean ± SD; n = 3 (**A**). Each dot in the bar graphs represents independent biological replicates (**A**). The exact *P*-values for panels (**A**) are represented in the source data.

ESCs whilst re-introduction of an empty MYC vector (KO2^EV) was not able to rescue the hypomethylation phenotype observed upon *Lsd1* loss (Fig. 4C, D). DNA hypomethylation levels of *Lsd1* KO mouse ESCs were similar to *Dnmt1* KO mouse ESCs, but relatively higher than *Dnmt3a/Dnmt3b* double KO mouse ESC lines (Supplementary Fig. 6E, F). Remarkably, LSD1^MUT mouse ESCs partially recovered the DNA methylation levels of *Lsd1* KO mouse ESCs (Fig. 4E), indicating that, in contrast to previous studies[6,20], the catalytic activity of LSD1 may not be necessary for influencing DNA methylation at certain specific genomic regions.

To further evaluate the role of LSD1 in DNA methylation, we next assessed the DNA methylation landscape by whole-genome bisulfite sequencing. The genome was divided into 1 kb tiles, and the distribution of CpG methylation values was compared within each tile across the samples. We identified 83,671 and 83,950 differentially methylated regions (DMRs) in *Lsd1* KO1 and KO2 compared to WT, respectively, confirming the hypomethylated phenotype upon loss of LSD1 (Fig. 4F, G). To gain a global perspective on the effect of DNA methylation on CpG islands and their neighboring regions, we generated composite plots for mean methylation levels across 25,489 CpG islands. We found a general decrease in methylation in CpG islands which was not just at the center of the island but also on CpG island shores (Supplementary Fig. 6G). Most of the DMRs were enriched in gene bodies (+500 bp from the transcription start site (TSS) to +500 bp from the transcription end site (TES)) and distal regions (>5 kb upstream or >500 bp downstream) whereas hypomethylated promoters (−5kb to +500 bp from TSS) were underrepresented (Fig. 4H). Such DMR distal regions were enriched for repetitive elements such as SINEs and LINEs, and for enhancer regions, whereas we found a quantitative association between DNA methylation at promoters of WT and *Lsd1* KO mouse ESCs, confirming our previous observation that promoter sequences do not suffer major changes in methylation upon loss of LSD1 (Supplementary Fig. S6H–J). Altogether, these data suggest that LSD1 regulates DNA methylation genome-wide, especially in repetitive elements, enhancers, and gene bodies.

To further understand the function of LSD1 in DNA methylation, we profiled 5mC levels in WT, *Lsd1* KO, LSD1^WT and LSD1^MUT mouse ESCs by employing the Mouse Methylation MM285 BeadChIP microarray[40]. We defined differentially methylated CpG sites as those with absolute methylation difference >66% and adjusted *p*-value < 0.05 (Supplementary Data 6). DNA methylation β-value density plots showed a bimodal distribution, being the unmethylated peak higher than the methylated peak, reflecting a genome-wide DNA methylation loss that mouse ESCs exhibit (Fig. 4I). Furthermore, the number of fully methylated CpGs was lower for *Lsd1* KO mouse ESCs compared to the other cell lines analyzed. Indeed, principal component analysis showed a clear separation between the WT and *Lsd1* KO mouse ESCs, whereas both LSD1^WT and LSD1^MUT clustered close to WT mouse ESCs, confirming that the hypomethylation phenotype of *Lsd1* KO mouse ESCs is

partially rescued upon the introduction of the catalytically-impaired LSD1 (Fig. 4J). Unsupervised analysis of 10,000 random CpGs revealed the same finding with similar methylation patterns in WT, LSD1^WT, and LSD1^MUT, but distinct from the *Lsd1* KO mouse ESCs (Fig. 4K). We identified a set of 1,099 loci that were hypomethylated upon LSD1 loss, whereas only 111 and 240 were hypomethylated in LSD1^WT and LSD1^MUT mouse ESCs, respectively, compared to WT mouse ESCs (Fig. 4L–O). Specifically, we found 111 probes in LSD1^WT mouse ESCs, corresponding to 10 genes, that remained hypomethylated upon reintroduction of a LSD1^WT, suggesting intrinsic differences amongst cell lines. Those genes remained hypomethylated also in LSD1^MUT mouse ESCs. In addition, 126 loci, corresponding to 46 genes, were hypomethylated in *Lsd1* KO and LSD1^MUT mouse ESCs, indicating that the catalytic activity of LSD1 is required to establish proper methylation at some specific loci (Supplementary Fig. 6K, L). Of note, we only found a negligible number of hypermethylated genes in LSD1^MUT mouse ESCs. Differential methylation between WT and *Lsd1* KO mouse ESCs at promoter regions, but not at body or 3′UTR, negatively correlated with changes in gene expression (Fig. 4P, Q; Supplementary Data 7). We next explored the genomic distribution of hypomethylated genomic elements from *Lsd1* KO mouse ESCs. Intriguingly, promoter regions (corresponding to TSS 1500, TSS 200, 5′UTR and 1st Exon) were more represented than in the bisulfite analysis, albeit we also detected hypomethylation in the gene bodies and intergenic regions (Fig. 4R). Most of the CpGs identified were significantly enriched in open sea regions (89.08%) and only a minority of them associated with (3.18%) CpG islands, (1.00%) with shelves (2–4 kb from the promoter CpG islands) and (6.73%) with shores (0–2 kb from the promoter CpG islands) (Fig. 4S). Integration of CUT&RUN datasets with their methylation profiles indicated that H3K4me1 peaks are hypomethylated globally and at enhancer regions, with no major differences between the different cell lines (Supplementary Fig. 6M, N), underscoring the interplay among chromatin marks. Altogether, this data show that loss of LSD1 leads to global hypomethylation, and that this effect is mostly independent of LSD1 demethylase activity.

## LSD1 regulates DNMT1 and UHRF1 protein levels independently of its catalytic activity

Because several studies have shown that LSD1 stabilizes both DNMT1 and UHRF1 proteins[6,35], we next assessed whether the protein expression levels of the DNA methylation maintenance machinery were altered in *Lsd1* KO mouse ESCs. Western blot analysis on whole cell and chromatin extracts showed reduced DNMT1 and UHRF1 upon *Lsd1* loss (Fig. 5A, B). Such decrease of DNMT1 and UHRF1 proteins were not associated with changes in mRNA levels as indicated by our RNA-seq analysis (Supplementary Fig. 7A, B). However, the expression of de novo DNA methyltransferases DNMT3A and DNMT3B, and the accessory factor DNMT3L, were increased in *Lsd1* KO compared to WT mouse ESCs, albeit exhibiting lower global 5mC levels than WT

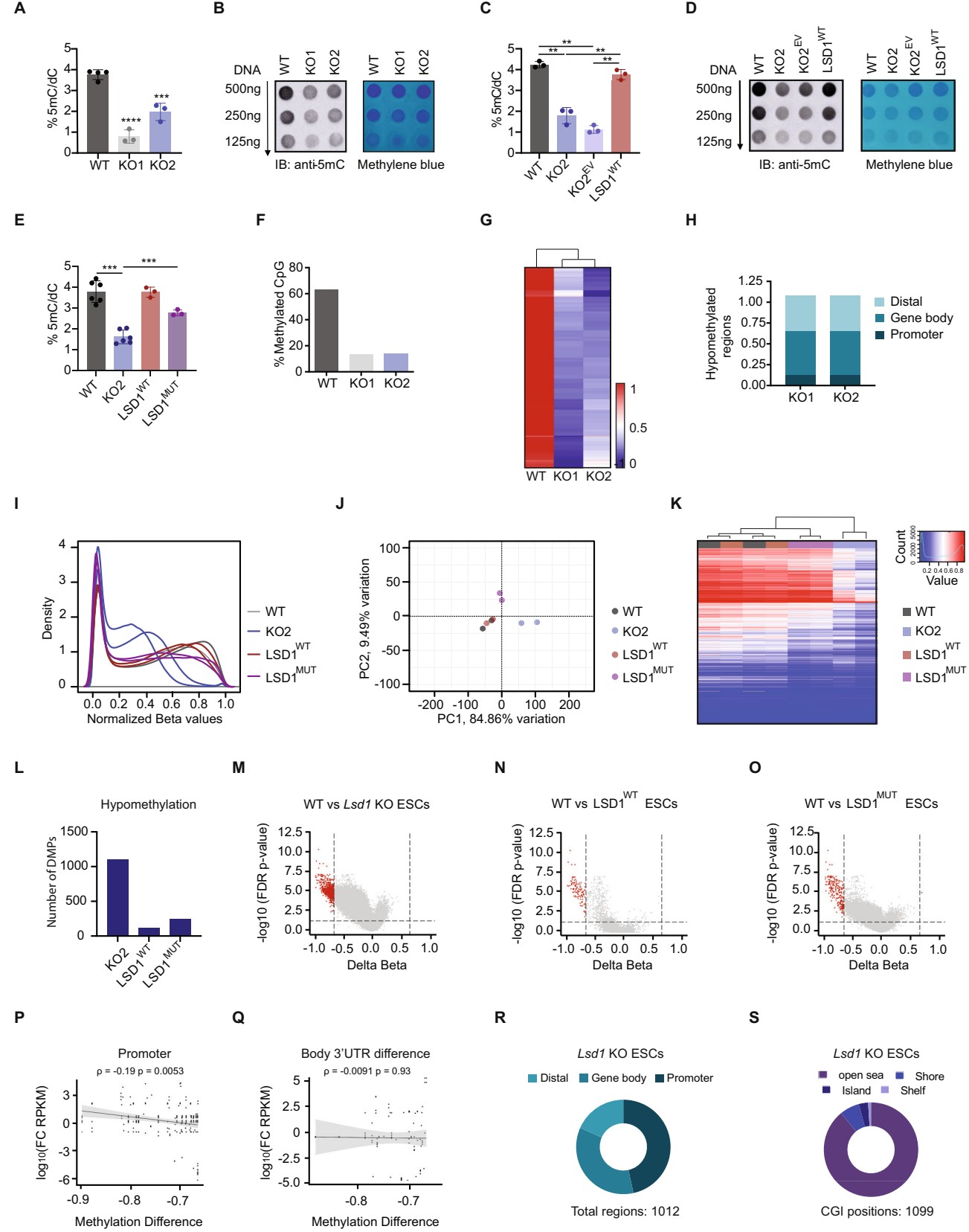

(Supplementary Fig. 7C, D). In this case, protein abundances correlated with mRNA expression levels of *Dnmt3a* and *Dnmt3b* (Supplementary Fig. 7E–G). Similarly, DNMT3A, DNMT3B, and DNMT3L proteins were also significantly increased in *Dnmt1* KO mouse ESCs (Supplementary Fig. 7H, I), suggesting that activated expression of de novo

methyltransferases is a compensatory mechanism to counteract the loss of DNMT1[41]. Reintroduction of LSD1 in *Lsd1* KO mouse ESCs rescued the DNMT1 and UHRF1 protein levels (Fig. 5C, D). In agreement with the observation that concomitant expression of mutant LSD1 in *Lsd1* KO mouse ESC, at least partially, recovered the hypomethylation

**Fig. 4 | Loss of LSD1 affects global DNA methylation. A** LC-MS/MS quantification of 5mC/dC and (**B**) dot blot analysis of 5mC/dC (left panel) of genomic DNA extracted from WT and *Lsd1* KO mouse ESCs. Methylene blue staining was used as loading control (right panel). **C** LC/MS-MS quantification and (**D**) dot blot analysis of 5mC/dC (left panel) of genomic DNA extracted from WT, *Lsd1* KO2, KO2$^{EV}$, and LSD1$^{WT}$ mouse ESCs. Methylene blue staining was used as loading control (right panel). **E** LC/MS-MS quantification of 5mC in genomic DNA extracted from WT, *Lsd1* KO2, LSD1$^{WT}$, and LSD1$^{MUT}$ mouse ESCs. **F** Bar graph showing percentage methylated CpGs in *Lsd1* KO mouse ESCs and WT. **G** Heatmap generated from whole-genome bisulfite-sequencing depicting global DNA methylation in WT and *Lsd1* KO mouse ESCs. Red corresponds to the hypermethylation CpG sites, and in blue, the hypomethylation CpG sites. **H** Percentage of hypomethylation regions in the promoter, gene body, and distal regulatory elements in *Lsd1* KOs. Promoter (−5kb to +500 bp from TSS), Gene body (+ 500 bp from TSS to +500 bp from TES), and Distal (> 5 kb upstream or >500 bp downstream) were considered for the analysis. **I** Density plot of DNA methylation β values of WT, *Lsd1* KO2, LSD1$^{WT}$, and LSD1$^{MUT}$ mouse ESCs. Methylation values range from zero (fully unmethylated) to one (fully methylated). **J** Principal component analysis of array-based DNA methylation profiles of WT, *Lsd1* KO2, LSD1$^{WT}$, and LSD1$^{MUT}$ mouse ESCs. **K** Heatmap and unsupervised hierarchical clustering of methylation levels in 10,000 random CpGs in WT, *Lsd1* KO2, LSD1$^{WT}$, and LSD1$^{MUT}$ mouse ESCs. Red corresponds to the hypermethylated CpG sites, and blue to the hypomethylated CpG sites. Hierarchical clustering was performed with Euclidean distance and Ward´s minimum variance agglomeration method. **L** Bar graph representing number of differentially hypomethylated positions in *Lsd1* KO2, LSD1$^{WT}$, and LSD1$^{MUT}$ compared to WT mouse ESCs. **M**−**O** Volcano plots of differentially methylated positions (DMPs) in (**M**) *Lsd1* KO2 (**N**) LSD1$^{WT}$ and (**O**) LSD1$^{MUT}$ compared to WT mouse ESCs. Red dots indicate significant results (FDR < 0.05 and value of Δβ < −0.66 or >0.66). For differential methylation analysis, DNA methylation values are fitted to a mixed linear model and the corresponding slope test is performed. The slope estimate is the DNA methylation difference for each CpG with respect to the reference level (X-axis). The y-axis contains the FDR adjusted *p*-value of two-tailed unpaired t-testing the slope. Scatter correlation plots of (**P**) promoter methylation and (**Q**) body or 3'UTR methylation and gene expression in *Lsd1* KO compared to WT mouse ESCs. Y-axis represents the log10 fold change of Reads Per Kilobase of transcript per Million mapped reads (FC RPKM) from RNA-seq and x-axis represents β -value methylation difference. Spearman rank correlation coefficient and corresponding *P*-value are shown. **R** Genomic distribution of hypomethylated regions in *Lsd1* KO2 compared to WT mouse ESCs. **S** Pie chart representing distribution of hypomethylated sites with respect to CGI positions. in *Lsd1* KO2 compared to WT mouse ESCs. Statistical analysis: Two-tailed unpaired t-test (**A**, **C**, and **E**). ∗∗*p* < 0.01, ∗∗∗*p* < 0.001, and ∗∗∗∗*p* < 0.0001. Error bars denote mean ± SD; n ≥ 3 (**A**, **C** and **E**). Each dot in the bar graphs represents independent biological replicates (**A**, **C**, and **E**). Results are one representative of n = 3 independent biological experiments (**B** and **D**).

phenotype (Fig. 4E), LSD1$^{MUT}$ mouse ESCs were also able to restore DNMT1 and UHRF1 protein abundance without affecting *Dnmt1* and *Uhrf1* mRNA expression levels (Fig. 5C−F). Additionally, single mutants LSD1$^{A540E}$ and LSD1$^{K662A}$ demonstrated the capability to restore the expression of DNMT1 and UHRF1 proteins, along with 5mC levels (Supplementary Fig. 7J, K). Altogether, and in contrast to previous studies, our data suggest that LSD1 sustains DNMT1 and UHRF1 protein stability independent of the demethylase activity. To validate this observation, we performed in vitro assays using a UHRF1 peptide containing the specified demethylated site K385 (ESKKKALys(Me1) MASATSS) and purified WT LSD1 protein[17]. Our findings indicate that LSD1 is unable to directly demethylate UHRF1 while maintaining its capacity to demethylate the H3K4me2 peptide (Fig. 5G).

To further investigate whether the catalytic activity of LSD1 is dispensable for the expression of DNMT1 and UHRF1, we treated mouse ESCs with GSK-LSD1, a chemical LSD1 inhibitor which has been shown to be involved in the covalent modification of the cofactor flavin adenine dinucleotide (FAD)[42]. The efficiency of the LSD1 inhibitor was determined by performing RT-qPCR on direct targets of LSD1, such as *Sox17* and *Eomes* (Supplementary Fig. 7L, M). Noteworthy, GSK-LSD1 treatment did not affect *Lsd1*, *Dnmt1* and *Uhrf1* mRNAs nor LSD1, DNMT1 or UHRF1 protein levels (Fig. 5H, I; Supplementary Fig. 7N−P). Furthermore, GSK_LSD1 treated mouse ESCs exhibited 5mC levels similar to WT mouse ESCs (Fig. 5J). Longer exposure (1 week) displayed the same result (Fig. 5K), confirming that the catalytic activity of LSD1 is not required for DNMT1 and UHRF1 expression, and DNA methylation maintenance.

Given that DNMT1 protein levels were drastically reduced upon loss of LSD1, we aimed to map DNMT1 binding by ChIP-seq in WT and in *Lsd1* KO mouse ESCs. We found 1286 DNMT1 peaks that were distributed along the genome and more abundant in near promoters (Fig. 5L; Supplementary Data 8). Consistent with decreased DNMT1 protein abundance upon LSD1 loss, we only recovered 27 peaks in *Lsd1* KO mouse ESCs (Fig. 5M). Furthermore, DNMT1 peaks correlated with a decrease in 5mC levels in *Lsd1* KO mouse ESCs (Fig. 5N). The density of DNMT1 signals was higher at the promoters than enhancers in WT mouse ESCs, whereas no distinct signal was observed in *Lsd1* KO mouse ESCs (Fig. 5O). GO analysis for biological processes indicated that DNMT1 binds preferentially to genes enriched in multicellular organism development, regulation of transcription, and neuron-related categories (Fig. 5P). Around 55% of these sites were co-occupied by LSD1, suggesting the possibility that, besides regulating

DNMT1 stability, LSD1 could be also involved in DNMT1 chromatin recruitment (Fig. 5Q, R; Supplementary Fig. 7Q, R).

## LSD1 promotes DNMT1 and UHRF1 stability by preventing proteasomal degradation

The observation that LSD1 regulated DNMT1 and UHRF1 protein levels indicated that LSD1 controls DNMT1 and UHRF1 stability. Thus, we first analysed whether LSD1 interacts with DNMT1 and UHRF1. To this aim, we performed co-immunoprecipitation experiments in whole cell extracts and nuclear fractions of WT, *Lsd1* KO, and LSD1$^{MUT}$ mouse ESCs. We detected interaction of DNMT1 and UHRF1 with LSD1 in both WT and LSD1$^{MUT}$ mouse ESCs (Fig. 6A, B). This interaction remained unaffected by DNase and RNase treatment, indicating its independence from DNA and RNA, respectively (Fig. 6C, D). We next examined whether DNMT1 and UHRF1 protein stability was affected in the absence of LSD1 or when the catalytic center of LSD1 was mutated. To this end, we treated WT, *Lsd1* KO, LSD1$^{WT}$, and LSD1$^{MUT}$ mouse ESCs with protein synthesis inhibitor cycloheximide (CHX) and then monitored the rates of DNMT1 and UHRF1 decline over time. Our data revealed that DNMT1 and UHRF1 half-life was dramatically decreased upon loss of LSD1. However, DNMT1 and UHRF1 had comparable rates of degradation and half-lives in WT, LSD1$^{WT}$, and LSD1$^{MUT}$ mouse ESCs, implying that the demethylase activity of LSD1 is not required to sustain DNMT1 and UHRF1 protein stability (Fig. 6E−G).

To further investigate whether LSD1 could mediate global protein expression levels independently of its catalytic activity, we performed proteomics analysis in WT, *Lsd1* KO, LSD1$^{WT}$, and LSD1$^{MUT}$ mouse ESCs using LC-MS/MS (Supplementary Data 9). We then integrated the transcriptomic and proteomic data to identify post-transcriptional changes. In this regard, a total of 5509 proteins were used for comprehensive analysis, of which 5253 had corresponding mRNAs in the transcriptome (Supplementary Data 10). In *Lsd1* KO compared to WT mouse ESCs, 414 proteins were downregulated, and 221 proteins were upregulated whilst their mRNA remained unchanged (Fig. 6H). These findings indicate that the loss of LSD1 can exert both stabilizing and destabilizing effects on global protein expression as previously reported[25,26,43]. LSD1$^{WT}$ mouse ESCs were able to partially rescue the number of dysregulated proteins (Fig. 6I). Furthermore, we found that only 251 proteins were downregulated and 98 proteins were upregulated in LSD1$^{MUT}$ mouse ESCs (Fig. 6J), implying that altering the enzymatic activity of LSD1 has a diminished impact on proteome regulation compared to the effects observed in *Lsd1* KO ESCs. GO

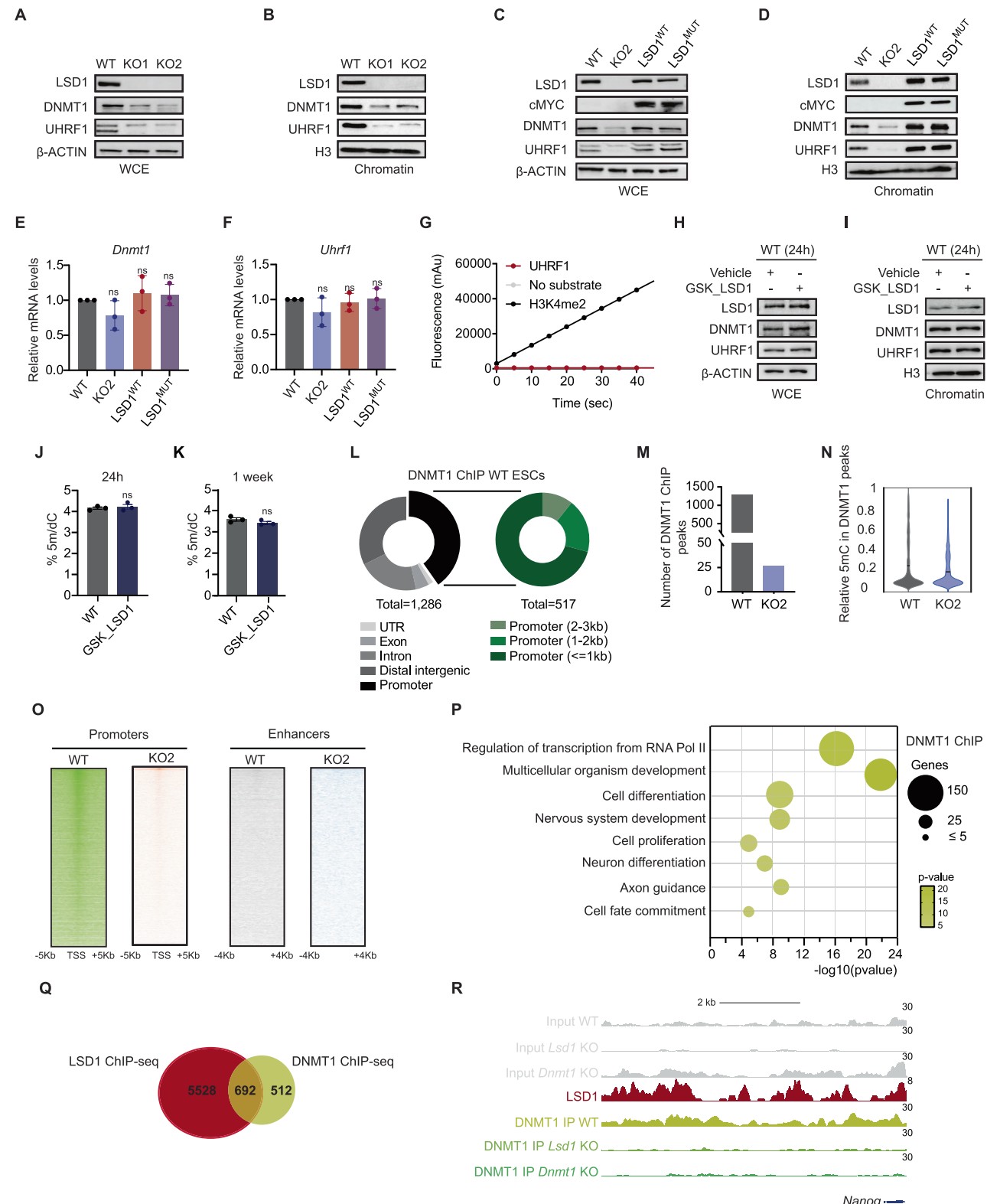

analysis of biological processes of downregulated proteins exclusive to KO revealed categories related to translation, chromatin organization, and protein stabilization (Fig. 6K). Conversely, the upregulated proteins exclusive to the *Lsd1* KO mouse ESCs were enriched in categories such as protein transport and apoptosis (Fig. 6L).

Given that LSD1 promotes DNMT1 and UHRF1 stability, we next aimed to address whether the ubiquitin-proteasome pathway is involved in mediating DNMT1 and UHRF1 turn over. To this end, we treated WT and *Lsd1* KO mouse ESCs with the proteasome inhibitor MG132. The addition of MG132 significantly recovered UHRF1 and DNMT1 protein levels in *Lsd1* KO mouse ESCs, albeit the effect was more robust in UHRF1 than in DNMT1 (Fig. 6M–O), suggesting that LSD1 protects DNMT1 and UHRF1 from proteasome-mediated degradation.

**Fig. 5 | Deletion of *Lsd1* leads to impaired DNA methylation machinery.** Western blots of LSD1, DNMT1, and UHRF1 on (**A**) WCE and (**B**) chromatin fractions of WT and *Lsd1* KO mouse ESCs. β-ACTIN and H3 are used as the loading controls. Western blotting assay with antibodies against LSD1, cMYC, DNMT1, and UHRF1 on (**C**) WCE and (**D**) chromatin fractions of WT, *Lsd1* KO2, LSD1[WT], and LSD1[MUT] mouse ESCs. β-ACTIN and H3 are used as the loading controls. **E, F** Relative levels of *Dnmt1* and *Uhrf1* mRNA in the WT, *Lsd1* KO2, LSD1[WT], and LSD1[MUT] mouse ESCs assessed by RT-qPCR. The mRNA levels are relative to WT mouse ESCs. **G** Line graph representing the demethylase activity of WT LSD1 proteins over time. H3K4me2 was used as a positive substrate for LSD1 demethylase activity. Western blots of LSD1, DNMT1, and UHRF1 on (**H**) WCE and (**I**) chromatin fractions of vehicle and LSD1 inhibitor (GSK_LSD1) treated WT mouse ESCs. β-ACTIN and H3 are used as the loading controls. LC-MS/MS quantification of 5mC on genomic DNA in WT mouse ESCs treated with a vehicle and inhibitor (GSK_LSD1) upon (**J**) 24 h and (**K**) 1 week of treatment. **L** Genomic distribution of DNMT1 binding regions (left) and within different regions of the promoter (right). **M** Bar diagram depicting number of DNMT1 ChIP peaks in WT and *Lsd1* KO2 mouse ESCs. **N** The violin plot depicts the relative 5mC distribution in DNMT1 ChIP peaks in WT and *Lsd1* KO2 mouse ESCs. **O** DNMT1 ChIP−seq heatmap in WT and *Lsd1* KO2 mouse ESCs in promoters (left) and enhancers (right). **P** GO analysis of biological processes of genes associated with DNMT1 peaks in WT mouse ESCs. **Q** Venn diagram of overlapped genes associated with LSD1 and DNMT1 ChIP-seq peaks in WT mouse ESCs. **R** LSD1 ChIP-seq signal in WT mouse ESCs and DNMT1 ChIP-seq signal in WT, *Lsd1* KO2 and *Dnmt1* KO mouse ESCs at the *Nanog* enhancer. Respective inputs are depicted in gray. Statistical analysis: ordinary one-way ANOVA (**E** and **F**) and two-tailed unpaired t-test (**J** and **K**). ns − non-significant. Error bars denote mean ± SD; n = 3 (**E**, **F**, **J** and **K**). Each dot in the bar graphs represents independent biological replicates (**E**, **F**, **J** and **K**). Results are one representative of n = 3 independent biological experiments (**A**–**D** and **H**, **I**). Uncropped blots are represented in the source data.

## LSD1 facilitates deacetylation and deubiquitination of DNMT1 and UHRF1

Previous studies have shown that the deubiquitinating enzyme USP7 plays a role in promoting protein stability of both DNMT1 and UHRF1[44,45]. Additionally, a direct interaction of LSD1 with USP7, in which LSD1 is a target of USP7, has also been described[46]. In order to investigate the role of USP7 in our system, we first examined USP7 expression in whole cell extracts of WT, *Lsd1* KO, LSD1[WT], and LSD1[MUT] mouse ESCs. We did not find any significant difference in USP7 protein levels amongst the four cell lines analysed (Supplementary Fig. 8A). We next sought to determine whether silencing of USP7 affected DNMT1 and UHRF1 protein. To this aim, we cloned two distinct short-hairpin RNAs (shRNAs) against *Usp7* (referred to as sh1 and sh2), and knockdown of *Usp7* was confirmed by Western blot (Supplementary Fig. 8B). Silencing of USP7 diminished DNMT1 and UHRF1 protein abundances, which were recovered after four hours of treatment with MG132 (Fig. 7A–C), while *Dnmt1* and *Uhrf1* mRNA levels remained unaffected (Supplementary Fig. 8C). Furthermore, cycloheximide experiments indicated a shortened half-life of DNMT1 and UHRF1 upon *Usp7* knockdown (Fig. 7D–F). Comparable outcomes were observed following treatment with the specific USP7 inhibitor, P22077 (Supplementary Fig. 8D–H). We then examined whether LSD1 interacted with USP7. Both WT and LSD1[MUT] interacted with USP7 (Fig. 7G). Such interaction was independent of DNA and RNA, as evidenced in WT mouse ESCs treated with DNase and RNase (Supplementary Fig. 8I, J). Conversely, while DNMT1 and UHRF1 interacted with USP7 in WT, LSD1[WT], and LSD1[MUT] cell lines, this interaction was mainly absent in *Lsd1* KO ESCs (Fig. 7H). To assess whether this lack of interaction was due to low protein levels of DNMT1 and UHRF1, we generated two new cell lines expressing doxycycline-inducible DNMT1 or UHRF1 in *Lsd1* KO ESCs (Supplementary Fig. 8K, L). Interestingly, we observed interactions between USP7 and both DNMT1 and UHRF1 in the absence of LSD1 (Fig. 7I, J). This suggests that although LSD1 does not directly bridge the interaction between USP7 and DNMT1/UHRF1, it plays a crucial role in mediating their stability, possibly by aiding their interaction with USP7, resulting in enhanced deubiquitinase activity.

Since LSD1 is a component of the CoREST complex, which includes RCOR1/2/3 and HDAC1/2[47], we aimed to investigate whether the impact of LSD1 on DNMT1 and UHRF1 protein stability was mediated through CoREST. In *Lsd1* KO mouse ESCs, there was a significant decrease in the protein levels of HDAC1, RCOR1, and RCOR2, the two most abundantly expressed RCOR paralogues in mouse ESCs[44], consequently impacting their recruitment to chromatin (Supplementary Fig. 9A, B). These levels were restored to WT levels in both LSD1[WT] and LSD1[MUT] cell lines, indicating that LSD1 serves as a scaffold to stabilize the CoREST complex independently of its enzymatic activity. Of note, no changes in the *Rcor1* and *Rcor2* mRNA levels were observed (Supplementary Fig. 9C, D).

Subsequent immunoprecipitation assays with RCOR1 and RCOR2 antibodies confirmed their interaction with LSD1 and HDAC1, but no interaction was observed with DNMT1 or UHRF1 (Supplementary Fig. 9E, F). Moreover, neither RCOR1 nor RCOR2 exhibited interactions with USP7 (Supplementary Fig. 9G, H). We next employed an shRNA approach to silence the expression of RCOR1 and RCOR2. In RCOR1 knockdown mouse ESCs, western blotting analysis revealed a moderate increase in LSD1 levels compared to WT mouse ESCs, aligning with prior findings (Supplementary Fig. 9I)[48]. This might potentially account for the slight elevated DNMT1 and UHRF1 protein levels observed upon RCOR1 silencing. However, no such changes were observed with RCOR2 loss (Supplementary Fig. 9J). Notably, depleting either RCOR1 or RCOR2 did not result in alterations in *Dnmt1* and *Uhrf1* transcripts levels, nor did it impact 5mC levels (Supplementary Fig. 9K, L). Strikingly, in RCOR2 knockdown mouse ESCs, there was a decrease in LSD1 recruitment to chromatin, corresponding to a reduction in LSD1 binding at its target genes which was assessed by LSD1 ChIP followed by RT-qPCR (Supplementary Fig. 9M, N). Although functionally linked, our data shows that LSD1 modulates DNMT1 and UHRF1 stability independently of RCOR1 and RCOR2.

Further immunoprecipitation assays with HDAC1 antibodies revealed an interaction between HDAC1 and LSD1, but not with DNMT1 and UHRF1 (Fig. 7K). Given that inhibition of HDAC has been shown to destabilize DNMT1 and UHRF1 proteins[49,50], we treated WT ESCs with trichostatin A (TSA) and suberoylanilide hydroxamic acid (SAHA). Consequently, treatment with HDAC inhibitors led to increased acetylation levels of both DNMT1 and UHRF1, correlating with a decrease in protein abundances (Supplementary Fig. 9O–R; Fig. 7L, M). No changes were observed at the transcriptional level, as assessed by RT-qPCR of *Dnmt1* and *Uhrf1* (Supplementary Fig. 9S, T). Additionally, TSA did not disrupt the interaction between LSD1 and DNMT1 or UHRF1 (Supplementary Fig. 9U)[51,52]. Treatment with the proteasome inhibitor MG132 rescued DNMT1 and UHRF1 levels upon HDAC inhibition, indicating destabilization through ubiquitination-mediated proteasomal degradation as previously demonstrated (Fig. 7N–P; Supplementary Fig. 9V-X)[44,45,49]. Although it has been reported that acetylation of DNMT1 negatively impacts its interaction with USP7[53], we detected USP7 binding to both DNMT1 and UHRF1 in the presence of TSA and SAHA (Fig. 7Q). Thus, although both HDAC1 and USP7 contribute to DNMT1 and UHRF1 stability via deacetylation and deubiquitylation, deacetylation does not appear to be a requirement for deubiquitylation to occur in mouse ESCs. Moreover, the combination of HDAC inhibitors with USP7 inhibitors had a synergistic effect on DNMT1 and UHRF1 protein stability, reducing their abundances to nearly undetectable levels (Supplementary Fig. 9Y).

We next examined whether the deletion of LSD1 enhances DNMT1 and UHRF1 ubiquitination. For this purpose, we performed endogenous DNMT1 and UHRF1 immunoprecipitation in MG132-treated WT, *Lsd1* KO, LSD1[WT], and LSD1[MUT] mouse ESCs, followed by immunoblotting with anti-ubiquitin antibodies. Significantly, we detected a higher-molecular-weight smear signal indicative of polyubiquitination in *Lsd1*

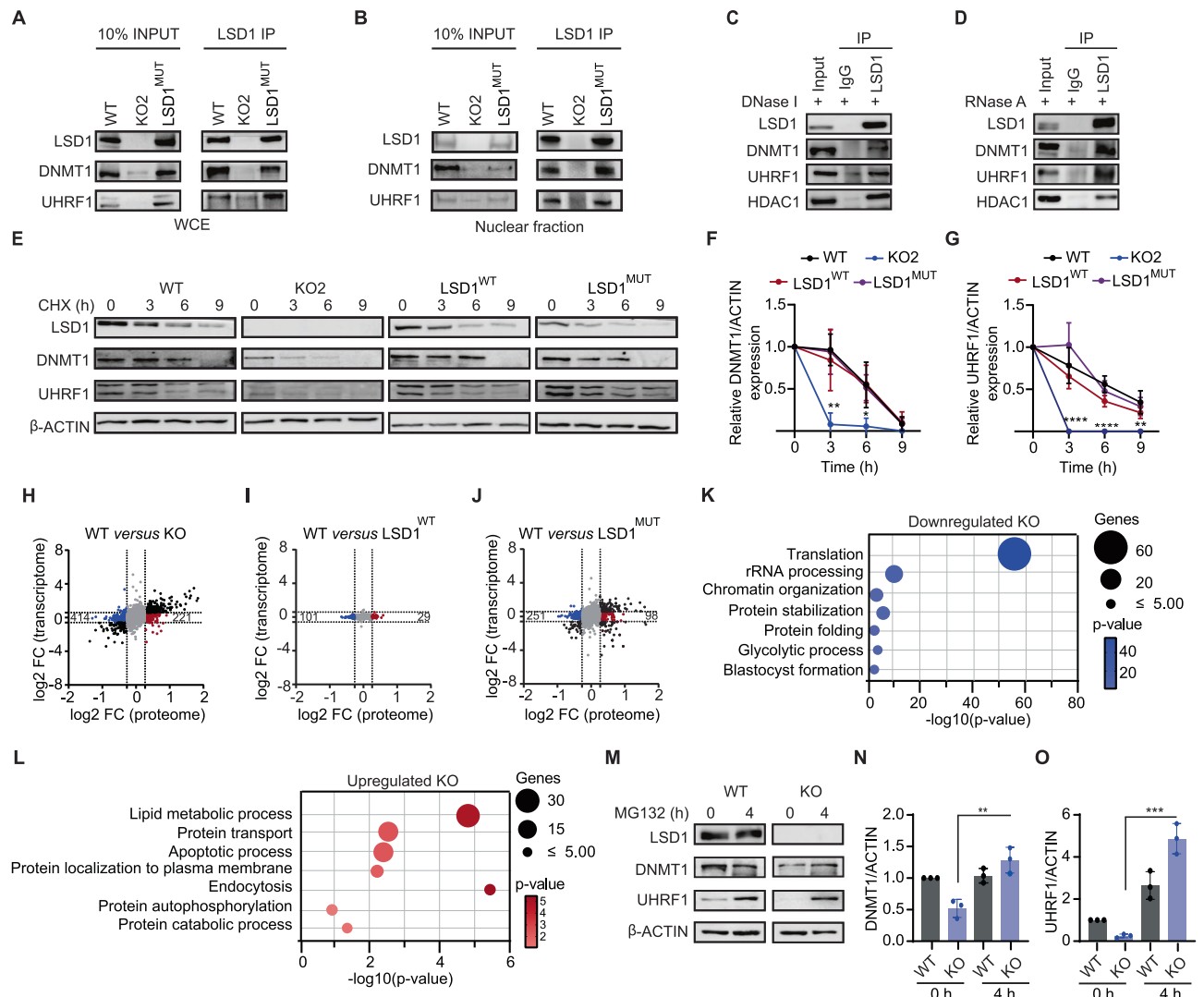

**Fig. 6 | Loss of LSD1 diminishes DNMT1 and UHRF1 stability.** LSD1 immunoprecipitation on the (**A**) WCE and (**B**) nuclear fraction of WT, *Lsd1* KO2, and LSD1^MUT^ mouse ESCs followed by immunoblotting of LSD1, DNMT1 and UHRF1. The percentage of input used is 10%. Immunoprecipitation of LSD1 in the presence of (**C**) DNase I on the nuclear fraction and (**D**) RNase on the WCE of WT ESCs followed by LSD1, DNMT1, UHRF1, and HDAC1 immunoblotting. The percentage of input used is 10%. **E** Western blots of LSD1, DNMT1, and UHRF1 on the WCE of WT, *Lsd1* KO2, LSD1^WT^, and LSD1^MUT^ mouse ESCs during a 9 h CHX time course treatment. β-ACTIN is used as the loading control. Protein degradation curves of (**F**) DNMT1 and (**G**) UHRF1 during the course of 9 h of CHX treatment generated from (**E**). Protein expression is normalized to β-ACTIN and relative to time 0 h. **H**–**J** Comparison of the expression between significantly differentially expressed transcripts (*y*-axis) and proteome (*x*-axis) profiling in (**H**) KO, (**I**) LSD1^WT^ and (**J**) LSD1^MUT^ compared to WT mouse ESCs. Proteins exclusively upregulated and downregulated in proteome are represented in red and blue, respectively (1.2 < FC > 1.2). FDR value was calculated with the Benjamini–Hochberg correction for transcriptome. For transcriptome profiling, $p < 0.05$ and fold change (FC) > 1.5 were considered. **K** GO

analysis of biological processes related to the downregulated proteins that are exclusive to *Lsd1* KO2 mouse ESCs ($p < 0.05$ and FC < 0.8). **L** GO analysis of biological processes related to the upregulated proteins that are exclusive to *Lsd1* KO2 mouse ESCs ($p < 0.05$ and FC > 1.2). **M** Western blots of LSD1, DNMT1, and UHRF1 on the WCE of WT and *Lsd1* KO2 mouse ESCs at 0 and 4 h after MG132 treatment. β-ACTIN is used as the loading control. **N**, **O** Bar graph representing the protein recovery of (**N**) DNMT1 and (**O**) UHRF1 after quantification and normalization of the bands from (**M**). Protein expression is relative to WT mouse ESCs at 0 h. The quantification of samples derive from the same experiment and blots were processed in parallel. Statistical analysis: Two-tailed unpaired t-test (**F**, **G**, **N** and **O**). $*p < 0.05$, $**p < 0.01$, and $***p < 0.001$. Error bars denote mean ± SD; n = F (WT = 5, KO2 = 3, LSD1^WT^ = 5 and LSD1^MUT^ = 4) and G (WT = 6, KO2 = 4, LSD1^WT^ = 6 and LSD1^MUT^ = 4) (independent biological experiments). Each dot in the bar graphs represents independent biological replicates (**N** and **O**). The exact *P*-values for panels (**F**–**G** and **N**–**O**) are represented in the source data. Results are one representative of n = 3 independent biological experiments (**A**–**E** and **M**). Uncropped blots are represented in the source data.

KO compared to WT mouse ESCs (Fig. 7R, S). This signal was reduced in LSD1^WT^ and LSD1^MUT^ mouse ESCs, similar to WT levels (Fig. 7R, S). Moreover, DNMT1 and UHRF1 ubiquitination increased in *Usp7* knockdown cells compared to control cells, and upon USP7 inhibitor treatment (Fig. 7T–W). Overall, our data revealed that demethylase-independent function of LSD1 modulates DNMT1 and UHRF1 protein levels, and LSD1 serves as a scaffolding protein to regulate their deubiquitination and stability (Fig. 8).

## Discussion

Despite the importance of repressive chromatin marks in maintaining ESC identity and transitions to differentiated fates, we still know very little about the mechanism by which LSD1 operates in pluripotency. In this study, we have employed CRISPR/Cas9-mediated genome editing and multi-layered integrative approaches to characterize the function of LSD1 in mouse ESCs. We propose that the demethylase activity of LSD1 does not play a major role in mouse ESC self-renewal, but it is

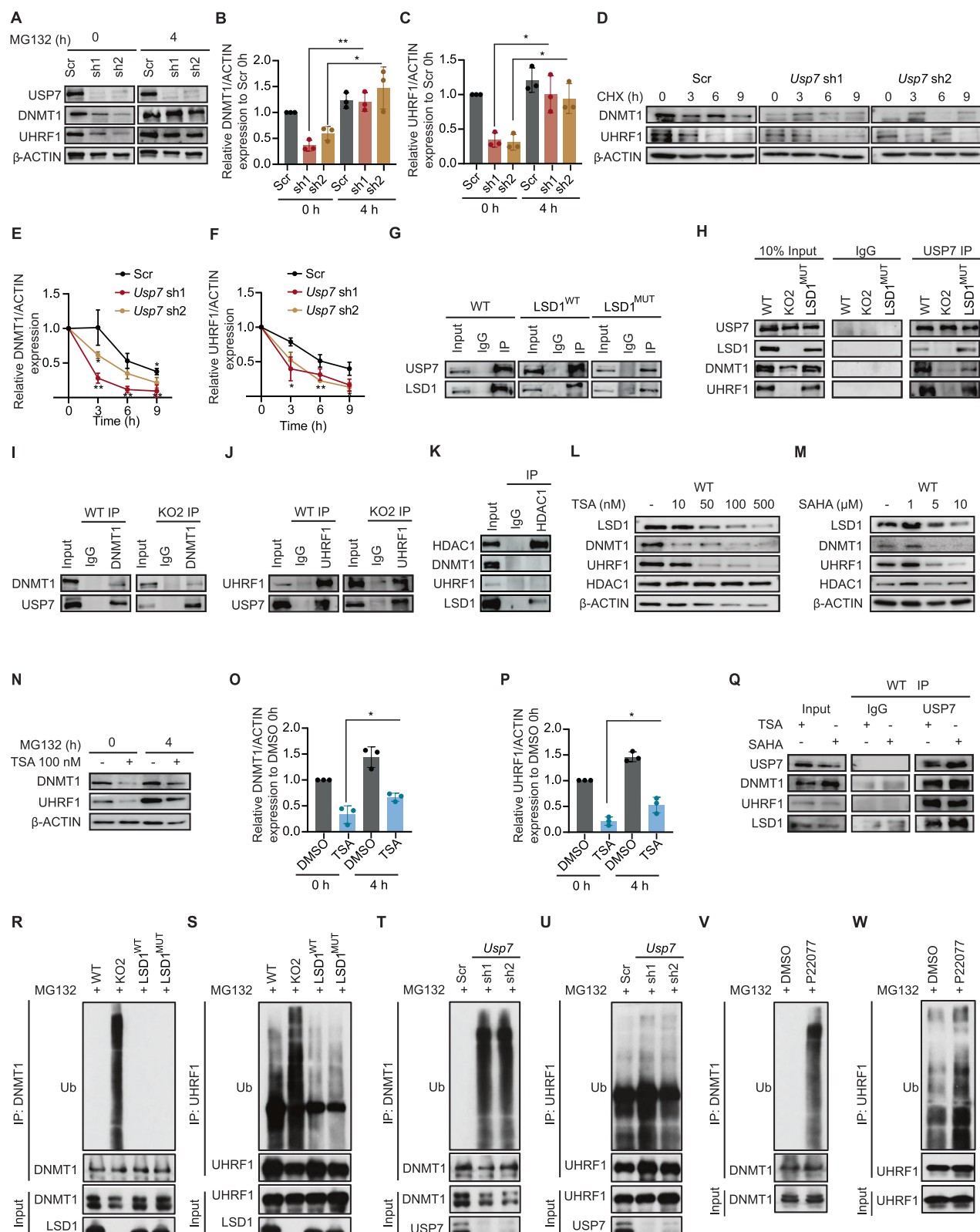

required for proper differentiation. In mouse ESCs, ablation of LSD1 results in decreased DNMT1 and UHRF1 proteins coupled with global hypomethylation. In this scenario, the catalytic activity of LSD1 is not essential as *Lsd1* KO with reintroduction of WT LSD1 or catalytically-impaired LSD1 recover the amount of DNMT1 and UHRF1 protein, and DNA methylation levels. Our studies are consistent with a model, as schematically illustrated in Fig. 8, in which LSD1 facilitates

deacetylation and deubiquitylation of DNMT1 and UHRF1. This involves interactions with HDAC1 and USP7, ultimately enhancing UHRF1 and DNMT1 protein stability. Our studies elucidate previously unidentified mechanism by which the LSD1-HDAC1-USP7 axis coordinates DNA methylation maintenance in mouse ESCs.

LSD1 is required for early embryogenesis[6,7]; however, our data show that LSD1 is dispensable for mouse ESC self-renewal. In this

**Fig. 7 | LSD1 promotes USP7-mediated DNMT1 and UHRF1 deubiquitination.**
**A** Western blots of USP7, DNMT1, and UHRF1 in the scramble and knockdown of *Usp7* at 0 and 4 h after MG132 treatment. β-ACTIN is used as the loading control. Relative (**B**) DNMT1 and (**C**) UHRF1 protein retrieval after quantification and normalization of band from (**A**) in *Usp7*-depleted mouse ESCs compared to scramble. The quantification of samples derive from the same experiment and blots were processed in parallel. **D** Western blotting of DNMT1 and UHRF1 on the WCE of WT and *Usp7*-depleted mouse ESCs during a 9 h CHX time course treatment. β-ACTIN is used as the loading control. Protein stability curves of (**E**) DNMT1 and (**F**) UHRF1 were generated from the measurement and normalization of bands from (**D**) upon depletion of USP7 in mouse ESCs. **G** USP7 immunoprecipitation in WT, LSD1^WT, and LSD1^MUT mouse ESCs followed by immunoblotting of USP7 and LSD1. The percentage of input used is 10%. **H** USP7 immunoprecipitation on the WCE of WT, *Lsd1* KO, and LSD1^MUT mouse ESCs followed by LSD1, DNMT1, and UHRF1 immunoblotting. The percentage of input used is 10%. **I** Immunoprecipitation of DNMT1 on the WCE of WT and DNMT1- inducible *Lsd1* KO mouse ESCs followed by USP7 immunoblotting. The percentage of input used is 10%. **J** UHRF1 immunoprecipitation on the WCE of WT and UHRF1- inducible *Lsd1* KO mouse ESCs followed USP7 immunoblotting. The percentage of input used is 10%. Immunoprecipitation of HDAC1 on the WCE of WT mouse ESCs followed by DNMT1, UHRF1, and LSD1 immunoblotting. The percentage of input used is 10%. **K** Western blotting of LSD1, DNMT1, UHRF1, and HDAC1 on the WCE of (**L**) TSA (**M**) SAHA-treated WT mouse ESCs at different

concentrations. β-ACTIN is used as the loading control. **N** Western blotting of DNMT1 and UHRF1 on the WCE of DMSO and TSA-treated WT mouse ESCs on the indicated time points of MG-123 treatment. β-ACTIN is used as the loading control. Bar graph depicting the protein recovery of (**O**) DNMT1 and (**P**) UHRF1 in TSA-treated WT mouse ESCs after quantification and normalization of bands from (**N**) compared to DMSO treated WT ESCs. **Q** USP7 immunoprecipitation on the WCE of TSA and SAHA-treated WT mouse ESCs followed by LSD1, DNMT1, and UHRF1 immunoblotting. The percentage of input used is 10%. Immunoprecipitation of endogenous DNMT1 (**R**) and UHRF1 (**S**) on the WCE from MG-132 treated WT, *Lsd1* KO2, LSD1^WT, and LSD1^MUT mouse ESCs, followed by immunoblotting for ubiquitin. Immunoprecipitation of endogenous DNMT1 (**T**) and UHRF1 (**U**) on the WCE from MG-132 treated scramble and *Usp7*-depleted mouse ESCs, followed by immunoblotting for ubiquitin. Immunoprecipitation of endogenous DNMT1 (**V**) and UHRF1 (**W**) on the WCE from WT mouse ESCs treated with DMSO or PD22077 in the presence of MG-132, followed by ubiquitin immunoblotting. Statistical analysis: Two tailed unpaired t-test (**B, C, E, F, O** and **P**). $*p < 0.05$, and $**p < 0.01$. Error bars denote mean ± SD; n = (E (Scr = 4, *Usp7* sh1 = 3 and *Usp7* sh2 = 3) and F (Scr = 3, *Usp7* sh1 = 3, and *Usp7* sh2 = 3)). Each dot in the bar graphs represents independent biological replicates (**B, C, O** and **P**). The exact *P*-values for panels (**B, C, E, F, O** and **P**) are represented in the source data. Results are one representative of n = 3 independent biological experiments (**A, D, G–N**, and **Q–W**). Uncropped blots are represented in the source data.

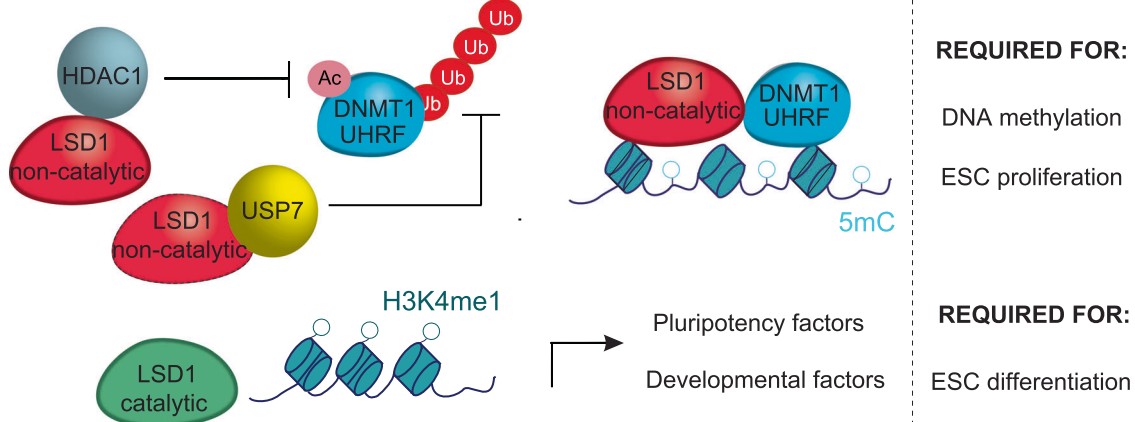

**Fig. 8 | Graphical illustration of the model.** LSD1 and LSD1^MUT (non-catalytic) interacts with HDAC1 to deacetylate DNMT1 and UHRF1, promoting their stability. Non-catalytic LSD1 also interacts with USP7 to facilitate DNMT1 and UHRF1 deubiquitination and stability. Additionally, non-catalytic LSD1 can also recruit DNMT1 at specific loci. Non-catalytic LSD1 is required for DNA methylation and mouse ESC proliferation. LSD1 catalytic activity is required for H3K4me1 demethylation at both pluripotency and developmental enhancers. Such demethylase activity is required for proper mouse ESC differentiation.

study, we have identified more than 3000 common differentially expressed genes in *Lsd1* KO mouse ESCs which are not related to stemness or differentiation. Specifically, upregulated genes were enriched for biological processes linked to cell proliferation and apoptosis, consistent with the defective proliferation phenotype observed upon LSD1 loss. On the other hand, downregulated genes were related to metabolic processes. Studies have demonstrated that ablation of *Lsd1* results in enhanced oxidative capacity in hepatocarcinoma cells and differentiating myoblast, while the reversed metabolic switch is observed during reprogramming[54,55]. Here, we reveal that deletion of *Lsd1* leads to reduced glycolytic activity without a substantial increase in oxidative respiration in mouse ESCs. Whether this LSD1-mediated metabolic switch has implications during mouse ESC differentiation needs to be further addressed.

To define the role of LSD1 in cell-fate commitment, we assessed the differentiation potential of *Lsd1* KO mouse ESCs using two independent in vitro differentiation methods: (i) EB-directed differentiation and (ii) gastruloids generation. *Lsd1* KO mouse ESCs formed EBs albeit significantly smaller than WT mouse ESCs. In addition, *Lsd1* KO mouse ESCs were unable to form proper gastruloids. Indeed, EBs and gastruloids

depleted of LSD1 showed severely compromised expression of lineage determination markers and failed to silence the pluripotency program.

The histone-demethylase independent activity of LSD1 in regulating gene expression programs have recently begun to unravel. Hence, LSD1 catalytic activity has been reported to be dispensable in promoting tumorigenesis in several cancer models where it seems to play a central role as a scaffold for assembling chromatin modifiers and transcription factor complexes[27,56,57]. To investigate the relative contributions of LSD1 enzymatic activity and scaffolding functions in pluripotency, we generated double point mutated (A540E/K662A) LSD1 mouse ESCs (LSD1^MUT), as a recent study has shown that single mutation (K661A) of human LSD1 (K662A in mouse), which has been widely used as a surrogate of a catalytically inactive LSD1, possesses significant H3K4 demethylase activity on nucleosome substrates[29]. Herein, we demonstrate that reconstitution of *Lsd1* KO mouse ESCs with either WT or a catalytically inactive LSD1 recovers the proliferation defect of mouse ESCs. Consistent with this finding, transcriptome profiling revealed a less pronounced impact of LSD1 catalytic inactivation on gene expression in mouse ESCs compared to *Lsd1* KO, aligning with the scaffolding function of LSD1. Nevertheless, EBs

derived from LSD1[MUT], akin to *Lsd1* KO EBs, failed to promote proper differentiation.

Ablation of LSD1 led to a global increase of H3K4me1; however, such increase of H3K4me1 in regulatory regions is not sufficient to trigger major gene expression changes. Indeed, other studies have found that this histone mark has only a minor effect on the maintenance of enhancer activity and function[58–60], but that enhancers acquiring H3K4me1 are primed for transcription activation upon differentiation[61]. Thus, the retention of H3K4me1 at pluripotency enhancers in *Lsd1* KO mouse ESCs could explain why pluripotency genes do not undergo silencing upon EB and gastruloid formation, in agreement with previous observations[16]. However, our findings substantially differ from the aforementioned study as we also observed LSD1 occupancy at a subset of genes that are induced during neuronal differentiation, experiencing these genes an increase of H3K4me1 deposition upon LSD1 loss. What is the role of LSD1-dependent H3K4me1 demethylation in these neural markers? One possibility would be that those enhancers are also poised in mouse ESCs and might be subjected to a context-dependent expression pattern, wherein they undergo activation under neuronal differentiation and repression when differentiating onto other lineage programs. While we noted an increase in H3K4me1 peaks in enhancers and super-enhancers co-occupied with LSD1 following deletion and inactivation of LSD1, the enrichment of neurogenesis-related categories was specific to *Lsd1* KO mouse ESCs. These findings suggest that LSD1 may recruit an additional factor that contributes to the specificity of the H3K4me1 mark at genes related to the neural fate determination. Nevertheless, whether H3K4me1 has a regulatory function within enhancers or is simply a useful mark to identify them remains unclear[62].

In addition, we found that knockout of *Lsd1* can elicit genome-wide loss of DNA methylation. In mammals, DNA and histone methylation are highly interrelated to maintain cellular epigenomic landscapes[63]. Hence, repressive histone marks, such as H3K9me3, usually coexist with DNA methylation to generate local formation of heterochromatin. Additionally, there is a strong anti-correlation between CpG methylation and H3K4 methylation that is particularly pronounced at CpG islands[64]. Particularly in our research context, it has been shown that LSD1-mediated demethylation of H3K4me1 is critical in guiding de novo DNA methylation at enhancers of pluripotency genes[23]. We speculate that some degree of DNA hypomethylation in *Lsd1* KO mouse ESCs may be attributed to increased H3K4me1 levels that these cells exhibit.

LSD1-mediated demethylation of DNMT1 and UHRF1, which has been reported to impede their proteasome degradation[6,20], has been proposed as an alternative mechanism underlying the molecular crosstalk between DNA and histone methylation. However, our evidence strongly indicates that both LSD1[WT] and LSD1[MUT], although to a less extent, are able to recover DNA methylation levels by two non-mutually exclusive mechanisms: (i) maintaining DNMT1 and UHRF1 protein stability and (ii) recruiting DNMT1 at specific genomic loci, suggesting that the demethylase activity of LSD1 is dispensable to maintain DNA methylation in mouse ESCs. In line with this, targeting LSD1 with the catalytic-specific irreversible inhibitor GSK_LSD1 had no impact on DNMT1 and UHRF1 protein levels, and global DNA methylation. Recently, UM171, a pyrimidoindole derivative, has been found to promote the degradation of the LSD1-containing chromatin remodeling complex CoREST in hematopoietic stem cells[65]. This compound provides a potential means to investigate how LSD1 functions as a scaffold in chromatin dynamics. However, it is worth noting that the LSD1-mediated stability of DNMT1 and UHRF1 operates independently of RCOR1 and RCOR2.

Lysine methylation can modulate numerous molecular processes by fine-tuning non-histone protein function[66]. Amongst them, lysine methylation has been shown to regulate protein stability. Hence, several studies have shown that lysine methylation can either block polyubiquitination-dependent proteasomal degradation or promote ubiquitination by recruiting the ubiquitin ligase machinery. Even though emerging evidence has revealed that LSD1 can demethylate non-histone proteins[8], structural biology studies show that there is insufficient space in the catalytic center of LSD1 to accommodate more than three residues N-terminal to the target lysine residue[29,67]. It is worth noting that the original study demonstrating that the catalytic activity of LSD1 controls DNA methylation by regulating DNMT1 protein stability relied largely on one assay using Pargyline[6]. Pargyline is an irreversible selective monoamine oxidase-B inhibitor that leads to LSD1 degradation, thereby affecting both catalytic and non-catalytic functions of LSD1 (Supplementary Fig. 9Z; left and right panel). Albeit we cannot rule out the possibility that a lysine-methylation-dependent proteolytic mechanism controls DNMT1 and UHRF1 protein stability, pathways involving lysine demethylation seem to operate independently of LSD1. Therefore, we propose that LSD1 facilitates deubiquitylation independent of its demethylase activity in mouse ESCs as both WT and LSD1[MUT] can restore DNMT1 and UHRF1 protein levels in *Lsd1* KO ESCs. Whether DNMT1 and UHRF1 methylation status is required for LSD1 recognition needs to be further addressed. A similar example can be found in the serine/threonine kinase AKT by which AKT K64 methylation is required to trigger ubiquitination of AKT following growth factor stimulation. The interaction between AKT and the E3 ligases is mediated through the histone demethylase JMJD2A, which recognizes methylated AKT independent of its catalytic activity[68].

While it has been documented that DNMT1 and UHRF1 directly interact with USP7, we have not observed such interaction in the absence of LSD1, except when DNMT1 and UHRF1 are overexpressed[53,69]. This suggests a potential role for LSD1 in recruiting an additional protein that facilitates post-translational modifications to enhance USP7 activity. HDAC1 emerges as a plausible candidate since it interacts with LSD1 but not with DNMT1 and UHRF1. Moreover, HDAC1 deacetylates DNMT1 and UHRF1, potentially priming them for deubiquitination by USP7. However, it is worth noting that acetylation and deubiquitination, although contradictory in their effects on UHRF1 and DNMT1 protein stabilities[70], are not mutually exclusive in our model and both can occur simultaneously. Nevertheless, the scenario is further complex as both LSD1 and USP7 protein levels are affected by HDAC inhibitors, which would reinforce the degradation loop[53]. Moreover, USP7 has been shown to deubiquitinate Tip60, the acetyltransferase that promotes acetylation of DNMT1 and UHRF1, triggering their ubiquitination[45]. Furthermore, UHRF1 acts as the ubiquitin ligase for DNMT1[45,70,71], and many of these interactions, such as those between USP7 and DNMT1, may be regulated in specific cell cycle phases, adding another layer of complexity to the mechanism[44,72]. Further investigation is warranted to elucidate these intricate regulatory pathways. Nonetheless, our findings demonstrate that DNMT1 and UHRF1 exhibit increased polyubiquitination in *Lsd1* knockout mouse ESCs but not in LSD1[MUT], indicating that LSD1 acts as a scaffolding protein to regulate their deubiquitination and stability independently of LSD1 lysine demethylase activity.

In summary, we demonstrate that both catalysis of histone demethylation and scaffolding function of LSD1 may be variably important for control gene expression depending on the cellular context i.e., in pluripotency or during mouse ESC differentiation. Our results prompt a re-evaluation of the proposed mechanism of action for LSD1 in demethylating non-histone substrates, especially DNMT1 and UHRF1, to increase their stability. They also bring light to LSD1-HDAC1-USP7 axis to coordinate DNA methylation maintenance in mouse ESCs.

## Limitations of the study

The proteomics analysis indicated both stabilizing and destabilizing effects of LSD1 on the global protein expression profile. However, the specific role of LSD1 in destabilization was not further analyzed in our

study. Understanding the mechanisms by which LSD1 mediates protein destabilization would provide valuable information on the broader regulatory functions of LSD1 in pluripotency. Secondly, in addition to USP7-mediated regulation, we cannot rule out the possibility that other protein deubiquitinases are involved in promoting the LSD1-mediated stability of DNMT1 and UHRF1. Finally, although we propose that DNMT1 and UHRF1 are not substrates of LSD1, we cannot rule out the possibility that DNMT1 and UHRF1 need to be methylated in order to be recognized by LSD1 and guide them for deubiquitination. LSD1 might exhibit similar non-catalytic functions in the methylation of human ESCs, but this requires validation in future studies.

## Methods

### Antibodies

The following commercially available antibodies were used for western blot: anti-LSD1 (Abcam, ab17721, 1:2500), anti-DNMT1 (Abcam, ab188453, 1:2500), anti-UHRF1 (Invitrogen, PA5-29884, 1:2000), anti-DNMT3A (Abcam, ab188470, 1:2500), anti-DNMT3B (Abcam, ab79822, 1:2000), anti-DNMT3L (Abcam, ab194094, 1:2000), anti-OCT4 (Santa-Cruz, SC-8628, 1:1500), anti-MYC (Cell Signaling, 2276S, 1:4000), anti-βACTIN (Sigma, A54411:5000), anti-H3 (Abcam, ab8895, 1:8000), anti-USP7 (Invitrogen, PA5-34911, 1:2000), anti-RCOR1 (Novus biologicals, NBP3-16225, 1:1500), anti-RCOR2 (Proteintech, 23969-1-AP, 1:2000), and anti-HDAC1 (Abcam, ab19845, 1:2500). ChIP was performed with anti-H3K4me1 (Abcam, ab8895, 2ug), anti-LSD1 (Abcam, ab17721, 10ug), and anti-DNMT1 (Abcam, ab19905, 10ug). Co-IP experiments were performed with the following antibodies: LSD1 (Abcam, ab17721, 5ug), anti-RCOR1 (Novus biologicals, NBP3-16225, 5 ug), anti-RCOR2 (Proteintech, 23969-1-AP, 5 ug), anti-HDAC1 (Abcam, ab19845, 4 ug), anti-USP7 (Invitrogen, PA5-34911, 4 ug), anti-lysine acetyl (Abcam, 9814S) and anti-Rabbit IgG (Abcam, ab37415, 5ug). Dot-blot experiments were performed with anti-5hmC (Active Motif, 39791, 1:4000) and anti-5mC (Active Motif, 39649, 1:4000). For IF staining we used anti-LSD1 (Abcam, ab129195, 1:500), anti-5hmC (Active Motif, 39791, 1:250), anti-5mC (Active Motif, 39649 1:250), anti-OCT4 (Santa-Cruz, SC-8628, 1:250), anti-SSEA1 (Thermo Scientific, MA5-17042, 1:250), anti-Rabbit (Thermo Scientific, A-11011, 1:1000), and anti-Mouse (Thermo Scientific, A11029, 1:1000).

### Cell culture

CCE murine ESCs were maintained on 0.1% gelatin-coated tissue culture plates under feeder-free culture conditions. The complete media composition consists of Dulbecco's modified Eagle's medium (DMEM) high glucose (Gibco, 41966-029), 15% fetal bovine serum (FBS; Gibco, 10500-064), 1% MEM non-essential amino acids (Sigma-Aldrich, M7145), 0.1 mM of 2-β mercaptoethanol, 1% l-glutamine (Hyclone, SH30034.01) and 1% penicillin/streptomycin (Gibco, 15140-122). Complete media was supplemented with Leukemia inhibitory factor (LIF; R&D systems, 8878-LIF-100/CF, 0.01 ng/μL). All cell cultures were maintained at 37 °C with 5% $CO_2$.

Mouse ESCs at a density of $1 \times 10^5$ per well in the 2iL medium were grown in a 1:1 mix of DMEM/F12 (Gibco) and Neurobasal medium (Gibco) with 1x N2-Supplement (Gibco, Cat. 17502048), 1x B27 minus insulin (Gibco, Cat. A1895601), 0.05% BSA (Gibco), Leukemia inhibitory factor (LIF; R&D systems), 2 mM Glutamine (Gibco), 1% penicillin/streptomycin (Gibco), 1 μM PD03259010 (Mek Inhibitor; MedChemExpress), 3 μM CHIR99021 (GSK3b Inhibitor; Stem cell technology), $1.4 \times 10^{-4}$ M Monothioglycerol (Sigma). The media was replenished every 48 h.

### Lentiviral production and generation of *Usp7*, *Rcor1* and *Rcor2* KD mouse ESCs

Lentiviral particles were produced by transfecting HEK-293 T with a series of lentiviral plasmids: (i) pLKO.1-Puro containing shRNAs against *Usp7*; (ii) pLKO.1-Puro containing shRNA against *Rcor1;* and (iii) pLKO.1-Puro containing shRNA against *Rcor2* (Supplementary Data 11) together with the packaging vector pCMV-dR8.2 and the helper plasmid pCMV-VSV-G using jet-PEI (Polyplus) following the manufacturer's instructions. After 48 h of incubation, lentiviral supernatants were harvested and concentrated using Amicon Ultra-15 Centrifugal Filter Units (Merck).

Mouse ESCs were then infected with the lentiviral particles containing shRNA targeting *Usp7*, *Rcor1*, and *Rcor2 in* complete media supplemented with polybrene (8 μg/ml) for 24 h followed by puromycin selection (2 μg/ml).

### Generation of stable cell line expressing LSD1[A540E], LSD1[K662A], LSD1[MUT,] and LSD1[WT] in *Lsd1* KO ESCs

To generate the rescue construct, the cDNA of *Lsd1* isoform 2 from mouse ESCs was amplified with RevertAid® First Strand cDNA Synthesis Kit and cloned into a pGEM®-T easy vector (Promega), and ultimately subcloned into the pSIN-E2F_MYC (Sigma-Aldrich) vector using EcoRI as restriction enzymes. All primers used for cloning purposes are described in Supplementary Data 11. A single and double point mutated (A540E/K662A) mouse LSD1 construct was generated using Quik-Change Multi Site-Directed Mutagenesis Kit following the manufacturer's protocol.

Constructs containing both single and double mutation in LSD1 and full-length wild-type LSD1 together with the packaging vector pCMV-dR8.2 and the helper plasmid pCMV-VSV-G were transfected in HEK-293T to produce lentiviral particles as described above. Then, *Lsd1* KO ESCs were transduced with these lentiviral particles to generate stable cell lines LSD1[A540E], LSD1[K662A], LSD1[MUT,] and LSD1[WT] mouse ESCs, respectively.

### Generation of DNMT1 and UHRF1-inducible *Lsd1* KO ESCs

To generate the doxycycline-inducible cell lines expressing DNMT1 and UHRF1 in *Lsd1* KO mouse ESCs, the cDNA of *Dnmt1 and Uhrf1* from mouse ESCs was amplified with Phusion polymerase and cloned into the Lenti-iCas9-neo (Addgene, 85400) using BstXI and Xhol as restriction enzymes.

*Lsd1* KO2 ESCs were transduced with concentrated lentiviral particles containing Lenti-Flag DNMT1 and Lenti-Flag UHRF1 constructs in complete media supplemented with polybrene (8 μg/ml) for 24 h, respectively. After transduction, cells were treated with neomycin (4 mg/ml) for 6 days to obtain a stable cell line. Following neomycin selection, $5 \times 10^4$ cells of *Lsd1* KO mouse ESCs were seeded in a 6-well plate with complete media supplemented with doxycycline (at a concentration of 0.5 μg/μL for UHRF1 induction and 1 μg/μL for DNMT1 induction respectively). The media supplemented with doxycycline was replenished every 48 h.

### RA and EB differentiation assays

To induce retinoic acid (RA)-mediated differentiation, $2.1 \times 10^3$ cells/cm² of mouse ESCs were seeded onto tissue culture plates pre-coated with 0.1% gelatin in complete media without LIF and with 5 μM RA (Sigma-Aldrich).

For EB differentiation, mouse ESCs at a density of $8.8 \times 10^4$ cells/cm² were seeded in low attachment plates in the presence of a complete medium without LIF. The medium was replenished every 48 h. EB sizes were quantified using NIS-elements software.

### Generation of CRISPR/Cas9 Knockouts

Mouse ESCs were seeded in a 12 wells-plate and transfected with 0.8 μg of Cas9 expression vector PX459 (Addgene plasmid #62988) containing the corresponding sgRNAs using Lipofectamine (Invitrogen). All sgRNA-Cas9 plasmids were obtained by ligation (T7 DNA ligase, Fermentas) of annealed complementary oligonucleotides with the

PX459 vector (digested with BbsI (BpiI) (Thermo Scientific)). All sgRNAs were designed using the Zhang lab's online tool. Knockouts (KOs) were screened by PCR and validated by western blot and Sanger sequencing. The specific primers flanking the cleavage sites were designed to detect the insertion/deletion (INDEL) in the target regions (Supplementary Data 11).

## Mismatch detection assay by T7 endonuclease of *Lsd1* knockouts

Genomic DNA was extracted using the GeneJET Genomic DNA Purification kit (Thermo Scientific, K0722) following the manufacturer's instructions. Target regions were PCR-amplified (Supplementary Data 11) with DreamTaq 2X Master Mix (Thermo Scientific, K1081). PCR products were denatured at 95 °C for 2 min and re-annealed at −2 °C /sec temperature ramp to 85 °C, followed by a −0.1 °C /sec temperature ramp to 25 °C. The PCR products (20 µL) were incubated with 5 U T7E1 enzyme (NEB #E3321) at 37 °C for 20 min. Products from mismatch assays were separated by electrophoresis on a 2% agarose gel in TAE.

## Alkaline phosphatase activity

Alkaline phosphatase (AP) activity was measured using the Stemgent Alkaline Phosphatase Staining kit (Stemgent), following the manufacturer's recommendations.

## Immunofluorescence staining

For immunofluorescence staining, cells were fixed in 4% paraformaldehyde (PFA) at room temperature (RT) for 15 min, followed by permeabilization at RT for 30 min with 0.25% Tween-20 in PBS (PBST). Cells were then washed twice with 1X PBST and blocked at 10% in normal goat serum (Invitrogen), 1% bovine albumin serum (Hyclone), and 0.05% Tween 20) at RT for 1 h. The cells were stained with the specified primary antibodies overnight at 4 °C in a blocking buffer followed by secondary antibodies staining at RT for 1 h. 6-diamino-2-phenylindole (DAPI) was used for nuclear DNA staining (4 min). Images were acquired using a Zeiss microscope.

## RT-qPCR analysis

Total RNA was extracted using the RNeasy Mini Kit (Qiagen). 1 µg of total RNA was reverse transcribed using the RevertAid First Strand cDNA Synthesis kit (Invitrogen). Quantitative PCR (qPCR) was performed using the Power Up SYBR Green qPCR Master Mix (Applied Biosystems). Gene expression-specific primers used for this study are listed in Supplementary Data 11.

## RNA-Seq library preparation

RNA-Seq library preparation was performed at Novogene (Hong Kong). Samples were sequenced by the Illumina HiSeqTM platform (Illumina) as 100 bp pair-ended reads.

## RNA-Seq analysis

RNA-Seq reads were filtered and aligned to the mouse transcriptome (mm10). Differentially expressed genes were identified by assuming a negative binomial distribution from reads (removing transcripts that are expressed on only two or less samples)[73] using a false discovery rate (FDR) < 0.01 and fold-change >1.5. All the process was done by built-in libraries in MATLAB.

## Cellular proliferation

$2 \times 10^4$ cells were plated in 6-well plate and counted every second day using trypan blue (BioRad).

## Apoptosis assay, and cell cycle analyses

Apoptosis assay, and cell cycle analysis were performed using Muse™ Cell Analyzer from Millipore following the manufacturer's instructions.

## MTT assay

Mouse ESCs at a density of $2 \times 10^4$ cells were plated in a 6-well plate with 2iL medium. MTT, at a final concentration of 0.5 µg/µL was added to cells and incubated at 37 °C for 4 h after 48 h and 96 h of culture, respectively. Purple formazan crystals formed after incubation were dissolved in DMSO, and the absorbance of samples was measured at wavelength 570 nm. The cell proliferation of *Lsd1* KO and mutated LSD1 cells was calculated relative to WT mouse ESCs.

## Metabolic analysis

Glycolysis and oxygen phosphorylation rate were measured using Agilent Seahorse XF Glycolysis Stress Test Kit and Agilent Seahorse XF Cell Mito Stress Test Kit by evaluating extracellular acidification rate and oxygen consumption rate, respectively, according to the manufacturer's instructions.

## Gastruloids aggregation assay

Gastruloids were generated following the previously described protocol[31,74]. In brief, 300 mouse ESCs were seeded onto low attachment u-bottomed 96-well plates in 40 µl N2B27 media (50% Neurobasal medium, Gibco), 50% advanced DMEM (Gibco), 1x N2 (Gibco, 17502048), 1x B27 (Gibco, 7504044), 2 mM glutamine (Hyclone, SH30034.01), 1x penicillin/streptomycin (Gibco, 15140-122), and 0.1 mM of 2-β mercaptoethanol. After 48 h of incubation, 3 µM CHIR (4423, Tocris Biosciences) was added for 12 h and replenished with N2B27 media. Gastruloids were harvested for total RNA and total protein after 120 h of aggregate formation. Media was changed every day and cells were maintained at 37 °C with 5% CO2. The gastruloids length was quantified using NIS-elements software.

## Whole cell extract preparation

Cells were washed with cold 1X PBS, pelleted, and incubated with lysis buffer containing 50 mM Hepes pH 7.5, 150 mM NaCl, 3 mM $MgCl_2$, 0,2% Triton X-100 0.2% Nonidet NP-40, 10% glycerol supplemented with a protease inhibitor in ice for 15 min. Then, lysates were sonicated for ten cycles with 30 s pulses on/off and centrifuged at 162000 g for 15 min. The supernatant containing the whole cell extract was frozen at −80 °C for further analysis.

## Nuclear extraction

Cells were washed with cold PBS, scraped off, and pelleted. The pellet was resuspended in at least 5 volumes of buffer A (10 mM HEPES pH 7.9, 1.5 mM $MgCl_2$, 10 mM KCl, 2 mM DTT) in the presence of protease inhibitors (Fisher Scientific) and incubated for 10 min on ice. After centrifugation, the pellet was resuspended in 2 volumes of buffer A, douncer-homogenized 10 times, and centrifuged at maximum speed for 10 min. Nuclei pellets were then resuspended in 2 volumes of buffer B (20 mM HEPES pH 7.9, 1.5 mM $MgCl_2$, 500 mM NaCl, 25% Glycerol, 0.5 mM EDTA, 1 mM DTT) supplemented with protease inhibitors, incubated on a rotator at 4 °C for at least 30 min, and at maximum speed for 20 min. The supernatant containing the nuclear fraction was frozen at −80 °C for further analysis.

## Subcellular protein fractionation

The cytosolic, soluble nuclear, and chromatin-bound proteins were separated following the manufacturer's protocol of the Subcellular protein fractionation kit (Thermo Scientific).

## Co-immunoprecipitation and immunoblotting

For immunoprecipitation experiments, 1 mg of nuclear extracts were pre-cleared with protein A magnetic beads (BioRad) for 1 h at 4 °C and incubated overnight on a rotator with specific antibodies at 4 °C. Following this, protein A magnetic beads were added for 3 h before washing 4 times in ice-cold IP buffer (10 mM Tris; adjust to pH 7.4, 1 mM EDTA, 1 mM EGTA; pH 8.0, 150 mM NaCl, 1% Triton X-100,

0.2 mM sodium orthovanadate) supplemented with protease inhibitors. For RNA and DNA-dependent protein interaction, nuclear extracts were treated with RNase A (100 μg/ml; 30 min at room temperature) or DNase I (0.1 U/μl; 30 min at 37 °C) were added before the immunoprecipitation procedure. DNase I was inactivated with 10 mM EDTA treatment before IP. Immunoprecipitated complexes were resolved by SDS-PAGE, transferred to nitrocellulose membranes (Invitrogen), and immunoblotted with the indicated antibodies, followed by ECL detection (Thermo Scientific).

### Ubiquitination assay
To assess the ubiquitination of UHRF1 and DNMT1, mouse ESC lines, including WT, *Lsd1* KO2, LSD1[WT], and LSD1[MUT], as well as those subjected to *Usp7* knockdown and pre-treated with 25 μM P22077 for 24 h, were exposed to 10 μM MG132 for 6 h. Total cell lysates were then prepared using RIPA-B buffer supplemented with protease inhibitors[75]. Endogenous UHRF1 and DNMT1 were precipitated using anti-UHRF1 (Santa Cruz, sc-373750) and anti-DNMT1 antibodies (BioAcademia, 70-201), respectively, while ubiquitination was assessed by immunoblotting with an anti-ubiquitin antibody (Cell Signaling, 3936).

### DNA dot blots
Indicated amounts of genomic DNA were denatured in denaturing buffer (200 mM NaOH, 20 mM EDTA) at 95 °C for 5 min and neutralized with an equal volume of neutralizing agent (2 M NH₄CH₃CO₂). Nitrocellulose membrane (Thermofisher Scientific) was hydrated in 6X SSC and then "sandwiched" in a Minifold 1 Filtration Manifold (GE Healthcare). Each well was equilibrated with 200 μl of 10x SSC and flushed by gentle suction vacuum twice. The membrane was UV-crosslinked using the default settings. After blocking for 1 h with 5% milk in PBST, the membrane was incubated with 5mC or 5hmC primary antibodies overnight at 4 °C, washed four times in PBST, and incubated with the secondary anti-rabbit antibody for 1 h at RT. The membrane was washed again in 1x PBST and the detection was done using the enhanced chemiluminescence substrate (ECL, Thermo Scientific). DNA levels were normalized with methylene blue staining.

### Whole-genome bisulfite sequencing analysis
Whole-genome bisulfite was performed at Novogene. Paired-end reads from Illumina were aligned to the mouse (mm10) genome using the BWA-meth algorithm[76], and the resulting BAM reads were processed for methylation calling using MethylDackel. We extracted subsets of regions from the final bedgraph methylation file with bedtools[77], as well as other BED files obtained from the UCSC Genome Browser. Statistical analyses and visualization were performed using MATLAB.

### Quantitative analysis of DNA methylation levels using LC-MS/MS
Quantification of 5mC and 5hmC was performed according to the previously described protocol[78]. Genomic DNA was extracted using the GeneJet DNA purification Kit (Thermofisher Scientific). In brief, extracted DNA was desalted with vertical ultrafiltration and digested by DNase I (New England Biolabs, Ipswich, MA, U.S.A.), 0.02 U snake venom phosphodiesterase (SVP; Worthington Biochemical Corporation, Lakewood, CO, U.S.A.), and 5.0 U calf intestine alkaline phosphatase (CIP; New England Biolabs, Ipswich, MA, U.S.A.) and incubated at 37 °C overnight. The digested DNA solutions were filtered by ultrafiltration tubes and then subjected to LC-MS/MS analysis for detection of 5mC and 5hmC.

### Mouse DNA methylation beadChip Array
Samples were bisulfite converted using EZ DNA Methylation-Gold™ Kit (Zymo Research, CA, USA) following the manufacturer's protocol with modifications for Illumina Infinium Methylation Assay. Infinium Mouse Methylation BeadChip (Illumina, Inc., San Diego, CA, USA) arrays were used to profile genome-wide DNA methylation. This platform interrogates over 285,000 methylation sites per sample at single-nucleotide resolution[79]. Samples were hybridized in the array following the manufacturer's instructions.

Raw signal intensities were preprocessed and analysed using SeSAMe (v1.14.2)[80]. DNA methylation beta values were obtained from raw IDAT files using the openSesame pipeline within R statistical environment (v4.0.3). Briefly, this pipeline includes a data preprocessing procedure consisting of strain-specific masking for mouse methylation array probes, followed by masking of non-uniquely mapped probes (poor design), channel inference for Infinium I probes, non-linear dye bias correction, detection *p*-value masking ($p > 0.05$) using the pOOBAH algorithm and background subtraction using the noob method[40].

Differential methylation loci were computed using DML function in SeSAMe by modeling DNA methylation values (beta values) using mixed linear models. Loci with a False Discovery Rate (FDR) adjusted $p < 0.05$, and absolute beta value difference between conditions > 0.66 were considered significant. Principal component analysis and the corresponding plot were computed using PCAtools (v.2.8.0) R package. Euclidean distance scores and Ward's minimum variance method were applied to attain hierarchical clustering represented as a heatmap using heatmap.2 function from the gplots (v3.1.3) package in R. Density plots were performed with minfi (v1.42.0) R package, and correlation analyses were computed using Spearman correlation coefficient and plotted using ggplot2 (v3.3.6) package in R. Finally, all downstream analyses were performed within the R statistical environment (v4.0.3).

The DNA methylation analysis was performed using the mm10 mouse genome reference build annotation from the Infinium Mouse Methylation BeadChip Array manifest file (http://zwdzwd.github.io/InfiniumAnnotation#mouse)[40].

### Quantitative analysis of histone modifications using LC-MS/MS
After homogenization of the cells with nuclear extraction buffer (15 mM Tris, 60 mM KCl, 15 mM NaCl, 5 mM MgCl₂, 1 mM CaCl₂, and 250 mM sucrose), histone proteins were extracted as previously described[81]. Briefly, histones were acid extracted from nuclei with 0.2 M H₂SO₄ for 2 h and precipitated with 33% trichloroacetic acid (TCA) overnight. The pellets, containing histone proteins, were dissolved in 30 μL of 50 mM NH₄HCO₃, pH 8.0. Derivatization reagent was prepared by mixing propionic anhydride with acetonitrile in a ratio of 1:3 (v/v), and the reagent was mixed with the histone sample in a ratio of 1:4 (v/v) for 20 min at RT. The reaction was performed twice to ensure derivatization completion. Histones were then digested with trypsin (enzyme: sample ratio 1:20, 6 h, RT) in 50 mM NH₄HCO₃. The derivatization procedure was repeated after digestion to derivatize peptide N-termini.

Prior to mass spectrometry analysis, samples were desalted using a 96-well plate filter (Orochem) packed with 1 mg of Oasis HLB C-18 resin (Waters). Briefly, the samples were resuspended in 100 μl of 0.1% TFA and loaded onto the HLB resin, which was previously equilibrated using 100 μl of the same buffer. After washing with 100 μl of 0.1% TFA, the samples were eluted with a buffer containing 70 μl of 60% acetonitrile and 0.1% TFA and then dried in a vacuum centrifuge.

Thereafter, Samples were resuspended in 10 μl of 0.1% TFA and loaded onto a Dionex RSLC Ultimate 300 (Thermo Scientific), coupled online with an Orbitrap Fusion Lumos (Thermo Scientific). Chromatographic separation was performed with a two-column system, consisting of a C-18 trap cartridge (300 μm ID, 5 mm length) and a picofrit analytical column (75 μm ID, 25 cm length) packed in-house with reversed-phase Repro-Sil Pur C18-AQ 3 μm resin. Peptides were separated using a 30 min gradient from 1-30% buffer B (buffer A: 0.1% formic acid, buffer B: 80% acetonitrile + 0.1% formic acid) at a flow rate of 300 nl/min. The mass spectrometer was set to acquire spectra in a data-independent acquisition (DIA) mode. Briefly, the full MS scan was set to 300-1100 m/z in the orbitrap with a resolution of 120,000

(at 200 m/z) and an AGC target of 5x10e5. MS/MS was performed in the orbitrap with sequential isolation windows of 50 m/z with an AGC target of 2x10e5 and an HCD collision energy of 30.

Histone peptides raw files were imported into EpiProfile 2.0 software[82]. From the extracted ion chromatogram, the area under the curve was obtained and used to estimate the abundance of each peptide. In order to achieve the relative abundance of post-translational modifications (PTMs), the sum of all different modified forms of a histone peptide was considered as 100% and the area of the particular peptide was divided by the total area for that histone peptide in all of its modified forms. The relative ratio of two isobaric forms was estimated by averaging the ratio for each fragment ion with different mass between the two species. The resulting peptide lists generated by EpiProfile were exported to Microsoft Excel and further processed for a detailed analysis.

## Chromatin immunoprecipitation (ChIP)

Mouse ESCs were chemically crosslinked by adding 1/10 volume of fresh 11% formaldehyde solution for 10 min at RT and quenched by adding 1/20 volume of 2.5 M of glycine for 5 min at RT. Cells were then washed twice with ice-cold PBS, scraped and collected by centrifugation. The cell pellet was first lysed in the lysis buffer 1 (50 mM, Hepes-KOH pH 7.5, 140 mM NaCl, 1 mM EDTA, glycerol (10% vol/vol) NP-40 (0.5% vol/vol), Triton X-100 (0.25% vol/vol) for 10 min at 4 °C, followed by lysis in buffer 2 (200 mM NaCl, 1 mM EDTA, 0.5 mM EGTA, 10 mM Tris (pH 8)) with complete protease inhibitors. After lysis, cells were sonicated in lysis buffer 3 (100 mM NaCl, 1 mM EDTA, 0.5 mM EGTA, 10 mM Tris pH 8, Na-Deoxycholate (DOC) (0.1% vol/vol), N-lauroyl sarcosine (0.5% vol/vol) for 16 × 30-second pulses (30-second pause between pulses) at high power in a Bioruptor® Sonication System (Diagenode) and centrifuged for high speed for 15 min. The supernatant containing the chromatin fraction was subjected to immunoprecipitation.

Spike-in control (human MCF10A cell line) was used for normalization of the ChIP-seq reads. For this, human MCF10A chromatin was prepared, as mentioned above. In the mouse ESCs chromatin and antibodies immunoprecipitation complex, 5% of MCF10A (vol/vol) chromatin was added. This complex was mixed with 50 µl of Dynabeads® Protein G magnetic beads and incubated overnight at 4 °C. Protein-DNA bead complexes were washed first with RIPA buffer (50 mM Hepes pH 7.6, 1 mM EDTA, DOC (0.7% vol/vol)), NP-40 (1% vol/vol), 0.5 M LiCl) for 5 times and then, with TE containing 50 mM NaCl. The protein-DNA complexes were eluted from the beads twice by incubation with 100 µL of elution buffer (50 mM Tris pH 8, 10 mM EDTA, SDS (1% vol/vol) at 65 °C for 15 min with shaking. Reverse crosslinking was performed by adding 200 mM NaCl in the eluate and incubating it at 65 °C overnight with shaking. Thereafter, RNA and protein were removed from the samples by treating with RNase A and proteinase K as the manufacturer's recommendations. DNA was subsequently purified by phenol-chloroform extraction, followed by ethanol precipitation The input sample was also treated for crosslink reversal and following steps after this.

## ChIP-seq analysis

Library construction and paired-end read sequencing (20 M per sample) of triplicate immunoprecipitated chromatin and control inputs were performed using Illumina technology at Novogene UK. Raw reads were aligned to the human (hg38) and mouse (mm10) genomes using BWA (v.0.7.18)[76]. The final mouse BAM files were normalized using the number of reads that uniquely mapped to the human genome[83]. Peaks were then called using MACS2 (v.2.2.9)[84] with pooled IPs and inputs (FDR < 0.05), and they were annotated using the chIPseekeR (v.1.28.3)[85] package in R. Subsequent analyses were performed using bedtools (v.2.28)[77] for peak overlapping, HOMER (v. 4.11)[86] for motif detection, and MATLAB (2021b), GBiB (July 2022)[87], and EaSEQ (v.1.2)[88] for visualization.

## CUT&RUN LoV-U

CUT&RUN LoV-U for H3K4me was performed as described in ref. 37 with the following modifications. 3 biological replicates of 250,000 cells/replicate were processed for each condition. Nuclei were pelleted and frozen after extraction in an isopropyl chamber, and then later thawed on ice to be processed for CUT&RUN in parallel. Antibodies used included anti-H3K4me (antibodies online, ABIN3023251) and anti-rabbit IgG (antibodies online, ABIN101961) at 1:100 dilutions. Library preparation was done using the KAPA Hyper Prep Kit for Illumina sequencing (Cat. #KK8504, KAPA Biosystems) following the manufacturer's guidelines with modifications, as described in ref. 37. Libraries were sequenced with 36 bp pair-end reads on the NextSeq 550 (Illumina) using the Illumina NextSeq 500/550 High Output Kit v2.5 (75 cycles) (Cat. #20024906, Illumina).

## CUT&RUN data analysis

Raw reads were trimmed with bbmap bbduk (version 38.18[89],) to remove adapters, artifacts, $[AT]_{18}$, $[TA]_{18}$, and poly G or C repeats. Reads were aligned to the mm10 genome with bowtie2 (version 2.4.5[90],) with the options –local –very-sensitive-local –no-unal –no-mixed -no-discordant –phred33 –dovetail -I 0 -X 500. Samtools (version 1.11[91],) was used to create bam files, fix mate pairing, and for deduplication. Bam files were filtered to remove reads falling within CUT&RUN suspect list regions[92]. For peak calling, visualization and signal graphs, replicate bam files were merged with samtools into a single file. Bedgraphs were created with bedtools (version 2.23.0[77],) genomecov on pair-end mode and default settings. Peaks were called using SEACR (version 1.3[93],) on non and relaxed mode against the negative control, or with GoPeaks[94] on broad mode with default settings against the negative control. Peak overlaps were performed using bedtools. Deeptools (version 3.5.1-0[95],) was used to convert bam files to normalized bigwig files (bamCoverage using -RPGC option), make log2FC difference tracks (bamCompare), and signal intensity plots and profiles (computeMatrix followed by plotHeatmap). Peak-gene annotation was performed using GREAT (version 4.0.4[96],) on default settings. Motif analysis was done with HOMER (version 4.11[86],). GO and KEGG analysis was done with ShinyGo[97].

## Gene ontology (GO) analysis

Gene ontology (GO) analysis was performed using the web tool The Database for Annotation, Visualization and Integrated Discovery (DAVID) (http://david.abcc.ncifcrf.gov/).

## Heatmaps

Heat maps were generated from the log2-transformed expression level for a given gene in a specific sample using Heatmapper.

## His10-SUMO-LSD1 expression and purification

Mouse His10-SUMO-LSD1 constructs were transformed into Rosetta (DE3) and the proteins were expressed by auto-induction media at 20 °C overnight. Cells were lysed, and proteins were purified using NiNTA affinity resin (Thermo Scientific, 88222). Briefly, cells were resuspended in lysis buffer (50 mM NaP 8.0, 500 mM NaCl, 10% glycerol, 0.2% Triton X100, 10 mM Imidazole, 1 mM βME, DNAse) and sonicated (15 cycles 10 on/off). Following centrifugation, the lysate was incubated with NiNTA resin at 4 °C for 1 h. The resin was washed with 25CV wash buffer 1 (50 mM NaP 8.0, 500 mM NaCl, 10% glycerol, 20 mM Imidazole,1 mM βME) followed by 25CV high salt buffer (50 mM NaP 8.0, 1 M NaCl, 10% glycerol, 20 mM Imidazole, 1 mM βME) and finally with 25CV wash buffer 1. The protein was eluted with buffer containing 50 mM NaP 8.0, 300 mM NaCl, 10% glycerol, 250 mM Imidazole, and 1 mM βME.

The His10SUMO-tag was cleaved off using SUMOprotease 1/100 (w/w) in reaction buffer 20 mM NaP 8.0, 150 mM NaCl, 10%glycerol, 1 mM βME, at 4 °C overnight. The tag and protease were

removed by NiNTA affinity purification. The protein was further purified by gel filtration on a Superdex200 column (Cytiva) equilibrated in a 20 mM NaP 8.0, 500 mM NaCl, 5% glycerol, and 1 mM DTT buffer.

## Enzymatic activity of LSD1

The enzymatic activity of the human full-length LSD1/D305CoREST1 complex was determined using a peroxidase-coupled assay[67]. Briefly, 0.3 μM of LSD1 was mixed with the reaction buffer containing 50 mM HEPES pH 8.5, 0.1 mM Amplex Red and 0.3 mM horseradish peroxidase. Then, the reaction mixture was added to serially diluted H3K4me2 peptide (Chinapeptides) from 40 μM to 0.31 μM and incubated at RT for 10 min. The fluorescence signal obtained from the conversion of amplex red to resorufin was measured on a Clariostar plate reader (BMG Labtech) with excitation at 510 nm and emission at 595 nm. The intensity of the fluorescence signal is directly proportional to the demethylase activity. Non-linear regression analysis was performed to calculate $K_{cat}$ and $K_m$ of enzyme activity.

## UHRF1 demethylation assay

Peptides were purchased from Genscript. Activity measurements were performed with peroxidase-coupled assays on a Clariostar plate reader (BMG Labtech). The reactions were carried out in 50 mM HEPES pH 7.5, 0.1 mM Amplex Red, 0.3 mM horseradish peroxidase, 0.3 mM LSD1-CoREST. Peptidic substrates were serially diluted from 40 mM to 0.31 mM. The measured fluorescence signal reflects the enzymatic conversion of Amplex Red to resorufin. LSD1-COREST was tested for activity on UHRF1 peptide at 40 mM.

## LSD1 inhibitors treatment

Mouse ESCs were seeded at a density of $8 \times 10^4$ in a 10 cm plate. After 24 h, cells were treated with a final concentration of 10 μM GSK_LSD1 and 3 mM Pargyline. Cells were harvested for whole cell extracts, chromatin, and RNA following 24 h of treatment.

## CHX treatment and proteasome inhibition

Mouse ESCs were seeded in a 6-well plate at the density of $2 \times 10^5$ cells and after 24 h, cells were treated with CHX at a final concentration of 50 μg/ml for 0, 3, 6, and 9 h, respectively. For proteasome inhibition, cells were incubated with MG-132 at a final concentration of 20 μg/ml for 4 h. Cells were harvested for whole cell extracts on indicated time points, and proteins were subjected to immunoblotting. Blots were quantified using Image Lab software.

## Trichostatin A (TSA) and SAHA treatment

Mouse ESCs were seeded at the density of $1 \times 10^5$ cells in a 6-well plate, and after 24 h, cells were treated with TSA and SAHA at a final concentration of 100 nM and 5 μm for 24 h. The following day, MG-132 at a final concentration of 20 μg/ml were added to cells for 4 h respectively. Cells were harvested for whole cell extracts (immunoblotting) and RNA at specified time points. Blots were quantified using Image Lab software.

## USP7 inhibitor (P22077) treatment

Mouse ESCs were seeded at the density of $1 \times 10^5$ cells in a 6-well plate. After 24 h, the cells underwent a 24-hour treatment with P22077 at a final concentration of 25 μM. CHX was added to cells at 0, 3, 6, and 9 h on the subsequent day at a final 50 μg/ml concentration. For proteasome inhibition, cells were incubated with MG-132 at a final concentration of 20 μg/ml for 4 h. Cells were harvested for whole cell extracts (immunoblotting) at specified time points. Blots were quantified using Image Lab software.

## Mass spectrometry-based proteomics

Mass spectrometry-based proteomics was prepared in four different biological replicates (1 technical replicate of each). 5 μg of whole cell extracts were denatured with a final concentration of 2% SDS and 20 mM TCEP. Samples were digested with a modified sp3 protocol, as previously described[98,99]. Briefly, samples were added to a bead suspension (10 μg of beads (Sera-Mag Speed Beads, 4515-2105-050250, 6515-2105-050250) in 10 μl 15% formic acid and 30 μl ethanol) and incubated shaking for 15 min at room temperature. Beads were then washed four times with 70% ethanol. Proteins were digested overnight by adding 40 μl of 5 mM chloroacetamide, 1.25 mM TCEP, and 200 ng trypsin in 100 mM HEPES pH 8.5. Peptides were eluted from the beads and dried under a vacuum. Peptides were then labeled with TMTpro (Thermo Fisher Scientific), pooled, and desalted with solid-phase extraction using a Waters OASIS HLB Elution Plate (30 μm). Samples were fractionated onto 48 fractions on a reversed-phase C18 system running under high pH conditions, pooling every twelve fractions together. Samples were analyzed by LC-MS/MS using a data-dependent acquisition strategy on a Thermo Fisher Scientific Vanquish Neo LC coupled with a Thermo Fisher Scientific Orbitrap Exploris 480. Liquid chromatography consisted of a 2 h gradient from 2% to 28% B (mobile phase A: 0.1% formic acid; mobile phase B: acetonitrile), with a final wash to 80% B. All LC and MS, acquisition setting details can be obtained from the raw files deposited on PRIDE with the dataset identifier PXD042495. Raw files were processed with MSFragger[100] against a *Mus musculus* FASTA database downloaded from UniProt (UP000000589) using the default workflow for TMT16. Briefly, trypsin is used as an enzyme with a maximum of 2 missed cleavages, variable modifications (M 15.9949, Protein N-terminus 42.0106, peptide N-terminus 304.20715) and fixed modifications (C 57.02146, K 304.20715), mass tolerance for precursors and fragment ions of 20 ppm, minimum peptide length seven amino acids, peptide and protein FDR 1%, and a minimum of 2 unique peptides for identification.

Data were normalized using vsn v3.70.0[101], and statistical significance was determined using a two-tailed t-test in limma v3.58.1[102].

## Statistical analysis

All values were expressed as mean ± SD. Statistical analysis was performed by the Two-tailed Unpaired Student's t-test, One-way ANOVA and 2way ANOVA. A probability value of $P < 0.05$ was considered statistically significant and non-significant data comparison are represented as ns. Statistical significances were derived from minimum of three independent biological replicates ($n = 3$). The experiments were not randomized. Blinding was implemented for alkaline phosphatase quantification.

## Reporting summary

Further information on research design is available in the Nature Portfolio Reporting Summary linked to this article.

# Data availability

All next-generation sequencing data can be publicly accessed in ArrayExpress or GEO webserver. The accession number for RNA-seq, ChIP-seq, CUT&RUN and Mouse DNA methylation beadChIP Array are E-MTAB-14221, PRJEB78609, E-MTAB-14338 and GSE273767 respectively. By now, all next-generation sequencing data has been uploaded in https://doi.org/10.6084/m9.figshare.23092151.v1. The mass spectrometry proteomics data have been deposited to the ProteomeXchange Consortium via the PRIDE partner repository with the dataset identifier PXD042495. All other data are provided in the Source Data file. Source data are provided with this paper.

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

## Acknowledgements

We thank Filipe Pereira (WCMM and Lund University, Sweden) for providing *Dnmt1* and double *Dnmt3a/b* KO mouse ESCs. We would like to thank Aguilo Lab members for useful discussion, the Biochemical Imaging Center Umeå, and Mikael Lindberg (Protein expertise platform) for assisting with the cloning. D.-F.L. was partially supported by NIH R01 CA246130 and R01 HL142704. D.-F.L. is a CPRIT Scholar in Cancer Research. A.Mattevi's laboratory was supported by FISR2019_00374 MeDyCa from MUR. C.C's laboratory is supported by Knut and Alice Wallenberg Foundation, Swedish Research Council (2021-03075), and Cancerfonden (21 1572Pj). The Aguilo's lab research is supported by grants from the Knut and Alice Wallenberg Foundation, Umeå University, Västerbotten County Council, Swedish Research Council (2017-01636; 2022-01322; 2021-03075), Kempe Foundation (JCK-2150 and JCSMK22-0109), and Cancerfonden (190337 Pj; 22 2455 Pj).

## Author contributions

S.M. and F.A. conceived and designed the study. S.M. and J.G. performed seahorse. J.C. and A.Mattevi performed enzymatic assays of LSD1. S.S. and S.Sidoli performed quantitative analysis of histone modifications using LC-MS/MS. A.N. and C.C. performed CUT&RUN. H.W., W.L., and C.L. performed LC-MS/MS on DNA. C.G.-P., D.A.-E., and M.E. performed methylation studies. T.T.T.P. and D.-F.L. performed ubiquitination assays. A.M. performed the proteomics analysis. A-C.R. performed bioinformatics analysis. S.M., K.K., C.M.-G., D.P.B., E.D., P.A.S., L.L. and F.A. performed experiments. S.M. and F.A. wrote the manuscript. All authors reviewed and edited the manuscript. F.A. supervised the study.

## Funding

## Competing interests

The authors declare no competing interests.

## Additional information

[1]Department of Molecular Biology, Umeå University, Umeå, Sweden. [2]Wallenberg Centre for Molecular Medicine, Umeå University, Umeå, Sweden. [3]Cancer Epigenetics Group, Josep Carreras Leukaemia Research Institute, Barcelona, Spain. [4]Life Sciences Department, Barcelona Supercomputing Center (BSC), Barcelona, Spain. [5]Department of Biology and Biotechnology, University of Pavia, Pavia, Italy. [6]Wallenberg Centre for Molecular Medicine, Linköping University, Linköping, Sweden. [7]Department of Biomedical and Clinical Sciences, Division of Molecular Medicine and Virology, Faculty of Medicine and Health Sciences, Linköping University, Linköping, Sweden. [8]Department of Integrative Biology and Pharmacology, McGovern Medical School, The University of Texas Health Science Center at Houston, Houston, TX, USA. [9]Department of Biochemistry, Albert Einstein College of Medicine, Bronx, NY, USA. [10]State Key Laboratory of Environmental Chemistry and Ecotoxicology, Research Center for Eco-Environmental Sciences, Chinese Academy of Sciences, Beijing, China. [11]Department of Physics, Integrated Science Lab, Umeå University, Umeå, Sweden. [12]Department of Medical and Translational Biology, Umeå University, Umeå, Sweden. [13]Department of Chemistry, Umeå University, Umeå, Sweden. [14]The Laboratory for Molecular Infection Medicine Sweden (MIMS), Umeå, Sweden. [15]The University of Texas MD Anderson Cancer Center UTHealth Houston Graduate School of Biomedical Sciences, Houston, TX, USA. [16]Center for Stem Cell and Regenerative Medicine, The Brown Foundation Institute of Molecular Medicine for the Prevention of Human Diseases, The University of Texas Health Science Center at Houston, Houston, TX, USA. [17]Center for Precision Health, McWilliams School of Biomedical Informatics, The University of Texas Health Science Center at Houston, Houston, TX, USA. [18]Centro de Investigacion Biomedica en Red Cancer (CIBERONC), Madrid, Spain. [19]Institucio Catalana de Recerca i Estudis Avançats (ICREA), Barcelona, Spain. [20]Physiological Sciences Department, School of Medicine and Health Sciences, University of Barcelona (UB), Barcelona, Spain. [21]Department of Biochemistry, Molecular Biology and Genetics, University of Extremadura, Badajoz, Spain. ✉e-mail: francesca.aguilo@umu.se

