## [Peer Review File · Nature Communications]

The scaffolding function of LSD1 controls DNA methylation in mouse ESCsReviewers' Comments:

Reviewer #1:

Remarks to the Author:

In the manuscript entitled "The catalytic-independent function of LSD1 modulates the epigenetic landscape of mouse embryonic stem cells" the authors show the importance of LSD1 in pluripotency and differentiation by multiple mechanisms. While LSD1 activity is not important for ESC self-renewal it is important for cell differentiation. Mechanistically, they provide evidence that LSD1 loss impairs DNA methylation by controlling DNMT1 and UHRF1 levels. The manuscript is well written, and the authors use multiple proteomics and genome wide assays to address a novel question about the scaffold function of LSD1 in regulating DNA methylation and cell differentiation. Indeed, the authors demonstrated a great research rigor by including all the right controls and multiple orthogonal methods to strongly demonstrate most the conclusions of the manuscript. For instance, it is very nice to see that they even validated downregulated pathways such metabolism in LSD1 KO cells. However, there are some observations that need further clarification. I recommend publication of this manuscript in Nature Comm after the authors explain and test the following points:

1. ESCs have been cultured in serum + LIF conditions. I wonder if the addition of the 2i would prevent the LSD1 KO ESCs to partially differentiate (Fig. 1G).
2. Figure 3 is a bit undeveloped and Figure 3a and 3c should be moved as supplemental panels. It was proposed that LSD1 can also demethylate H3K9 but structural studies indicated that LSD1 can only demethylate H3K4. It is not clear how enhancers were defined in panel G and publicly available H3K27ac and H3K4me3 should be used to map typical enhancers and super enhancers. Given the very broad H3K4me1 ChIP-seq signal, peak calling can be quite inaccurate. Consider other methods to detect H3K4me1 ChIP-seq signal.
3. For the H3K4me1 and LSD1 ChIP-seq analysis, the authors should analyze the H3K4me1 signal at LSD1 sites. How the authors explained that the majority of deregulated genes in LSD1 KO ESC are not LSD1 targets?
4. Along the lines of my previous point, throughout the manuscript it is not clear whether the effects reported are by the CoREST complex or LSD1 alone. The authors should address this by depleting the main RCOR subunit associated with CoREST in ESC and perform few key experiments including ChIP-seq.
5. LSD1 can be recruited to both active and repressed genes. A deeper analysis of LSD1 and CoREST is necessary to better determine the contribution of LSD1, or CoREST, in ESC differentiation.
6. The authors concluded that loss of LSD1 leads to global hypomethylation that is mostly independent of its enzymatic activity. However, LSD1 enzymatic activity is essential for the proper methylation at specific loci. Can the authors comment about this? Can the authors integrate the H3K4me1 data with their methylation profiles?
7. Experiments with the GSK-LSD1 are important but a 24h treatment might not be sufficient to detect changes in the epigenome. Moreover, I could not find the concentration GSK-LSD1 that they used. Experiments should be performed in a time course to up to 5 or 7 days since epigenetic marks sometimes require cell divisions to be diluted.
8. Since the authors propose to publish the first DNMT1 genome-wide binding, ChIP-seq experiments should be performed in the DNMT1 KO cells as a control.
9. The Re-ChIP data is rather weak. Perhaps they could remove this experiment.
10. Is the interaction of LSD1 with DNMT1, UHRF1 and USP7 DNA/RNA dependent?
11. Does the CoREST complex interact with DNMT1, UHRF1 and USP7 or the interaction is only with LSD1?
12. Are DNMT1 and UHRF1 polyubiquitinated in LSD1 KO cells?
13. Data from figure 6L is not convincing and should be extended.

Minor points:

1. "LSD1MUT ESCs exhibited comparable DNA methylation levels to WT and LSD1WT ESCs (Figure 4E), indicating that, in contrast to previous studies [6, 20], the catalytic activity of LSD1 does not play

a role in DNA methylation". Data in Figure 4e indeed suggest that the catalytic activity of LSD1 partially contribute to DNA methylation and conclusion should be toned-down.

2. "Altogether, these data suggest that LSD1 regulates DNA methylation genome-wide, especially in repetitive elements and enhancers". Loss of methylation in gene bodies was also quite strong.

3. The analyses of the imprinted genes do not add much to the paper. I would remove the data.

4. In figure 1 the authors used OCT4 as a control of RA induced differentiation, however they should include an ectoderm marker such as PAX6, SOX1, etc. For the EB differentiation, markers from at least one differentiation marker should be included.

5. In figure 1F the authors check for apoptosis and claimed an increase in early apoptosis. To make this distinction between early and late apoptosis they should include a fragmented DNA marker such as PI.

Reviewer #2:

Remarks to the Author:

In this manuscript Malla et al attempted to investigate the roles of LSD in mouse ES cell proliferation, differentiation, transcription, histone and DNA methylation. Although some of the data are interesting, especially the finding LSD1 appeared to promote UHRF1 and DNMT1 protein stability and DNA methylation in demethylase activity-independent manner, some other data are not novel and confusing. The specific questions are as follow:

1. Authors claimed that LSD1 is dispensable for ESC self-renewal. This is inconsistent with previous publications and also not supported by their own data. It was shown that LSD1 knockout led to very severe defect in cell proliferation (Fig. 1E) and partial differentiation (Fig. 1F-1H). To me, this indicates LSD1 is required for ESC self-renewal.

2. In Fig. 3D authors analyzed LSD genomic distribution by ChIP-seq. It is not clear whether the identified peaks are specific to LSD1. Are these peaks specific to WT cells and absent in KO cells?

3. In Fig. 3H, it appears that there is little overlap between H3K4me1 peaks in WT and LSD1 KO cells. While LSD1 KO resulted in an increased H3K4me1 level and thus increased number of H3K4me1 peaks, why the majority of peaks observed in WT cells disappeared?

4. In Fig. 5J authors performed ChIP-seq for DNMT1. What is the relationship between DNMT1 binding regions and DNA hypomethylation in Lsd1 KO ES cells?

5. In Fig. 6 (E-G) authors identified differentially expressed proteins in Lsd1 KO2, LSD1 WT, and LSD1 MUT ESCs in comparison to WT ESCs. As LSD1 knockout resulted in changed expression of several thousands of gene, I do not understand what the authors intend to conclude here.

6. Authors proposed that LSD1 may function as a scaffold to target USP7 to DNMT1 and UHRF1, thus protecting them from ubiquitin-dependent degradation. Yet, as shown in 6L, knockdown of USP7 downregulated UHRF1 and DNMT1 proteins to a much less extent than knockout of LSD1 (Fig. 5A-5B). Thus, the effect of LSD1 on UHRF1 and DNMT1 protein stability could not be easily explained by their working model. Furthermore, there is no any evidence that LSD1 actually bridges the interaction between USP7 and UHRF1/DNMT1. In this regard, it is well known that USP7 interacts directly and potently with both DNMT1 and USP7.

7. As regard to the working model, I would suggest to test whether in ESCs LSD1 may protect UHRF1 and DNMT1 protein stability by its association with HDAC1/2. It has also been reported that DNMT1 protein stability is regulated acetylation in which acetylation induces DNMT1 degradation. This, it is possible in ESCs acetylation may be a major pathway that regulates UHRF1 and DNMT1 protein stability. LSD1 may function as a scaffold by bringing HDAC1/2 to deacetylate and stabilize UHRF1/DNMT1. This may explain why in ESCs LSD1 regulates UHRF1 and DNMT1 protein stability and function largely independent of its demethylase activity, whereas in somatic cells may rely on its demethylase activity to stabilize UHRF1 and DNMT1.

Reviewer #3:

Remarks to the Author:

The study by Malla and colleagues characterizes the role of the Lsd1/Kdm1a chromatin modifier and transcriptional regulator in mouse ESC biology. Lsd1 is a well-established demethylase that targets both histone (e.g., H3K4 and H3K9) and non-histone proteins (e.g., Oct4, DNMT1, Uhrf1, Hif1a). Lsd1 also acts as a scaffold to recruit SNAG-domain transcription factors (e.g., Gfi1/1b and Snail proteins) via its hydrophobic pocket and chromatin modifiers (e.g., CoREST and NuRD complexes) via its Tower and/or Swirl domains. Here, the authors use CrispR-mediated methods to generate Lsd1 knockout (KO) mouse ESC and show these cells maintain their self-renewal/pluripotency properties but have defects in growth/proliferation and differentiation. These findings are consistent with previous studies using germline Lsd1 KO or conditional KO mouse ESC. The authors next use Lsd1 rescue constructs that contain the wild-type (Lsd1WT) or a double mutant sequence (A540E/K662A, called Lsd1MUT) that was previously shown to significantly reduce Lsd1 demethylase activity using a histone nucleosome assay. Using these constructs, the authors determined the extent to which Lsd1WT and Lsd1MUT can rescue the ESC phenotypes and conclude that a Lsd1 non-catalytic function is required for ESC proliferation, whereas the Lsd1 catalytic function is required for differentiation. The authors also show that Lsd1 loss affects DNA methylation levels globally and at specific loci, coincident with decreased levels of DNMT1, which has been previously described. DNA methylation is partially rescued by the Lsd1MUT construct, suggesting a possible non-catalytic role of Lsd1 in regulating DNA methylation levels. Finally, the authors show that Lsd1 can affect the stability of DNMT1 and Uhrf1 via proteolysis by binding to the USP7 deubiquitinating enzyme, consistent with prior studies.

Overall, this study is a thorough description of the Lsd1 KO cell lines, using genomic, biochemical and bioinformatic techniques to support a model in which a non-catalytic function of Lsd1 is important for normal DNA methylation patterns in mouse ESC via stabilization of DNMT1 protein levels, which in turn impacts proliferation, but not differentiation. As most of the important findings in this study have already been published, with some differences in ESC cell lines or biochemical assays used, the impact of the study is somewhat diminished. In addition, the study is heavily reliant on the use of one rescue construct (Lsd1MUT) to make conclusions regarding the importance of Lsd1's non-catalytic role. This lack of rigor is particularly concerning for this field, as many prior studies relied solely on the use of one Lsd1 allele (K661A) to infer non-catalytic functions of Lsd1, but this allele was later shown to retain catalytic activity that depended on the biochemical assay being used- such studies now need to be re-evaluated. Therefore, it would be imprudent to continue to use this strategy and rely solely on the use of one Lsd1 mutant rescue construct to assess potential non-catalytic functions of Lsd1. In addition, the Lsd1 A540E/K662A variant disrupts two separate functions of Lsd1, with the A540E variant impacting substrate binding in the hydrophobic pocket, while the K662A variant impacts the FAD-dependent oxidase activity. Thus, the data presented cannot distinguish between a catalytic or substrate binding-dependent role for Lsd1 in ESC self-renewal or methylation. Based on these concerns and others (see below), the study is considered too preliminary to support the conclusions being made, with many alternative interpretations and mechanisms possible. Other suggestions and concerns are noted below.

1) There is already a significant body of literature describing the role of Lsd1 in both mouse and human ESC biology, so to state that role of Lsd1 in ESC self-renewal and differentiation is "poorly understood" in the Introduction is inaccurate and demonstrates a lack of effort to articulate the current knowledge base for Lsd1 in ESC biology, what the outstanding questions in the field are, and how this study will address them. Indeed, there are many reviews published on this topic due to the large body of work already performed. In general, the manuscript lacks clarity for understanding why experiments were performed and how the data rigorously support the conclusions.

2) DNMT1 and UHRF1 are already identified targets of Lsd1, with the target lysine residues defined (e.g., K1096 and K142 in DNMT1). The Lsd1MUT construct used in this study impairs histone demethylation, however it is not known if it can still demethylate DNMT1 or UHRF1 proteins, as this assay was not performed. Thus, the data shown cannot exclude the possibility that Lsd1 demethylase activity is important for DNMT1 stability, particular since other groups have already demonstrated this

mechanism (e.g., Wang et al., Nat. Genetics 2008). If the authors propose to refute the previously published mechanism, they need to perform the same assays with the Lsd1MUT variant to show it no longer de-methylates DNMT1 and/or the K1096 and K142 lysine's are demethylated via an alternative mechanism (e.g., another demethylase that binds Lsd1MUT).

3) Using the phrase "catalytically-dead" to describe the Lsd1MUT construct is imprecise given the history of describing similar Lsd1 alleles in past studies that were later shown to retain activity. The original manuscript describing the Lsd1 A539E/K661A allele (Kim et al., Mol. Cell 2020) used the term "catalytically-impaired", which is more appropriate and should be used instead.

4) The reliance on just one construct to infer Lsd1 non-catalytic roles in ESC biology is not rigorous. The K662A allele preferentially impacts FAD-dependent oxidase function, whereas the A540E allele introduces a bulky positive charge into Lsd1's hydrophobic pocket that interferes with substrate binding, at least for the Histone H3 tail (Kim et al., Mol. Cell 2020). Thus, the double mutant construct used in this study impacts two different functions of Lsd1. Therefore, it is not possible to discern if Lsd1 substrate binding or catalysis is important for its role in ESC growth and methylation. The study should include the analysis of each single mutant to allow comparison with the double mutant in order to determine the extent to which catalytic activity or substrate binding is important for Lsd1's role in ESC methylation and growth. Such an analysis was recently performed in zebrafish hematopoietic stem cells to differentiate between these two functions of Lsd1 in controlling transcriptional programs during differentiation in vivo (Casey et al., iScience 2023).

5) An important non-catalytic function of Lsd1 is binding to SNAG-domain transcription factors (e.g., Gfi1 & Snail proteins) via its substrate binding domain, which in turn controls Lsd1 recruitment to specific genomic loci and transcription during differentiation (e.g., Gfi1/1b binding to Lsd1 in AML cell lines). This alternative mechanism was not investigated in this study and may explain the inability of the Lsd1MUT variant to rescue the differentiation program in ESC.

6) Figure 1I and S1E. The Western blot for Lsd1 in KO cell lines should show the complete blot/film (at least as a supplemental figure), as Lsd1 has multiple splice isoforms, and it is possible the CrispR-generated mutations produce an Lsd1 protein via alternative splicing that runs at a lower molecular weight.

7) Figure 1M. There is significant transcriptional heterogeneity between the two Lsd1 KO cell lines. As the authors point out, this may be due to genetic compensation. However, this was not examined further, such as analysis of Lsd2 levels (a possible mechanism of compensation), which limits the conclusion in this section.

8) "There were no differences in the expression of multiple pluripotency and germ layer-specific markers between WT and Lsd1 KO ESCs". RT-PCR analysis shows an increase in endodermal markers (Fig S1J), which is inconsistent with the statement.

9) Figure 2B. The demethylase assay is specific for H3K4me2, so no inferences can be made regarding Lsd1 demethylase activity on other substrates used in this study, such as DNMT1.

10) Figure 2I. The Lsd1MUT protein levels should be included in this analysis.

11) Figure 2. Overall, the data does not rule out the possibility that Lsd1's catalytic and non-catalytic function plays a role in ESC differentiation, as the Lsd1MUT partially rescues the expression of many differentiation genes, just not as robustly as the Lsd1WT version.

12) Text: "Such an increase in the percentage of partially differentiated colonies is a consequence of the generic proliferative defects of loss of Lsd1 as ablation of Lsd1 did not affect the expression of the pluripotency factors OCT4 and NANOG"- why would a loss of proliferative capacity impact

differentiation? This should be more fully explained.

13) The findings and potential conclusions of this study are limited to mouse ESC, as previous studies have shown that the function of Lsd1 in mouse versus human ESC biology differs significantly. This should be stated in the limitations of the study.

Reviewers' comments:

Reviewer #1 (Remarks to the Author):

In the manuscript entitled "The catalytic-independent function of LSD1 modulates the epigenetic landscape of mouse embryonic stem cells" the authors show the importance of LSD1 in pluripotency and differentiation by multiple mechanisms. While LSD1 activity is not important for ESC self-renewal it is important for cell differentiation. Mechanistically, they provide evidence that LSD1 loss impairs DNA methylation by controlling DNMT1 and UHRF1 levels. The manuscript is well written, and the authors use multiple proteomics and genome wide assays to address a novel question about the scaffold function of LSD1 in regulating DNA methylation and cell differentiation. Indeed, the authors demonstrated a great research rigor by including all the right controls and multiple orthogonal methods to strongly demonstrate most the conclusions of the manuscript. For instance, it is very nice to see that they even validated downregulated pathways such metabolism in LSD1 KO cells. However, there are some observations that need further clarification. I recommend publication of this manuscript in Nature Comm after the authors explain and test the following points:

We thank the referee for acknowledging the interesting aspect and findings of our work.

1. ESCs have been cultured in serum + LIF conditions. I wonder if the addition of the 2i would prevent the LSD1 KO ESCs to partially differentiate (Fig. 1G).

Response: We appreciate the reviewer's comment. As suggested by the reviewer, we have grown mouse ESCs in 2iL i.e., serum-free medium containing inhibitors of Mek/Erk and Gsk3 β , which directs ESCs to a naïve ground state [1]. We have evaluated potential spontaneous differentiation by performing alkaline phosphatase assays. As expected, the number of undifferentiated colonies was increased in wild-type, *Lsd1* KO1 and KO2 when growing in 2iL compared to serum + LIF conditions (**revised Figures S2D and S2E**). Additionally, in contrast to what we observed in serum-cultured ESCs, loss of LSD1 did not visibly alter the count of partially differentiated colonies when compared to wild-type cells (**revised Figure S2D and S2E**). These findings underscore the distinct characteristics of mouse ESCs cultured in diverse conditions, emphasizing that those grown in serum/LIF and 2iL represent differing pluripotency stages [2].

Given the consistent expression of core pluripotency factors despite LSD1 loss and our ability to maintain *Lsd1* KO ESCs without compromising self-renewal over extended culture periods, we conclude that LSD1 may not be essential for maintaining pluripotency in either ground (2iL) or metastable (LIF + serum) states. This observation aligns with findings from other studies [3-5].

New text: *To examine the role of LSD1 in the naïve ground state, mouse ESCs were cultured in 2iL medium, consisting of kinase inhibitors targeting MAP kinase (MEK) and glycogen synthase kinase 3 β (GSK-3 β ; known as "2i"), along with leukemia inhibitory factor (LIF). Similar to the observed phenotype when growing*

the cells in medium containing serum and LIF, loss of LSD1 resulted in decreased cellular viability and increased apoptosis (**Figures S2B and S2C**). However, despite these effects, AP staining revealed consistent counts of both undifferentiated and partially differentiated colonies across the different cell lines (**Figures S2D and S2E**). This data aligns with the notion that while LSD1 may not be essential for maintaining pluripotency, its absence triggers significant proliferation deficiencies in both the metastable (LIF + serum) and in the naïve ground states (2iL).

2. Figure 3 is a bit undeveloped and Figure 3a and 3c should be moved as supplemental panels. It was proposed that LSD1 can also demethylate H3K9 but structural studies indicated that LSD1 can only demethylate H3K4. It is not clear how enhancers were defined in panel G and publicly available H3K27ac and H3K4me3 should be used to map typical enhancers and super enhancers. Given the very broad H3K4me1 ChIP-seq signal, peak calling can be quite inaccurate. Consider other methods to detect H3K4me1 ChIP-seq signal.

Response: The reviewer's suggestion has been taken into account, leading to the relocation of Figure 3A to the supplementary section as new **Figure S5A**. We agree with the reviewer that recent studies have shown that LSD1 exclusively demethylates H3K4. To enhance clarity, former Figure 3C, illustrating H3K9me1, H3K9me2, and H3K9me3 levels, has been removed. Additionally, the following sentence has been omitted from the manuscript: '*Strikingly, no changes in H3K9me1 were observed between WT and Lsd1 KO ESCs, whilst the latest displayed a decrease of H3K9me2 and H3K9me3 marks compared to WT (Figure 3C)*'.

In order to better define the H3K4me1 peaks, we have employed CUT&RUN which have been shown to have lower background noise levels compared to ChIP-seq [6]. We have expanded the analysis in order to map H3K4me1 not only in wild type and *Lsd1* KO ESCs but also in rescue WT (LSD1^{WT}) and mutant cell lines (LSD1^{MUT}) (revised **Figures 3I-3N, S5J and S5K**). As suggested by the reviewer, publicly available H3K27ac and H3K4me3 has been used to define enhancers and super-enhancers, and map H3K4me1 CUT&RUN signal (new **Figures 3J and K**).

New text: *Following the adapted Cleavage Under Targets and Release Using Nuclease (CUT&RUN) protocol CUT&RUN-LoV-U (low volume and urea), we mapped H3K4me1 in WT, KO, LSD1^{WT}, and LSD1^{MUT} ESCs. Principal component analysis unveiled a clear distinction between WT and Lsd1 KO ESCs, while LSD1^{WT} and LSD1^{MUT} clustered closely with WT ESCs (Figure S5J). We identified 5288 target genes in WT, 8540 in KO, 5391 in LSD1^{WT}, and 7230 in LSD1^{MUT}, consistent with the demethylating role of LSD1 on H3K4me1 in vivo (Figure 3I; Table S5). Likewise, LSD1 deletion and inactivation led to amplified H3K4me1 levels at enhancers and superenhancer regions (Figures 3J and 3K), and co-occupied LSD1 binding sites (Figure 3L). Genes shared among all lines, as well as those from the LSD1^{MUT} cell line, showed enrichment in generic functions such as regulation of transcription, cell cycle and RNA processing, amongst others (Figures 3M and S5K). However, H3K4me1 peaks exclusively identified in Lsd1 KO mouse ESCs (1699;*

Figure 3I) were associated with genes involved in neurogenesis-related categories (**Figure 3N**). These findings underscore the distinctions in the H3K4me1 landscape mediated by LSD1 deletion or enzymatic activity of LSD1.

3. For the H3K4me1 and LSD1 ChIP-seq analysis, the authors should analyze the H3K4me1 signal at LSD1 sites. How the authors explained that the majority of deregulated genes in LSD1 KO ESC are not LSD1 targets?

Response: Although LSD1 directly inhibits a portion of genes, a larger impact arises from its secondary role in modulating key transcription factors. In our analysis, we observed that 31% of the genes with increased expression in *Lsd1* KO, compared to wild-type ESCs, were also found in the LSD1 ChIP-seq. This observation aligns with previous findings. For instance, Egolf et al. reported that only 16% of the genes that were upregulated upon chemical inhibition of LSD1 were bound by LSD1 [7]. Similarly, Vinckier et al. found that 24% of the upregulated genes after LSD1 inhibition were targeted by LSD1 [8]. Importantly, the decrease in gene expression following LSD1 deletion further reinforces the idea of LSD1's indirect modulation of transcriptional outcomes. However, we acknowledge an error in the highlighted sentence by the reviewer. Our intention was to convey that a substantial majority of LSD1-bound genes—approximately 70% did not experience dysregulation.

New text: *Integration of RNA-seq in WT and Lsd1 KO mouse ESCs with ChIP-seq data showed that the majority of genes near to LSD1 peaks are not differentially expressed (~70%; Figure 3D).*

Moreover, in response to the reviewer's suggestion, we have integrated H3K4me1 CUT&RUN data with the LSD1 ChIP-seq analysis, presenting the results in the form of a heat map in the updated **Figure 3L**.

New text: *Likewise, LSD1 deletion and inactivation led to elevated H3K4me1 levels at enhancers and superenhancer regions (Figures 3J and 3K), and co-occupied LSD1 binding sites (Figure 3L).*

4. Along the lines of my previous point, throughout the manuscript it is not clear whether the effects reported are by the CoREST complex or LSD1 alone. The authors should address this by depleting the main RCOR subunit associated with CoREST in ESC and perform few key experiments including ChIP-seq.

Response: We appreciate the reviewer's feedback. To investigate the interaction between LSD1 and the CoREST complex in mediating DNA methylation, we conducted several experiments focusing on RCOR1 and RCOR2, the two more expressed isoforms in mouse ESCs [5].

a) We evaluated RCOR1, RCOR2, and HDAC1 protein levels in both whole cell extracts and chromatin extracts in WT, *Lsd1* KO, LSD1^{WT}, and LSD1^{MUT} ESCs, along with RT-qPCR. A decrease in protein levels was observed in *Lsd1* KO ESCs, with levels resembling those in WT in the LSD1^{WT} and LSD1^{MUT} cell lines (revised **Figures S9A** and **S9B**). No changes were observed at the mRNA level (**Figures S9C** and **S9D**). These experiments suggest that LSD1 acts as a scaffold, promoting the stability of the CoREST complex.

- b) Co-IP experiments using RCOR1 and RCOR2 antibodies revealed interactions of RCOR1/2 with LSD1 and HDAC1 but not with DNMT1 and UHRF1 (revised **Figures S9E** and **S9F**; see also comment 11 of the reviewer).
- c) We knocked down the expression of *Rcor1* and *Rcor2* using shRNAs. *Rcor1* knockdown ESCs showed a moderate increase in LSD1 levels compared to wild-type ESCs that could potentially account for the slight elevation in DNMT1 and UHRF1 protein levels upon RCOR1 silencing (new **Figure S9I**). No such changes were observed with RCOR2 loss (new **Figure S9J**). Depleting either RCOR1 or RCOR2 did not result in alterations in *Dnmt1* and *Uhrf1* mRNA levels (new **Figure S9K**).
- d) No changes in 5mC DNA levels were observed upon RCOR1 and RCOR2 knockdown (revised **Figure S9L**).
- e) Silencing RCOR2, but not RCOR1, led to a slight decrease in LSD1 chromatin recruitment (revised **Figure S9M**). This resulted in a decrease in LSD1 binding, indicating reduced binding at LSD1 target genes, which was assessed by ChIP RT-qPCR (revised **Figure S9N**).

Our experiments show that LSD1 has a close association with the CoREST complex, influencing its stability, while RCOR2 appears to modulate LSD1 chromatin recruitment. However, our findings led us to conclude that LSD1 promotes the stability of DNMT1 and UHRF1 independently of the RCOR1 and RCOR2 in mouse ESCs.

New text: *Since LSD1 is a component of the CoREST complex, which includes RCOR1/2/3 and HDAC1/2, we aimed to investigate whether the impact of LSD1 on DNMT1 and UHRF1 protein stability was mediated through CoREST. In Lsd1 KO mouse ESCs, there was a significant decrease in the protein levels of HDAC1, RCOR1, and RCOR2, the two most abundantly expressed RCOR paralogues in mouse ESCs, consequently impacting their recruitment to chromatin (Figures S9A and S9B). These levels were restored to WT levels in both LSD1^{WT} and LSD1^{MUT} cell lines, indicating that LSD1 serves as a scaffold to stabilize the CoREST complex independently of its enzymatic activity. Of note, no changes in the Rcor1 and Rcor2 mRNA levels were observed (Figures S9C and S9D).*

Subsequent immunoprecipitation assays with RCOR1 and RCOR2 antibodies confirmed their interaction with LSD1 and HDAC1, but no interaction was observed with DNMT1 or UHRF1 (Figures S9E and S9F). Moreover, neither RCOR1 nor RCOR2 exhibited interactions with USP7 (Figures S9G and S9H). We next employed an shRNA approach to silence the expression of RCOR1 and RCOR2. In RCOR1 knockdown mouse ESCs, western blotting analysis revealed a moderate increase in LSD1 levels compared to WT mouse ESCs, aligning with prior findings (Figure S9I). This might potentially account for the slight elevated DNMT1 and UHRF1 protein levels observed upon RCOR1 silencing. However, no such changes were observed with RCOR2 loss (Figure S9J). Notably, depleting either RCOR1 or RCOR2 did not result in alterations in Dnmt1 and Uhrf1 transcripts levels, nor did it impact 5mC levels (Figures S9K and S9L). Strikingly, in RCOR2 knockdown mouse ESCs, there was a decrease in LSD1 recruitment to chromatin, corresponding to a reduction in LSD1 binding at its target genes which was assessed by LSD1 ChIP

followed by RT-qPCR (**Figures S9M and S9N**). Although functionally linked, our data shows that LSD1 modulates DNMT1 and UHRF1 stability independently of RCOR1 and RCOR2.

5. LSD1 can be recruited to both active and repressed genes. A deeper analysis of LSD1 and CoREST is necessary to better determine the contribution of LSD1, or CoREST, in ESC differentiation.

Response: We appreciate the reviewer's suggestion to explore further the relationship between LSD1 and RCOR1/RCOR2 in ESC gene regulation during differentiation. However, a comprehensive investigation into this aspect has been extensively covered in a recently published manuscript [5]. This study highlights distinct roles for RCOR1 and RCOR2 in cellular differentiation, with RCOR2 significantly impacting gene expression during EB differentiation compared to RCOR1. Moreover, the simultaneous absence of RCOR1 and RCOR2 underscores their collective influence, revealing redundant control over cellular differentiation. Additionally, their findings reveal a functional connection between LSD1 and RCOR1/2 in regulating stem cell differentiation, demonstrating a notable overlap in genes affected in LSD1 null EBs and RCOR1/2 double KO EBs.

6. The authors concluded that loss of LSD1 leads to global hypomethylation that is mostly independent of its enzymatic activity. However, LSD1 enzymatic activity is essential for the proper methylation at specific loci. Can the authors comment about this? Can the authors integrate the H3K4me1 data with their methylation profiles?

Response: We recognize this insightful comment. Our principal component analysis showed a distinct methylation profile of WT and *Lsd1* KO ESCs, while WT, LSD1^{WT} and LSD1^{MUT} ESCs exhibited a similar methylation pattern (**Figure 4J**). The reintroduction of LSD1^{WT} and LSD1^{MUT} effectively rescued the hypomethylation observed in *Lsd1* KO ESCs. However, we observed persistent hypomethylation of 12.6% of probes in LSD1^{MUT} (**Figure 4O**). Additionally, mass spectrometry data demonstrated that LSD1^{MUT} partially, but not fully, alleviated the hypomethylation phenotype (**Figure 4E**). Therefore, our data suggests that LSD1 enzymatic activity is essential for the proper methylation at specific loci.

Despite the ability of mutant LSD1 to stabilize DNMT1 and UHRF1 protein levels, we must consider the potential crosstalk between chromatin marks. Existing evidence indicates that histone modifications can influence the recruitment of DNA methyltransferases and demethylases, impacting DNA methylation patterns. The literature on H3K4me1 and DNA methylation suggests both positive [9, 10] and negative [11-13] correlations, along with an enrichment of H3K4me1 at intermediate DNA methylation levels [14]. Recent work using CUT&Tag-BS, simultaneously profiling histone modification and DNA methylation, reveals that while H3K4me1 typically associates with unmethylated DNA, it also occurs at methylated DNA at specific genomic sites in a subset of cells, as seen in intermediately methylated H3K4me1 CpGs [15]. This aligns with previously reported cell-to-cell DNA methylation heterogeneity at enhancers in mouse ESCs [16], contributing to the complexity of the interplay between chromatin marks.

Following the reviewer's recommendation, we integrated H3K4me1 CUT&RUN data with the methylation profile datasets. Our observations indicate that H3K4me1 peaks are hypomethylated with no major differences in such hypomethylated regions between the cell lines (revised **Figures S6M** and **S6N**), emphasizing the significance of the histone landscape in influencing DNA methylation.

***New text:** Integration of CUT&RUN datasets with their methylation profiles indicated that H3K4me1 peaks are hypomethylated globally and at enhancer regions, with no major differences between the different cell lines (**Figures S6M** and **S6N**), underscoring the interplay among chromatin marks.*

7. Experiments with the GSK-LSD1 are important but a 24h treatment might not be sufficient to detect changes in the epigenome. Moreover, I could not find the concentration GSK-LSD1 that they used. Experiments should be performed in a time course to up to 5 or 7 days since epigenetic marks sometimes require cell divisions to be diluted.

Response: We apologize for the lack of clarity. As stated in the methods section, the concentration of GSK_LSD1 used in our experiments is 10 μ M. As suggested by the reviewer, we examined the 5mC levels on the genomic DNA of both WT and GSK_LSD1-treated ESCs after one week of treatment (new **Figure 5K**). However, we did not detect any significant changes in DNA methylation upon longer exposure to GSK_LSD1 treatment. These results align with our previous finding regarding the non-catalytic function of LSD1 in DNA methylation regulation.

***New text:** Longer exposure (1 week) displayed the same result (**Figure 5K**), confirming that the catalytic activity of LSD1 is not required for DNMT1 and UHRF1 expression, and DNA methylation maintenance.*

8. Since the authors propose to publish the first DNMT1 genome-wide binding, ChIP-seq experiments should be performed in the DNMT1 KO cells as a control.

Response: According to the reviewer's suggestion, we have performed DNMT1 ChIP-seq in *Dnmt1* KO ESCs. We have included the associated genomic tracks in the revised **Figure 5R, S7Q and S7R**.

9. The Re-ChIP data is rather weak. Perhaps they could remove this experiment.

Response: As suggested by the reviewer, we have removed the Re-ChIP data.

10. Is the interaction of LSD1 with DNMT1, UHRF1 and USP7 DNA/RNA dependent?

Response: In order to assess whether the interaction of LSD1 with DNMT1, UHRF1, and USP7 relies on DNA or RNA, we have treated the protein extracts of mouse ESCs with DNase I and RNase A individually. Subsequently, we performed immunoprecipitation of LSD1 followed by immunoblotting of LSD1, DNMT1, UHRF1, and USP7. The results, as illustrated in the newly generated **Figures 6C, 6D and S8I-SJ**, reveal that the interaction of LSD1 with the aforementioned proteins is DNA and RNA independent.

New text: This interaction remained unaffected by DNase and RNase treatment, indicating its independence from DNA and RNA, respectively (**Figures 6C and 6D**).

New text2: Such interaction was independent of DNA and RNA, as evidenced in WT mouse ESCs treated with DNase and RNase (**Figures S8I and S8J**).

11. Does the CoREST complex interact with DNMT1, UHRF1 and USP7 or the interaction is only with LSD1?

Response: We appreciate the reviewer's input. To address the reviewer's comment, we conducted immunoprecipitation of endogenous RCOR1 and RCOR2, followed by western blotting of DNMT1, UHRF1, and USP7. LSD1 and HDAC1 were utilized as positive controls. The revised **Figures S9E, S9F, S9G and S9H** demonstrate that RCOR1 and RCOR2 do not exhibit interaction with DNMT1, UHRF1, and USP7. This observation reaffirms our earlier findings indicating that LSD1 promotes the stability of DNMT1 and UHRF1 independently of the RCOR1 and RCOR2 in mouse ESCs.

New text (1): Subsequent immunoprecipitation assays with RCOR1 and RCOR2 antibodies confirmed their interaction with LSD1 and HDAC1, but no interaction was observed with DNMT1 or UHRF1 (**Figures S9E and S9F**). Moreover, neither RCOR1 nor RCOR2 exhibited interactions with USP7 (**Figures S9G and S9H**).

12. Are DNMT1 and UHRF1 polyubiquitinated in LSD1 KO cells?

Response: As suggested by the reviewer, we conducted an assessment of the polyubiquitination status of DNMT1 and UHRF1 under several conditions:

- a) In WT, *Lsd1* KO, LSD1^{WT}, and LSD1^{MUT} ESCs; we observed an increase in ubiquitination levels of both DNMT1 and UHRF1 in *Lsd1* KO ESCs. Ubiquitination levels returned to control levels in LSD1^{WT} and LSD1^{MUT} cell lines (**Figures 7R and 7S**).
- b) In *Usp7* knockdown ESCs, we observed an increase in ubiquitination of DNMT1 and UHRF1 upon USP7 loss (**Figures 7T and 7U**).
- c) Treatment with the USP7 inhibitor P22077 resulted in increased ubiquitination of DNMT1 and UHRF1 (**Figures 7V and 7W**).

These findings suggest a role of LSD1 in preventing the ubiquitination of DNMT1 and UHRF1 independent of its demethylase activity. Additionally, they underscore the significant role of USP7 in deubiquitinating DNMT1 and UHRF1, as previously suggested [17, 18].

New text: We next examined whether the deletion of LSD1 enhances DNMT1 and UHRF1 ubiquitination. For this purpose, we performed endogenous DNMT1 and UHRF1 immunoprecipitation in MG132-treated WT, *Lsd1* KO, LSD1^{WT}, and LSD1^{MUT} mouse ESCs, followed by immunoblotting with anti-ubiquitin antibodies. Significantly, we detected a higher-molecular-weight smear signal indicative of polyubiquitination in *Lsd1* KO compared to WT mouse ESCs (**Figures 7R and 7S**). This signal was reduced in LSD1^{WT} and LSD1^{MUT} mouse ESCs, similar to WT levels (**Figures 7R and 7S**). Moreover, DNMT1 and

UHRF1 ubiquitination increased in Usp7 knockdown cells compared to control cells, and upon USP7 inhibitor treatment (Figures 7T-7W).

13. Data from figure 6L is not convincing and should be extended.

Response: We apologize for the lack of clarity. We have repeated this experiment multiple times and we provide a clearer representative western blot with the corresponding quantification (new **Figures 7A-7C**). As suggested by the reviewer, in order to extend this data, we have used the specific USP7 inhibitor P22077, which corroborates the same result (new **Figures S8D-S9H**). Additionally, treatment with cycloheximide of control and *Usp7* shRNA ESCs indicated that USP7 is necessary in controlling the stability of both DNMT1 and UHRF1 (new **Figures 7D-7F**).

New text: *Silencing of USP7 diminished DNMT1 and UHRF1 protein abundances, which were recovered after four hours of treatment with MG132 (Figure 7A-C), while Dnmt1 and Uhrf1 mRNA levels remained unaffected (Figure S8C). Furthermore, cycloheximide experiments indicated a shortened half-life of DNMT1 and UHRF1 upon Usp7 knockdown (Figures 7D-7F). Comparable outcomes were observed following treatment with the specific USP7 inhibitor, P22077 (Figures S8D-S8H).*

Minor points:

1. "LSD1MUT ESCs exhibited comparable DNA methylation levels to WT and LSD1WT ESCs (Figure 4E), indicating that, in contrast to previous studies [6, 20], the catalytic activity of LSD1 does not play a role in DNA methylation". Data in Figure 4e indeed suggest that the catalytic activity of LSD1 partially contribute to DNA methylation and conclusion should be toned-down.

Response: We regret the lack of clarity in our depiction of **Figure 4E**. As the reviewer rightly noted, LSD1 enzymatic activity does not fully recover 5mC levels to the same levels as WT ESCs. This suggests that the demethylase activity of LSD1 is crucial for establishing precise methylation at particular loci, a point we elaborate on in our manuscript when discussing the **Figures S6K** and **S6L**. Following the reviewer's suggestion, we've made revisions to the text accordingly.

New text: *Remarkably, LSD1^{MUT} mouse ESCs partially recovered the DNA methylation levels of Lsd1 KO mouse ESCs (Figure 4E), indicating that, in contrast to previous studies, the catalytic activity of LSD1 may not be necessary for influencing DNA methylation at certain specific genomic regions.*

2. "Altogether, these data suggest that LSD1 regulates DNA methylation genome-wide, especially in repetitive elements and enhancers". Loss of methylation in gene bodies was also quite strong.

Response: We appreciate the reviewer's comment and we have amended the text to reflect the reviewer's point.

New text: *"Altogether, these data suggest that LSD1 regulates DNA methylation genome-wide, especially in repetitive elements, enhancers, and gene bodies".*

3. The analyses of the imprinted genes do not add much to the paper. I would remove the data.

Response: We have removed the analysis of the imprinted genes.

4. In figure 1 the authors used OCT4 as a control of RA induced differentiation, however they should include an ectoderm marker such as PAX6, SOX1, etc. For the EB differentiation, markers from at least one differentiation marker should be included.

Response: We have conducted RT-qPCR to assess the expression of *Nestin* during retinoic acid differentiation and *Sox17* in embryoid bodies (new **Figures 1A** and **1B**).

5. In figure 1F the authors check for apoptosis and claimed an increase in early apoptosis. To make this distinction between early and late apoptosis they should include a fragmented DNA marker such as PI.

Response: We appreciate the reviewer's comment. We used the Muse® Annexin V & Dead Cell Kit for our apoptosis assay, employing two stains to discern live, dead, and apoptotic cells. During the early stages of apoptosis, phosphatidylserine molecules migrate to the outer cell membrane, detected by Annexin V-PE. 7-AAD, as it is excluded from live and healthy cells, specifically permeates late-stage apoptotic and dead cells. However, upon careful review, we acknowledged an error in our manuscript concerning the analysis of apoptotic cells, as we meant that both early and late apoptosis (7-AAD negative and positive) are increased upon *LSD1* loss. We have made the necessary amendments to the text accordingly.

Figure R1: Apoptosis profile of WT and *Lsd1* KO ESCs performed with Muse™ Annexin V & Dead Cell Assay. Quadrant 1 (UL) - dead cells (7-AAD +/ Annexin V -); quadrant 2 (UR) - late apoptotic/dead cells (7-AAD +/ Annexin V +); quadrant 3 (LL) - live cells (7-AAD -/ Annexin V -); and quadrant 4 (LR) - early apoptotic cells (7-AAD -/ Annexin V +).

Reviewer #2 (Remarks to the Author):

In this manuscript Malla et al attempted to investigate the roles of LSD in mouse ES cell proliferation, differentiation, transcription, histone and DNA methylation. Although some of the data are interesting, especially the finding LSD1 appeared to promote UHRF1 and DNMT1 protein stability and DNA methylation in demethylase activity-independent manner, some other data are not novel and confusing.

We thank the reviewer for recognizing the novelty of our study uncovering LSD1's role in promoting UHRF1 and DNMT1 protein stability and DNA methylation in a manner independent of its demethylase activity. We've re-evaluated specific data sections to improve clarity and ensure that our findings contribute meaningfully to the existing body of knowledge. Moreover, we have refined some aspects for a clearer and more impactful presentation of our research.

The specific questions are as follow:

1. Authors claimed that LSD1 is dispensable for ESC self-renewal. This is inconsistent with previous publications and also not supported by their own data. It was shown that LSD1 knockout led to very severe defect in cell proliferation (Fig. 1E) and partial differentiation (Fig. 1F-1H). To me, this indicates LSD1 is required for ESC self-renewal.

Response: We respectfully disagree with the reviewer's comment. Although the impact of LSD1 in ESC self-renewal and pluripotency remains debated as highlighted in our review [19], the majority of studies suggest that LSD1 is dispensable for ESC self-renewal. Hence, it has been reported that while LSD1 deficiency in mouse ESCs affects growth, these cells maintain their undifferentiated state and the expression of key pluripotency factors [20] [4] [5]. Additionally, it has been shown that LSD1-deficient mouse ESCs displayed a differentiation impairment produced by the aberrant expression of developmental markers [3]. Nevertheless, contrasting findings highlight LSD1's involvement in ESC maintenance. It promotes OCT4 protein stability in pluripotent stem cells and regulates the transcription of genes essential for pluripotency [21]. Moreover, LSD1 seems crucial in maintaining human ESCs by silencing developmental genes through specific histone markers at enhancers occupied by OCT4 and NANOG [22].

Our interpretation of our data stemmed from observing consistent expression of core pluripotency factors despite LSD1 loss and the sustained self-renewal of *Lsd1* KO ESCs over extended culture periods. However, we acknowledge that the severe defects observed in cell proliferation and partial differentiation, as presented in **Figures 1E-1H**, might suggest a crucial role for LSD1 in these processes, which might indeed impact ESC self-renewal. To explore this possibility, we cultured ESCs in the presence of 2i, directing them to a naïve ground state. Although these culture conditions prevented partial differentiation in *Lsd1* KO ESCs, deficiencies in proliferation and apoptosis persisted (revised **Figures S2B-S2E**). From these findings, we conclude that while LSD1 might not be indispensable for maintaining pluripotency in either ground (2iL) or metastable (LIF + serum) states, it plays a crucial role in ensuring proper proliferation of mouse ESCs. For further details, refer to the suggestions provided by reviewer 1 (comment 1).

New text: To examine the role of LSD1 in the naïve ground state, mouse ESCs were cultured in 2iL medium, consisting of kinase inhibitors targeting MAP kinase (MEK) and glycogen synthase kinase 3 β (GSK-3 β ; known as "2i"), along with leukemia inhibitory factor (LIF). Similar to the observed phenotype when growing the cells in medium containing serum and LIF, loss of LSD1 resulted in decreased cellular viability and increased apoptosis (**Figures S2B and S2C**). However, despite these effects, AP staining revealed consistent counts of both undifferentiated and partially differentiated colonies across the different cell lines (**Figures S2D and S2E**). This data aligns with the notion that while LSD1 may not be essential for maintaining pluripotency, its absence triggers significant proliferation deficiencies in both the metastable (LIF + serum) and in the naïve ground states (2iL).

2. In Fig. 3D authors analyzed LSD genomic distribution by ChIP-seq. It is not clear whether the identified peaks are specific to LSD1. Are these peaks specific to WT cells and absent in KO cells?

Response: In response to the reviewer's recommendation, we conducted LSD1 ChIP in *Lsd1* KO ESCs. Notably, we have detected only 140 called peaks, which we deem artifacts due to their lack of overlap with LSD1 peaks observed in WT ESCs. The genomic tracks associated with this ChIP have been included in the revised **Figures 3G, 3H, and S5G-S5I** for your reference.

New text: We identified a total of 9,740 LSD1 peaks throughout the genome in WT ESCs whereas only a negligible number of peaks (140) were found in *Lsd1* KO ESCs (**Figure 3B; Table S3**).

3. In Fig. 3H, it appears that there is little overlap between H3K4me1 peaks in WT and LSD1 KO cells. While LSD1 KO resulted in an increased H3K4me1 level and thus increased number of H3K4me1 peaks, why the majority of peaks observed in WT cells disappeared?

Response: We appreciate the reviewer for bringing this to our attention. Upon thorough reevaluation, we acknowledge that the initial analysis was flawed. The venn diagrams were generated based on the distance to the TSS which we found to be inaccurate, as a single change in one numerical value did not produce the expected overlap. Subsequently, we re-evaluated the analysis by considering the gene names. The revised findings, presented in the new **Figure 3E**, now reveal a more accurate representation with a 63% overlap between H3K4me1 peaks in WT and *Lsd1* KO ESCs, aligning more closely with the anticipated outcome.

New text: In WT mouse ESCs, we identified 4,262 genes marked with H3K4me1, while in *Lsd1* KO ESCs, the count was 9,965 genes, with a significant 63% of the genes in WT mouse ESCs overlapping with *Lsd1* KO mouse ESCs (**Figure 3E and Table S4**).

4. In Fig. 5J authors performed ChIP-seq for DNMT1. What is the relationship between DNMT1 binding regions and DNA hypomethylation in *Lsd1* KO ES cells?

Response: In response to the reviewer's suggestion, we have compared DNMT1 target signals in WT with 5mC datasets from the Mouse Methylation MM285 BeadChIP microarrays. As illustrated in the updated **Figure 5N**, there is a correlation between DNMT1 binding and hypomethylated regions in *Lsd1* KO ESCs.

New text: *Furthermore, DNMT1 peaks correlated with a decrease in 5mC levels in Lsd1 KO mouse ESCs (Figure 5N).*

5. In Fig. 6 (E-G) authors identified differentially expressed proteins in *Lsd1* KO2, LSD1 WT, and LSD1 MUT ESCs in comparison to WT ESCs. As LSD1 knockout resulted in changed expression of several thousands of genes, I do not understand what the authors intend to conclude here.

Response: We appreciate the reviewer's observation, and we sincerely apologize for the oversight. In response, we have integrated transcriptomics and proteomics datasets to address the error. Our objective is to illustrate that the impact of *Lsd1* KO extends beyond transcriptional regulation (e.g., via scaffolding of COREST and other transcription factors [23]) to the post-transcriptional level (e.g., by modulating protein stability through ubiquitination-deubiquitination mechanisms). The updated plots now offer visual evidence of the global changes in protein levels, highlighting that the effect of *Lsd1* KO is more pronounced than that of LSD1^{MUT} (new **Figures 6H-6J**).

New text: *We then integrated the transcriptomic and proteomic data to identify post-transcriptional changes. In this regard, a total of 5509 proteins were used for comprehensive analysis, of which 5253 had corresponding mRNAs in the transcriptome (Table S10). In Lsd1 KO compared to WT mouse ESCs, 414 proteins were downregulated, and 221 proteins were upregulated whilst their mRNA remained unchanged (Figure 6H). These findings indicate that the loss of LSD1 can exert both stabilizing and destabilizing effects on global protein expression as previously reported. LSD1^{WT} mouse ESCs were able to partially rescue the number of dysregulated proteins (Figure 6I). Furthermore, we found that only 251 proteins were downregulated and 98 proteins were upregulated in LSD1^{MUT} mouse ESCs (Figure 6J), implying that altering the enzymatic activity of LSD1 has a diminished impact on proteome regulation compared to the effects observed in Lsd1 KO ESCs*

6. Authors proposed that LSD1 may function as a scaffold to target USP7 to DNMT1 and UHRF1, thus protecting them from ubiquitin-dependent degradation. Yet, as shown in 6L, knockdown of USP7 downregulated UHRF1 and DNMT1 proteins to a much less extent than knockout of LSD1 (Fig. 5A-5B). Thus, the effect of LSD1 on UHRF1 and DNMT1 protein stability could not be easily explained by their working model. Furthermore, there is no any evidence that LSD1 actually bridges the interaction between USP7 and UHRF1/DNMT1. In this regard, it is well known that USP7 interacts directly and potently with both DNMT1 and USP7.

Response: We apologize for any lack of clarity in our previous presentation of Figure 6L. In this revised manuscript, we have included a clearer representative western blot, with a higher efficiency of *Usp7*

knockdown, demonstrating a more pronounced downregulation in the protein levels of DNMT1 and UHRF1 upon *Usp7* silencing (**Figure 7A**). Additionally, we have validated this observation by treating cells with the specific USP7 inhibitor P22077 (**Figures S8D-S8H**). However, we acknowledge the reviewer's observation that the impact of *Usp7* depletion on UHRF1 and DNMT1 downregulation is less significant compared to *Lsd1* knockout cells. It's important to emphasize that our approach involves knockdown rather than knockout, which may contribute to differences in the observed effects.

Furthermore, while it has been documented that USP7 interacts directly with UHRF1 and DNMT1, we have been unable to co-immunoprecipitate them in *Lsd1* knockout ESCs (revised **Figure 7H**). To address whether this lack of immunoprecipitation was due to low protein levels of DNMT1 and UHRF1, and avoid misinterpretation of the data, we generated two new cell lines expressing doxycycline-inducible DNMT1 or UHRF1 in *Lsd1* KO ESCs (**Figures S8K and S8L**). As pointed out by the reviewer, we observed interactions between USP7 and DNMT1/UHRF1 in the absence of LSD1 (**Figures 7I and 7J**). This suggests that while LSD1 does not serve as a direct bridge for the interaction between USP7 and DNMT1/UHRF1, it plays an important role in mediating their stability (see comment 7 of the same reviewer).

In agreement with this observation, in the revised manuscript, we present evidence of an increase in polyubiquitination of DNMT1 and UHRF1 in *Lsd1* KO but not LSD1^{WT} and LSD1^{MUT} cell lines (refer to new **Figures 7R and 7S**). This data strongly suggests a role of LSD1, independent of its demethylase activity, in mediating DNMT1 and UHRF1 ubiquitination. Additionally, increased polyubiquitination of DNMT1 and UHRF1 is also observed upon *Usp7* knockdown and USP7 inhibitor treatment, indicating that USP7 deubiquitinates DNMT1 and UHRF1 in mouse ESCs (**Figures 7T-7W**).

Nevertheless, we agree with the reviewer that we cannot discount the possibility that LSD1 mediates the interaction between USP7 and an additional protein, such as a demethylase or deacetylase (as discussed in the original version of the manuscript) which may enhance USP7-mediated deubiquitination of DNMT1 and UHRF1.

New text: *Conversely, while DNMT1 and UHRF1 interacted with USP7 in WT, LSD1^{WT}, and LSD1^{MUT} cell lines, this interaction was mainly absent in Lsd1 KO ESCs (Figure 7H). To assess whether this lack of interaction was due to low protein levels of DNMT1 and UHRF1, we generated two new cell lines expressing doxycycline-inducible DNMT1 or UHRF1 in Lsd1 KO ESCs (Figure S8K and S8L). Interestingly, we observed interactions between USP7 and both DNMT1 and UHRF1 in the absence of LSD1 (Figure 7I and 7J). This suggests that although LSD1 does not directly bridge the interaction between USP7 and DNMT1/UHRF1, it plays a crucial role in mediating their stability, possibly by aiding their interaction with USP7, resulting in enhanced deubiquitinase activity.*

New text 2: *We next examined whether the deletion of LSD1 enhances DNMT1 and UHRF1 ubiquitination. For this purpose, we performed endogenous DNMT1 and UHRF1 immunoprecipitation in MG132-treated WT, Lsd1 KO, LSD1^{WT}, and LSD1^{MUT} mouse ESCs, followed by immunoblotting with anti-ubiquitin antibodies. Significantly, we detected a higher-molecular-weight smear signal indicative of*

polyubiquitination in *Lsd1* KO compared to WT mouse ESCs (**Figures 7R and 7S**). This signal was reduced in *LSD1^{WT}* and *LSD1^{MUT}* mouse ESCs, similar to WT levels (**Figure 7R and 7S**). Moreover, DNMT1 and UHRF1 ubiquitination increased in *Usp7* knockdown cells compared to control cells, and upon USP7 inhibitor treatment (**Figures 7T-7W**).

7. As regard to the working model, I would suggest to test whether in ESCs LSD1 may protect UHRF1 and DNMT1 protein stability by its association with HDAC1/2. It has also been reported that DNMT1 protein stability is regulated acetylation in which acetylation induces DNMT1 degradation. This, it is possible in ESCs acetylation may be a major pathway that regulates UHRF1 and DNMT1 protein stability. LSD1 may function as a scaffold by bringing HDAC1/2 to deacetylate and stabilize UHRF1/DNMT1. This may explain why in ESCs LSD1 regulates UHRF1 and DNMT1 protein stability and function largely independent of its demethylase activity, whereas in somatic cells may rely on its demethylase activity to stabilize UHRF1 and DNMT1.

Response: We appreciate the reviewer's input. We would like to emphasize that recent cryoEM studies have shed light on LSD1's specificity for H3K4 demethylation, as the catalytic site of LSD1 cannot accommodate more than 3 residues N-terminal to the target lysine for H3K4 [24]. Additionally, our *in vitro* enzymatic assays have demonstrated LSD1's incapacity to directly demethylate UHRF1, while still retaining its ability to demethylate the H3K4me2 peptide (see comment 2 from reviewer 3). Therefore, it is implausible that LSD1 relies on its demethylase activity to stabilize UHRF1 and DNMT1.

As mentioned by the reviewer, it has been reported that DNMT1 expression is regulated in an acetylation-dependent manner [25]. In this context, we conducted several experiments to investigate whether HDAC1/2-mediated deacetylation stabilizes UHRF1/DNMT1.

- a) We first assessed the interaction between HDAC1 and UHRF1/DNMT1. HDAC1 interacts with LSD1, with no observed interaction with either UHRF1 or DNMT1 (**Figure 7K**).
- b) We treated WT ESCs with the HDAC inhibitors TSA and SAHA. This treatment led to a decrease in DNMT1 and UHRF1 protein levels, independent of mRNA levels (**Figures 7L, 7M, S9S and S9T**). Additionally, we found that co-treatment with the proteasome inhibitor MG132 rescued DNMT1 and UHRF1 protein levels following TSA and SAHA treatment (**Figures 9N-9O and S9V-S9X**).
- c) We also explored the possibility that the acetylation/deacetylation status affected binding of DNMT1/UHRF1 with LSD1 and USP7. In this context, we found that treatment with TSA did not alter the binding of LSD1 with DNMT1 and UHRF1 (**Figure S9U**). Additionally, DNMT1 and UHRF1 were still able to interact with USP7 in the presence of TSA and SAHA (**Figure 7Q**).
- d) Simultaneous treatment with USP7 and HDAC inhibitors demonstrated a synergistic effect, resulting in undetectable protein levels of both DNMT1 and UHRF1 (**Figure S9Y**).

In summary, our additional experiments provide valuable insights suggesting that HDAC1-mediated deacetylation of DNMT1 and UHRF1 promotes their stability. We propose that LSD1 may mediate such deacetylation by interacting with HDAC1. Furthermore, our data indicate that acetylation and

deubiquitination, despite their contradictory effects on UHRF1 and DNMT1 protein stabilities, are not mutually exclusive in our model and can occur simultaneously. We appreciate the valuable input from the reviewer, which has contributed to the refinement and deeper understanding of our findings. **Figure 8**, depicting the model, has been amended accordingly to reflect these insights.

New text: *Further immunoprecipitation assays with HDAC1 antibodies revealed an interaction between HDAC1 and LSD1, but not with DNMT1 and UHRF1 (Figure 7K). Given that inhibition of HDAC has been shown to destabilize DNMT1 and UHRF1 proteins [49, 50], we treated WT ESCs with trichostatin A (TSA) and suberoylanilide hydroxamic acid (SAHA). Consequently, treatment with HDAC inhibitors led to increased acetylation levels of both DNMT1 and UHRF1, correlating with a decrease in protein abundances (Figures S9O-R and 7L-7M). No changes were observed at the transcriptional level, as assessed by RT-qPCR of Dnmt1 and Uhrf1 (Figures S9S and S9T). Additionally, TSA did not disrupt the interaction between LSD1 and DNMT1 or UHRF1 (Figure S9U) [51, 52]. Treatment with the proteasome inhibitor MG132 rescued DNMT1 and UHRF1 levels upon HDAC inhibition, indicating destabilization through ubiquitination-mediated proteasomal degradation as previously demonstrated (Figures 7N-P and S9V-X) [44, 45, 49]. Although it has been reported that acetylation of DNMT1 negatively impacts its interaction with USP7 [53], we detected USP7 binding to both DNMT1 and UHRF1 in the presence of TSA and SAHA (Figure 7Q). Thus, although both HDAC1 and USP7 contribute to DNMT1 and UHRF1 stability via deacetylation and deubiquitylation, deacetylation does not appear to be a requirement for deubiquitylation to occur in mouse ESCs. Moreover, the combination of HDAC inhibitors with USP7 inhibitors had a synergistic effect on DNMT1 and UHRF1 protein stability, reducing their abundances to nearly undetectable levels (Figure S9Y).*

Reviewer #3 (Remarks to the Author):

The study by Malla and colleagues characterizes the role of the Lsd1/Kdm1a chromatin modifier and transcriptional regulator in mouse ESC biology. Lsd1 is a well-established demethylase that targets both histone (e.g., H3K4 and H3K9) and non-histone proteins (e.g., Oct4, DMnt1, Uhrf1, Hif1a). Lsd1 also acts as a scaffold to recruit SNAG-domain transcription factors (e.g., Gfi1/1b and Snail proteins) via its hydrophobic pocket and chromatin modifiers (e.g., CoREST and NuRD complexes) via its Tower and/or Swirl domains. Here, the authors use CrispR-mediated methods to generate Lsd1 knockout (KO) mouse ESC and show these cells maintain their self-renewal/pluripotency properties but have defects in growth/proliferation and differentiation. These findings are consistent with previous studies using germline Lsd1 KO or conditional KO mouse ESC. The authors next use Lsd1 rescue constructs that contain the wild-type (Lsd1WT) or a double mutant sequence (A540E/K662A, called Lsd1MUT) that was previously shown to significantly reduce Lsd1 demethylase activity using a histone nucleosome assay. Using these constructs, the authors determined the extent to which Lsd1WT and Lsd1MUT can rescue the ESC phenotypes and conclude that a Lsd1 non-catalytic function is required for ESC proliferation, whereas the Lsd1 catalytic function is required for differentiation. The authors also show that Lsd1 loss affects DNA methylation levels globally and at specific loci, coincident with decreased levels of DNMT1, which has been previously described. DNA methylation is partially rescued by the Lsd1MUT construct, suggesting a possible non-catalytic role of Lsd1 in regulating DNA methylation levels. Finally, the authors show that Lsd1 can affect the stability of DNMT1 and Uhrf1 via proteolysis by binding to the USP7 deubiquitinating enzyme, consistent with prior studies.

Overall, this study is a thorough description of the Lsd1 KO cell lines, using genomic, biochemical and bioinformatic techniques to support a model in which a non-catalytic function of Lsd1 is important for normal DNA methylation patterns in mouse ESC via stabilization of DNMT1 protein levels, which in turn impacts proliferation, but not differentiation. As most of the important findings in this study have already been published, with some differences in ESC cell lines or biochemical assays used, the impact of the study is somewhat diminished. In addition, the study is heavily reliant on the use of one rescue construct (Lsd1MUT) to make conclusions regarding the importance of Lsd1's non-catalytic role. This lack of rigor is particularly concerning for this field, as many prior studies relied solely on the use of one Lsd1 allele (K661A) to infer non-catalytic functions of Lsd1, but this allele was later shown to retain catalytic activity that depended on the biochemical assay being used- such studies now need to be re-evaluated. Therefore, it would be imprudent to continue to use this strategy and rely solely on the use of one Lsd1 mutant rescue construct to assess potential non-catalytic functions of Lsd1. In addition, the Lsd1 A540E/K662A variant disrupts two separate functions of Lsd1, with the A540E variant impacting substrate binding in the hydrophobic pocket, while the K662A variant impacts the FAD-dependent oxidase activity. Thus, the data presented cannot distinguish between a catalytic or substrate binding-dependent role for Lsd1 in ESC self-renewal or methylation. Based on these concerns and others (see below), the study is considered too preliminary to

support the conclusions being made, with many alternative interpretations and mechanisms possible. Other suggestions and concerns are noted below.

We respectfully disagree with the assertion that the findings we present have been previously published. To the best of our knowledge, no published evidence has addressed the role of the non-catalytic function of LSD1 in maintaining normal DNA methylation patterns in mouse embryonic stem cells (ESCs) by stabilizing the protein levels of DNMT1 and UHRF1. We demonstrate that both LSD1 and a non-catalytic mutant form of LSD1 (LSD1^{MUT}) regulate the stability of UHRF1 and DNMT1 proteins, inhibiting the ubiquitylation of DNMT1 and UHRF1. These findings reveal a novel mechanism through which the scaffolding function of LSD1 exerts control over DNA methylation in ESCs.

We apologize for any confusion regarding the terminology used in our study. In our investigation, the term LSD1^{MUT} does not refer to K661A but rather to A540E/K662A. We took in consideration that K661A has been demonstrated to retain substantial H3K4 demethylase activity on nucleosome substrates [24]. Therefore, we opted to utilize the A540E/K662A double mutant. Since its enzymatic activity had not been previously assessed in mouse cells, we conducted additional experiments, including the purification of A540E/K662A LSD1 and subsequent enzymatic assays, confirming that this enzyme fails to demethylate H3K4me2. Our aim was not to differentiate between the various functions of LSD1, including substrate binding and FAD-dependent oxidase activity, but rather to demonstrate that LSD1 promotes the stability of DNMT1 and UHRF1 independently of its enzymatic activity. However, in response to the reviewer's suggestion, we have included experiments involving the single mutants LSD1^{A540E} and LSD1^{K662A} in the revised version.

- a) We conducted cell proliferation assays (revised **Figure S4A**).
- b) We subjected LSD1^{A540E} and LSD1^{K662A} mutant cell lines to EB differentiation and performed RT-qPCR of pluripotency factors and developmental genes (revised **Figure S4B-S4G**).
- c) We evaluated DNMT1 and UHRF1 protein levels (revised **Figure S7J**).
- d) We analyzed 5mC levels using Mass Spectrometry (revised **Figure S7K**).
- e) We assessed chromatin recruitment of RCOR1 and RCOR2 in LSD1^{A540E} and LSD1^{K662A} mutant cell lines (**Figure Rebuttal 5**).

Our overall data suggests that single mutants, akin to double mutant cell lines, do not exhibit proliferation defects but demonstrate differentiation deficiencies (see comment 4 from the same reviewer). Furthermore, these mutants exhibit the same phenotype of rescuing UHRF1/DNMT1 protein levels as well as 5mC levels. This inclusion further supports the novel mechanism we have identified through the non-catalytic function of LSD1 in controlling DNA methylation in ESCs. Finally, both LSD1^{A540E} and LSD1^{K662A} are able to recruit the CoREST complex to chromatin.

- 1) There is already a significant body of literature describing the role of Lsd1 in both mouse and human ESC biology, so to state that role of Lsd1 in ESC self-renewal and differentiation is "poorly understood" in the Introduction is inaccurate and demonstrates a lack of effort to articulate the current knowledge base for

Lsd1 in ESC biology, what the outstanding questions in the field are, and how this study will address them. Indeed, there are many reviews published on this topic due to the large body of work already performed. In general, the manuscript lacks clarity for understanding why experiments were performed and how the data rigorously support the conclusions.

Response: We thank the reviewer for his/ her valuable feedback and apologize for any oversight in characterizing the current understanding of LSD1's functions. While it is indeed acknowledged that several studies have investigated the involvement of LSD1 in ESC self-renewal and differentiation, our intent was to highlight that despite these efforts, certain aspects in the LSD1's precise mechanisms in ESC biology remain unclear. For instance, as discussed in the introduction, conflicting conclusions have arisen regarding whether LSD1 primarily maintains ESC self-renewal or facilitates ESC differentiation. Additionally, recent exploration has shed light on LSD1's roles beyond its demethylase activity. We recognize the presence of existing reviews on this topic, including one of the most recent ones from our laboratory [19]. In response to the reviewer's suggestions, we have revised the introduction and moderated the aforementioned statement.

New text: *Although LSD1 has been shown to be involved in early embryogenesis, our comprehension of its function in ESC self-renewal and differentiation is still evolving.*

2) DNMT1 and UHRF1 are already identified targets of Lsd1, with the target lysine residues defined (e.g., K1096 and K142 in DNMT1). The Lsd1MUT construct used in this study impairs histone demethylation, however it is not known if it can still demethylate DNMT1 or UHRF1 proteins, as this assay was not performed. Thus, the data shown cannot exclude the possibility that Lsd1 demethylase activity is important for DNMT1 stability, particular since other groups have already demonstrated this mechanism (e.g., Wang et al., Nat. Genetics 2008). If the authors propose to refute the previously published mechanism, they need to perform the same assays with the Lsd1MUT variant to show it no longer de-methylates DNMT1 and/or the K1096 and K142 lysine's are demethylated via an alternative mechanism (e.g., another demethylase that binds Lsd1MUT).

Response: In our investigation, we demonstrate that LSD1 plays a crucial role in maintaining the stability of DNMT1 and UHRF1, as evidenced by the decrease in protein levels and 5mC levels observed in *Lsd1* knockout ESCs (**Figures 4A-4D** and **5A-5B**). Moreover, our findings indicate that both LSD1^{MUT} (A540E/K662A) and single mutants (LSD1^{A540E} and LSD1^{K662A}, now included in the revised manuscript) effectively restore the observed phenotype, suggesting that LSD1 demethylase activity is dispensable in this context (**Figures 4E, 4O, 5C, 5D, S7J** and **S7K**). In response to the reviewer's suggestion, we conducted in vitro assays utilizing a UHRF1 peptide containing the pertinent demethylated sites (ESKKKA{Lys(Me1)}MASATSS) [26]. Revised **Figure 5G** demonstrates LSD1's incapacity to directly demethylate UHRF1, while retaining its ability to demethylate the H3K4me2 peptide. Critically, cryoEM

studies validate these findings, revealing that LSD1 is specific for H3K4 demethylation because there is no space in the LSD1 catalytic site to fit more than 3 residues N-terminal to the target lysine for H3K4 [24].

New text: *To validate this observation, we performed in vitro assays using a UHRF1 peptide containing the specified demethylated site K385 (ESKKKA{Lys(Me1)}MASATSS) and purified WT LSD1 protein. Our findings indicate that LSD1 is unable to directly demethylate UHRF1 while maintaining its capacity to demethylate the H3K4me2 peptide (Figure 5G).*

3) Using the phrase “catalytically-dead” to describe the Lsd1MUT construct is imprecise given the history of describing similar Lsd1 alleles in past studies that were later shown to retain activity. The original manuscript describing the Lsd1 A539E/K661A allele (Kim et al., Mol. Cell 2020) used the term “catalytically-impaired”, which is more appropriate and should be used instead.

Response: We value the reviewer's feedback. We want to emphasize once more that the LSD1^{MUT} construct mentioned in our study is the double mutant A540/K662A, which has been demonstrated to lack enzymatic activity [24]. However, in line with the reviewer's suggestion, we have now revised the description of LSD1^{MUT} as catalytically-impaired.

4) The reliance on just one construct to infer Lsd1 non-catalytic roles in ESC biology is not rigorous. The K662A allele preferentially impacts FAD-dependent oxidase function, whereas the A540E allele introduces a bulky positive charge into Lsd1's hydrophobic pocket that interferes with substrate binding, at least for the Histone H3 tail (Kim et al., Mol. Cell 2020). Thus, the double mutant construct used in this study impacts two different functions of Lsd1. Therefore, it is not possible to discern if Lsd1 substrate binding or catalysis is important for its role in ESC growth and methylation. The study should include the analysis of each single mutant to allow comparison with the double mutant in order to determine the extent to which catalytic activity or substrate binding is important for LSD1's role in ESC methylation and growth. Such an analysis was recently performed in zebrafish hematopoietic stem cells to differentiate between these two functions of Lsd1 in controlling transcriptional programs during differentiation in vivo (Casey et al., iScience 2023).

Response: We appreciate the insightful comment provided by the reviewer. As previously stated, we conducted a series of experiments with each of the single mutants. As depicted in the new **Figure S4A**, LSD1^{A540E} and LSD1^{K662A} ESC lines did not exhibit any proliferation defects, mirroring the behavior of the double mutant (A540E/K662A) and the wild-type counterpart (**Figure S3H**).

New text: *Studies have highlighted that the A540E mutation disrupts substrate binding in LSD1 by introducing a positive charge, while the K662A allele primarily impacts the FAD-dependent oxidase function of LSD1. Therefore, our aim was to investigate the impact of substrate binding or catalytic function on both mouse ESC growth and EB formation by using cell lines individually carrying each of these single mutants (LSD1^{A540E} and LSD1^{K662A}). In line with our findings in the double mutant cell line (LSD1^{MUT}), both LSD1^{A540E} and LSD1^{K662A} mouse ESCs exhibited a proliferation rate similar to that of WT mouse ESCs (Figure S4A).*

Moreover, both LSD1^{A540E} and LSD1^{K662A} ESCs restored the levels of DNMT1 and UHRF1 proteins, as well as 5mC levels which were absent in the *Lsd1* KO ESCs (revised **Figures S7J** and **S7K**), underscoring once more the scaffolding role of LSD1 in maintaining the stability of these proteins.

New text: *Additionally, single mutants LSD1^{A540E} and LSD1^{K662A} demonstrated the capability to restore the expression of DNMT1 and UHRF1 proteins, along with 5mC levels (Figures S7J and S7K).*

5) An important non-catalytic function of *Lsd1* is binding to SNAG-domain transcription factors (e.g., Gfi1 & Snail proteins) via its substrate binding domain, which in turns controls *Lsd1* recruitment to specific genomic loci and transcription during differentiation (e.g., Gfi1/1b binding to *Lsd1* in AML cell lines). This alternative mechanism was not investigated in this study and may explain the inability of the *Lsd1*MUT variant to rescue the differentiation program in ESC.

Response: Thank you for your insightful comment. We have observed several factors that suggest the implausibility of the involvement of the SNAG-domain transcription factors in explaining the inability of LSD1^{MUT} to rescue the differentiation program in ESCs.

- a) To address the reviewer's comment, we conducted EB differentiation on LSD1^{A540E} and LSD1^{K662A} single mutant cell lines (new **Figures S4B-S4G**). During this process, LSD1^{K662A} exhibited a more prominent defect in silencing the expression of the pluripotency factor *Oct4* compared to LSD1^{A540E}, while no significant differences were observed between LSD1^{A540E} and wild-type EBs for *Nanog* and *Sox2* (**Figure S4D**). This suggests that the substrate binding activity of LSD1 may play a less crucial role in repressing the expression of pluripotency factors. However, both LSD1^{A540E} and LSD1^{K662A} showed similar outcomes regarding the incapability of activating the expression of developmental genes, with the effect being more pronounced for the K662A mutant (**Figures S4E-S4G**).

New text: *Our analysis further revealed that the EBs derived from LSD1^{A540E} and LSD1^{K662A} were significantly smaller than WT EBs, albeit partially rescuing the size of EBs derived from *Lsd1* KO ESCs, which was only 50% of their wild-type counterparts (Figures S4B, S4C, S3A, and S3B). LSD1^{K662A} EBs demonstrated a more prominent defect in suppressing the expression of *Oct4* compared to LSD1^{A540E} EBs, while no significant differences were observed between LSD1^{A540E} and WT-derived EBs for *Nanog* and *Sox2* on day 8 of differentiation (Figure S4D). Additionally, although both mutants failed to promote developmental gene expression at WT levels, the impact was more pronounced in LSD1^{K662A}-derived EBs (Figures S4E-G). This data suggests that the catalytic function of LSD1 holds more importance in EB differentiation than its substrate binding function.*

- b) Gfi1 and Gfi1b are recognized as major regulators of both early hematopoiesis and hematopoietic stem cells, whereas SNAIL1 has been shown to be important for ESC differentiation. Hence, we focused on SNAIL1. We attempted to co-immunoprecipitate LSD1 in LSD1^{MUT}, LSD1^{A540E} and

LSD1^{K662A} cell lines. However, despite our efforts, three different antibodies failed to work in our mouse protein extracts, despite being indicated as suitable for both mice and humans (**Figure Rebuttal 2**).

Figure R2: Western blot analysis was performed using whole protein extracts from mouse ESCs, MCF7, MDA-MB-231, and HEK293T cells. Fifty micrograms of protein were loaded onto the gel. ACTIN served as the loading control.

To resolve the concern about antibody specificity, we transfected HEK293T cells with MYC-LSD1^{MUT}, MYC-LSD1^{K662A}, and MYC-LSD1^{A540E} constructs. Subsequent immunoprecipitation using MYC antibodies revealed interactions with SNAIL1 for all three constructs. This finding suggests that SNAIL1 may not be implicated in the observed differentiation defects observed in the mutant cell lines.

Figure R3: HEK293T cells were transfected with the indicated plasmids. MYC immunoprecipitation followed by immunoblotting of MYC, SNAIL1, and ACTIN (only in input). The percentage of input used is 10%. Two independent experiments are shown.

In agreement with this observation, it is worth noticing that as per the literature [27], *Snail1* KO EBs exhibit an increase in neuroectodermal commitment combined with marked defects in mesoderm- and endoderm-associated gene expression (**Figure Rebuttal 4**), a pattern not observed in *Lsd1* KO or mutated cell lines, as we observed defects across all three germ layers (**Figures 2M-O** and **S4E-G**).

Figure R4: Loss of *Snail1* inhibits mesodermal/endodermal while promoting neuroectodermal gene expression (mean \pm 1 s.d.; n=3); from [27].

c) As per the manuscript referenced by the reviewer in zebrafish hematopoietic stem cells (comment 4) [28], SNAG domains of SNAG-domain-containing proteins recruit LSD1, which acts as a scaffold to assemble chromatin-modifying complexes such as CoREST. As depicted in Figure **Rebuttal 5**, both A540E and K662A are capable of recruiting CoREST (RCOR1 and RCOR2) to the chromatin, indicating that a different mechanism is at play in our cellular model.

Figure R5: Western blots of LSD1, C-MYC, RCOR1, RCOR2, and HDAC1 on the chromatin fractions of (CH) WT, Lsd1 KO, LSD1^{A540E}, and LSD1^{K662A} ESCs. H3 is used as the loading control.

Collectively, this data suggests that SNAIL1 likely does not play a role in mediating the differentiation defects observed in LSD1^{MUT} EBs.

6) Figure 1I and S1E. The Western blot for Lsd1 in KO cell lines should show the complete blot/film (at least as a supplemental figure), as Lsd1 has multiple splice isoforms, and it is possible the CrispR-generated mutations produce an Lsd1 protein via alternative splicing that runs at a lower molecular weight.

Response: We appreciate the reviewer's feedback, and in response, we've replaced the previous western blot with a complete blot displaying the absence of smaller LSD1 isoforms in our KO cell lines (revised **Figure S1E**).

7) Figure 1M. There is significant transcriptional heterogeneity between the two Lsd1 KO cell lines. As the authors point out, this may be due to genetic compensation. However, this was not examined further, such as analysis of Lsd2 levels (a possible mechanism of compensation), which limits the conclusion in this section.

Response: We would like to apologize for the confusion. We failed to clarify that the variation in *Lsd1* KO cell lines relates not to genetic compensation but to stochastic heterogeneity. The challenge of establishing reliable genotype–phenotype connections arises from the inherent variability among different knockout clones of the same gene. Despite our efforts to reduce off-target effects by generating two independent *Lsd1* CRISPR KO cell lines using two distinct approaches, encountering phenotypic variability among these knockout clones is a common occurrence. This phenomenon has been noted in studies where even apparently homogeneous stable cell lines show significant differences in phenotypes when individual wild-type clones are isolated [29].

However, to bolster the conclusion of this section, specifically the findings from the transcriptomic analysis, we conducted RNA-seq analyses upon reintroduction of wild-type LSD1 and double-mutated LSD1 (LSD1^{WT} and LSD1^{MUT} cell lines, respectively). This was done to obtain a more comprehensive understanding of the impact of LSD1 on the mouse ESC transcriptome (refer to new **Figures S3K–S3P**). As illustrated in the updated **Figure S3L**, LSD1^{MUT} displays a 60% overlap in upregulated genes with *Lsd1* KO ESCs. This suggests that while loss of *Lsd1* results in more pronounced dysregulation compared to the catalytically inactive form, both cell lines share a similar regulatory program. Furthermore, WT and LSD1^{WT} clustered closely together, affirming the consistency of the transcriptomics analysis.

New text: *In LSD1^{MUT}, RNA-seq analysis unveiled 685 downregulated genes and 1115 upregulated genes compared to their wild-type counterparts (Figure S3K; Table S1). Among the 1115 upregulated genes in LSD1^{MUT} ESCs, 668 genes were similarly upregulated due to LSD1 deletion (Figure S3L). Further insight into the distinction between the impacts of LSD1 deletion and catalytic inactivation on the ESC transcriptome was underscored through visualization of RNA-seq data (Figure S3M). Notably, LSD1^{WT} cells were able to rescue the transcriptomic defects as only 390 genes were dysregulated, potentially reflecting clonal variability (Figures S3N). GO analysis of the biological processes associated with upregulated genes in LSD1^{MUT} ESCs identified categories related to regulation of transcription, chromatin organization, DNA repair, and neuron projection development, amongst others (Figures S3O). Conversely, downregulated genes in LSD1^{MUT} exhibited enrichment in generic categories such as cell differentiation, cell cycle, phosphorylation, and chromatin organization, among others (Figures S3P). Once again, despite these alterations in gene expression, there were no observable differences in the expression of pluripotency-related markers between the wild-type and LSD1^{MUT} ESCs.*

Furthermore, in response to the reviewer's suggestion, we evaluated *Lsd2* mRNA levels in the *Lsd1* KO ESC lines. As depicted in the **Figure Rebuttal 6**, we did not detect an elevation in the expression of *Lsd2* that could compensate for the loss of LSD1. Conversely, *Lsd2* levels were diminished following *Lsd1* knockout.

Figure R6: RT-qPCR of *Lsd2* in WT and *Lsd1* KO1 and KO2 ESCs. mRNA levels are relative to the expression of WT at day 0. β -actin is used as an internal control.

8) “There were no differences in the expression of multiple pluripotency and germ layer-specific markers between WT and *Lsd1* KO ESCs”. RT-PCR analysis shows an increase in endodermal markers (Fig S1J), which is inconsistent with the statement.

Response: We apologize for the confusion caused by our statement. We intended to convey that GO analysis did not reveal any enrichment of germ-layer specific categories affected by the loss of LSD1, which does not exclude the possibility that specific markers, such as *Sox17* and *Gata6*, may be dysregulated. To clarify, the text has been amended.

New text: GO analysis did not reveal global alterations in categories associated with pluripotency and germ layer-specific markers between WT and *Lsd1* KO mouse ESCs. Nevertheless, we did detect a specific upregulation of the endoderm genes *Sox17* and *Gata6*, which were used to validate the RNA-seq data by RT-qPCR (**Figure S2H**)”.

9) Figure 2B. The demethylase assay is specific for H3K4me2, so no inferences can be made regarding *Lsd1* demethylase activity on other substrates used in this study, such as DNMT1.

Response: Enzymatic assays have been conducted using a UHRF1 peptide containing the specified demethylated sites. Please refer to comment 2 from the same reviewer for further details.

10) Figure 2I. The *Lsd1*MUT protein levels should be included in this analysis.

Response: As suggested by the reviewer, we have assessed the *Lsd1*^{MUT} protein levels at day 0 and at day 8 of the differentiation (revised **Figure 2I**).

11) Figure 2. Overall, the data does not rule out the possibility that *Lsd1*’s catalytic and non-catalytic function plays a role in ESC differentiation, as the *Lsd1*MUT partially rescues the expression of many differentiation genes, just not as robustly as the *Lsd1*WT version.

Response: We respectfully disagree with the reviewer's comment. Our data clearly demonstrate that during EB differentiation, *LSD1*^{MUT}-derived EBs fail to suppress the pluripotency markers *Oct4*, *Nanog*, and *Sox2*, mirroring the observations made in *Lsd1* KO ESCs. Additionally, the mutant cell line *LSD1*^{MUT} exhibits an inability to induce the expression of differentiation markers, except for *Sox11*, which did not show a significant effect. Furthermore, we observed that the size of the EBs derived from the *LSD1*^{MUT} cell line is markedly smaller compared to the wild-type (WT) counterparts. These findings strongly support the critical

role of LSD1's catalytic activity in governing gene expression during differentiation. Furthermore, similar results were obtained during gastrulation, reinforcing the significance of LSD1 in regulating developmental processes.

In order to clarify our observations, we provide the Tables R1 and R2 with the raw values below:

Fold Induction EBs									
Targets	Day	WT		KO2		LSD1 ^{WT}		LSD1 ^{MUT}	
		mean	stdev	mean	stdev	mean	stdev	mean	stdev
Oct4	D0	1.028	0.055	0.749	0.116	1.542	0.299	1.523	0.153
	D8	0.212	0.118	1.081	0.223	0.473	0.125	1.223	0.067
Nanog	D0	1.003	0.005	1.132	0.228	1.433	0.250	2.063	0.720
	D8	0.176	0.098	0.822	0.124	0.333	0.103	0.77	0.328
Sox2	D0	1.003	0.005	1.115	0.061	1.687	0.423	1.783	0.240
	D8	0.252	0.171	1.063	0.082	0.407	0.110	1.02	0.327
Sox17	D0	1.03	0.005	3.156	1.804	2.72	0.517	3.477	1.470
	D8	234.517	106.62	1.962	0.571	374.47	64.386	3.347	1.263
Foxa2	D0	1.005	0.005	1.456	0.552	0.397	0.223	0.943	0.741
	D8	118.139	29.56	4.086	1.190	148.583	44.60	13.68	7.71
T	D0	1.026	0.038	0.435	0.226	0.908	0.600	1.233	0.687
	D8	260.418	80.484	1.073	0.264	563.1	128.205	11.637	6.730
Msx1	D0	0.991	0.021	0.723	0.297	1.367	0.472	1.383	0.311
	D8	182.183	23.57	4.176	0.465	44.61	11.362	11.707	3.643
Sox11	D0	1.003	0.005	0.701	0.138	1.503	0.399	2.823	2.655
	D8	46.730	6.916	8.859	0.925	23.52	4.815	19.68	6.249
Nestin	D0	1.005	0.005	0.18	0.051	0.703	0.172	0.403	0.040
	D8	3.167	0.831	0.095	0.062	2.746	1.165	0.316	0.204

Table R1: Raw data displaying the mean and standard deviation of the fold induction of pluripotency and differentiation markers at specified time points during EB differentiation. The values represent the mean of three independent experiments.

Fold induction Gastruloids											
Targets	h	WT		KO1		KO2		LSD1WT		LSD1MUT	
		mean	stdev	mean	stdev	mean	stdev	mean	stdev	mean	stdev
Oct4	0	1.037	0.064	0.647	0.101	0.77	0.079	1.543	0.299	1.493	0.195
	120	0.227	0.142	1.133	0.281	1.277	0.297	0.283	0.051	0.81	0.142

Sox17	0	1.013	0.006	4.013	1.083	3.467	0.696	2.72	0.517	4.227	1.051
	120	118.08	16.531	4.00	1.045	3.44	1.442	41.133	20.789	2.776	1.191
T	0	1.033	0.040	0.323	0.119	0.447	0.240	1.397	0.715	1.087	0.471
	120	1512.2	755.18	102.81	82.545	0.943	0.719	1081.1	182.47	410.77	192.23
Sox11	0	1.003	0.006	0.82	0.4551	0.843	0.2055	1.267	0.186	1.12	0.386
	120	49.368	7.408	12.941	2.678	7.290	3.2455	23.253	4.427	12.567	3.831
Nestin	0	1.007	0.006	0.187	0.031	0.130	0.104	0.703	0.172	0.39	0.056
	120	3.55	0.979	0.272	0.167	0.02	0	2.413	1.000	0.991	0.13
Hoxd3	0	1	0	1.6	0.291	26.963	14.120	2.447	1.481	2.5867	0.1620
	120	3973.5	1681.2	16.342	2.278	196.39	77.090	4998.3	861.81	212.57	58.708

Table R2: Raw data displaying the mean and standard deviation of the fold induction of pluripotency and differentiation markers at specified time points during gastrulation. The values represent the mean of three independent experiments.

Moreover, we conducted EB differentiation concurrently with LSD1 inhibitor treatment. As depicted in **Figure Rebuttal 7**, treatment of EBs with the GSK_LSD1 inhibitor led to the upregulation of pluripotency factors *Oct4* and *Sox2* by day 8 of differentiation compared to DMSO, while no notable differences were observed in *Nanog* expression compared to the control. Additionally, we observed a decrease in the expression of endodermal, mesodermal, and ectodermal factors by day 8 of differentiation upon inhibitor treatment. These findings suggest that the demethylase activity of LSD1 is vital for establishing a proper differentiation program, as suggested previously [3] [21].

Figure legends on the next page.

Figure R7: EB differentiation of WT mouse ESCs treated with the LSD1 inhibitor GSK_LSD1. (A and B) (A) Representative bright field images at (4x (left) and 10x (right)) magnification and (B) quantification of the size of EBs derived from DMSO and GSK_LSD1 treated WT mouse ESCs on the day 8 of differentiation. Scale bars, 200 μ m. (C-F) RT-qPCR analysis of (C) the pluripotency (Oct4, Nanog and Sox2), (D) the endodermal (Sox17 and Foxa2), (E) the mesodermal (T and Msx1), and (F) the ectodermal (Sox11 and Nestin) markers in DMSO and GSK_LSD1 treated WT mouse ESCs on the indicated time points. mRNA levels are relative to the expression of WT at day 0. Statistical analysis: unpaired t-test (B and C-F). * $p < 0.05$, ** $p < 0.01$ and *** $p < 0.001$. Error bars denote mean \pm SD; $n \geq 3$ (B and C-F). Results are one representative of $n = 3$ independent experiments (A).

12) Text: "Such an increase in the percentage of partially differentiated colonies is a consequence of the generic proliferative defects of loss of LSD1 as ablation of Lsd1 did not affect the expression of the pluripotency factors OCT4 and NANOG"- why would a loss of proliferative capacity impact differentiation? This should be more fully explained.

Response: The decline in proliferative capacity in mouse ESCs can disrupt the balance between self-renewal and differentiation, leading to a diminished pool of stem cells available for differentiation and alterations in signalling pathways, potentially inducing premature differentiation. Nonetheless, we have removed the aforementioned sentence to enhance the comprehensiveness of the text.

13) The findings and potential conclusions of this study are limited to mouse ESC, as previous studies have shown that the function of Lsd1 in mouse versus human ESC biology differs significantly. This should be stated in the limitations of the study.

Response: We have included the aforementioned statement in the limitations of the study.

New text: *It is important to note that the findings and potential conclusions of this study are confined to mouse ESCs, as previous research has demonstrated significant differences in the function of LSD1 between mouse and human ESCs.*

References:

1. Martello, G. and A. Smith, *The nature of embryonic stem cells*. Annu Rev Cell Dev Biol, 2014. **30**: p. 647-75.
2. Ghimire, S., et al., *Comparative analysis of naive, primed and ground state pluripotency in mouse embryonic stem cells originating from the same genetic background*. Sci Rep, 2018. **8**(1): p. 5884.
3. Whyte, W.A., et al., *Enhancer decommissioning by LSD1 during embryonic stem cell differentiation*. Nature, 2012. **482**(7384): p. 221-5.
4. Wang, J., et al., *The lysine demethylase LSD1 (KDM1) is required for maintenance of global DNA methylation*. Nat Genet, 2009. **41**(1): p. 125-9.
5. Zeng, C., et al., *Demethylase-independent roles of LSD1 in regulating enhancers and cell fate transition*. Nat Commun, 2023. **14**(1): p. 4944.
6. Zambanini, G., et al., *A new CUT&RUN low volume-urea (LoV-U) protocol optimized for transcriptional co-factors uncovers Wnt/beta-catenin tissue-specific genomic targets*. Development, 2022. **149**(23).

7. Egolf, S., et al., *LSD1 Inhibition Promotes Epithelial Differentiation through Derepression of Fate-Determining Transcription Factors*. Cell Rep, 2019. **28**(8): p. 1981-1992 e7.
8. Vinckier, N.K., et al., *LSD1-mediated enhancer silencing attenuates retinoic acid signalling during pancreatic endocrine cell development*. Nat Commun, 2020. **11**(1): p. 2082.
9. Alajem, A., et al., *DNA methylation patterns expose variations in enhancer-chromatin modifications during embryonic stem cell differentiation*. PLoS Genet, 2021. **17**(4): p. e1009498.
10. Teng, L. and K. Tan, *Finding combinatorial histone code by semi-supervised biclustering*. BMC Genomics, 2012. **13**: p. 301.
11. Baubec, T., et al., *Methylation-dependent and -independent genomic targeting principles of the MBD protein family*. Cell, 2013. **153**(2): p. 480-92.
12. Gifford, C.A., et al., *Transcriptional and epigenetic dynamics during specification of human embryonic stem cells*. Cell, 2013. **153**(5): p. 1149-63.
13. Xie, R., et al., *Dynamic chromatin remodeling mediated by polycomb proteins orchestrates pancreatic differentiation of human embryonic stem cells*. Cell Stem Cell, 2013. **12**(2): p. 224-37.
14. Sharifi-Zarchi, A., et al., *DNA methylation regulates discrimination of enhancers from promoters through a H3K4me1-H3K4me3 seesaw mechanism*. BMC Genomics, 2017. **18**(1): p. 964.
15. Li, R., S.A. Grimm, and P.A. Wade, *CUT&Tag-BS for simultaneous profiling of histone modification and DNA methylation with high efficiency and low cost*. Cell Rep Methods, 2021. **1**(8).
16. Song, Y., et al., *Dynamic Enhancer DNA Methylation as Basis for Transcriptional and Cellular Heterogeneity of ESCs*. Mol Cell, 2019. **75**(5): p. 905-920 e6.
17. Du, Z., et al., *DNMT1 stability is regulated by proteins coordinating deubiquitination and acetylation-driven ubiquitination*. Sci Signal, 2010. **3**(146): p. ra80.
18. Ahmad, T., et al., *TIP60 governs the auto-ubiquitination of UHRF1 through USP7 dissociation from the UHRF1/USP7 complex*. Int J Oncol, 2021. **59**(5).
19. Martinez-Gamero, C., S. Malla, and F. Aguilo, *LSD1: Expanding Functions in Stem Cells and Differentiation*. Cells, 2021. **10**(11).
20. Foster, C.T., et al., *Lysine-specific demethylase 1 regulates the embryonic transcriptome and CoREST stability*. Mol Cell Biol, 2010. **30**(20): p. 4851-63.
21. Dan, S., et al., *LSD1-mediated demethylation of OCT4 safeguards pluripotent stem cells by maintaining the transcription of PORE-motif-containing genes*. Sci Rep, 2021. **11**(1): p. 10285.
22. Adamo, A., et al., *LSD1 regulates the balance between self-renewal and differentiation in human embryonic stem cells*. Nat Cell Biol, 2011. **13**(6): p. 652-9.
23. Garcia-Martinez, L., et al., *Endocrine resistance and breast cancer plasticity are controlled by CoREST*. Nat Struct Mol Biol, 2022. **29**(11): p. 1122-1135.
24. Kim, S.A., et al., *Crystal Structure of the LSD1/CoREST Histone Demethylase Bound to Its Nucleosome Substrate*. Mol Cell, 2020. **78**(5): p. 903-914 e4.
25. Cheng, J., et al., *Molecular mechanism for USP7-mediated DNMT1 stabilization by acetylation*. Nat Commun, 2015. **6**: p. 7023.
26. Hahm, J.Y., et al., *Methylation of UHRF1 by SET7 is essential for DNA double-strand break repair*. Nucleic Acids Res, 2019. **47**(1): p. 184-196.
27. Lin, Y., et al., *Snail1-dependent control of embryonic stem cell pluripotency and lineage commitment*. Nat Commun, 2014. **5**: p. 3070.
28. Casey, M.J., et al., *The scaffolding function of LSD1/KDM1A reinforces a negative feedback loop to repress stem cell gene expression during primitive hematopoiesis*. iScience, 2023. **26**(1): p. 105737.

29. Westermann, L., et al., *Wildtype heterogeneity contributes to clonal variability in genome edited cells*. Sci Rep, 2022. **12**(1): p. 18211.

Reviewers' Comments:

Reviewer #1:

Remarks to the Author:

The authors successfully addressed very thoughtfully all of my comments. I congratulate all the authors for this very nice work. I now full support publication of the paper in Nature Communications.

Reviewer #2:

Remarks to the Author:

The authors have to a large extent addressed my concerns. Giving the extensive effort they have spent and comprehensiveness of the study, I will now recommend its acceptance for publication.

Reviewer #3:

Remarks to the Author:

The authors have added substantial new data to the revised manuscript to address concerns regarding rigor and alternative mechanisms of LSD1 function in mouse ES cell self-renewal, proliferation, and differentiation. The inclusion of LSD1 single mutant data adds substantial rigor to the study and strengthens many of the catalytic-dependent/independent conclusions, particularly the importance of LSD1's catalytic function in differentiation and non-catalytic role in DNA methylation. In addition, while not included in the manuscript, the additional data showing interactions between SNAG-domain containing proteins (SNAIL1) is much appreciated and shows for the first time that SNAIL1 still binds to a catalytically impaired LSD1, which may play roles in later developmental processes (beyond the scope of this study). Overall, the authors have significantly strengthened their conclusions with new data that is appropriate for publication.

A couple of minor points regarding the text (these are optional and only provided to help increase the impact of the study):

1) In my opinion, the authors have shown the non-catalytic role of LSD1 controls DNA methylation in mouse ES cells, although the exact mechanism(s) remain unknown, but likely includes regulation HDAC1 activity toward the DNMT1/UHRF1/USP7 complex. Nonetheless, a discussion of their findings in comparison to the Wang et al., Nature Genetics 2009 study may be warranted in the Discussion, as this prior study concluded that LSD1's catalytic activity controls DNMT1-dependent DNA methylation via regulating DNMT1 stability- the opposite conclusion from this study. Thus, the authors have an opportunity (but not obligated) to help the field by clarifying why such different conclusions are possible. In my opinion, the prior study conclusions (Wang et al) relied largely on one assay that used an irreversible LSD1 inhibitor (2-PCPA) that likely impacts both catalytic and non-catalytic LSD1 functions. Thus, the 2009 study was not as rigorous as this study, which I believe led to different conclusions. So, it might be helpful to point out how these technical differences led to different conclusions, while at the same time highlighting the rigor of the current study that would shift the field to accept the non-catalytic/scaffold role of LSD1 in controlling DNA methylation more decisively.

2) The new title is not very specific- "both catalytic and non-catalytic functions of LSD1..." it is quite nebulous and doesn't do justice to the great work described in the study. The authors previously stated in the original version (in the abstract) the following statement "The scaffolding function of LSD1 controls DNA methylation in mouse ESCs". The new data added to the revised manuscript now further strengthens this statement and I believe more succinctly describes the impact of the study.

3) In the limitations section, while I appreciate the authors stating the findings are confined to mouse ES cells, it appears to diminish the current study too much. I would recommend the authors instead state that LSD1 may have similar non-catalytic functions in human ESC methylation, but that needs to

be determined in future studies.

Rodney Stewart

Reviewers' comments:

Reviewer #1 (Remarks to the Author):

The authors successfully addressed very thoughtfully all of my comments. I congratulate all the authors for this very nice work. I now full support publication of the paper in Nature Communications.

We appreciate your kind words and thoughtful feedback.

Reviewer #2 (Remarks to the Author):

The authors have to a large extent addressed my concerns. Giving the extensive effort they have spent and comprehensiveness of the study, I will now recommend its acceptance for publication.

We thank the referee for the thorough review and constructive feedback.

Reviewer #3 (Remarks to the Author):

The authors have added substantial new data to the revised manuscript to address concerns regarding rigor and alternative mechanisms of LSD1 function in mouse ES cell self-renewal, proliferation, and differentiation. The inclusion of LSD1 single mutant data adds substantial rigor to the study and strengthens many to the catalytic-dependent/independent conclusions, particularly the importance of LSD1's catalytic function in differentiation and non-catalytic role in DNA methylation. In addition, while not included in the manuscript, the additional data showing interactions between SNAG-domain containing proteins (SNAIL1) is much appreciated and shows for the first time that SNAIL1 still binds to a catalytically impaired LSD1, which may play roles in later developmental processes (beyond the scope of this study). Overall, the authors have significantly strengthened their conclusions with new data that is appropriate for publication.

Thank you for your constructive feedback. We appreciate your recognition of the effort and comprehensiveness of our study. We are pleased that you now recommend our paper for publication.

A couple of minor points regarding the text (these are optional and only provided to help increase the impact of the study):

1) In my opinion, the authors have shown the non-catalytic role of LSD1 controls DNA methylation in mouse ES cells, although the exact mechanism(s) remain unknown, but likely includes regulation HDAC1 activity toward the DNMT1/UHRF1/USP7 complex. Nonetheless, a discussion of their findings in comparison to the Wang et al., Nature Genetics 2009 study may be warranted in the Discussion, as this prior study concluded that LSD1's catalytic activity controls DNMT1-dependent DNA methylation via regulating DNMT1 stability- the opposite conclusion from thus study. Thus, the authors have an opportunity (but not obligated) to help the field by clarifying why such different conclusions are possible. In my opinion, the prior study conclusions (Wang et al) relied largely on one assay that used an irreversible LSD1 inhibitor (2-PCPA) that likely impacts both catalytic and non-catalytic LSD1 functions. Thus, the 2009 study was not as rigorous as this study, which I believe led to different conclusions. So, it might be helpful to point out how these technical

differences led to different conclusions, while at the same time highlighting the rigor of the current study that would shift the field to accept the non-catalytic/scaffold role of LSD1 in controlling DNA methylation more decisively.

We appreciate the reviewer's acknowledgment of the rigor of our study. In response to the reviewer's suggestion, we have added the following paragraph to the discussion section. Furthermore, as indicated by the reviewer, we have included data demonstrating that pargyline degrades LSD1 protein. This data can be found in Supplementary Figure S9Z.

New text: *It is worth noting that the original study demonstrating that the catalytic activity of LSD1 controls DNA methylation by regulating DNMT1 protein stability relied largely on one assay using Pargyline (REF). Pargyline is an irreversible selective monoamine oxidase-B inhibitor that leads to LSD1 degradation, thereby affecting both catalytic and non-catalytic functions of LSD1 (Figure S9Z; left and right panel).*

2) The new title is not very specific- "both catalytic and non-catalytic functions of LSD1..." it is quite nebulous and doesn't do justice to the great work described in the study. The authors previously stated in the original version (in the abstract) the following statement "The scaffolding function of LSD1 controls DNA methylation in mouse ESCs". The new data added to the revised manuscript now further strengthens this statement and I believe more succinctly describes the impact of the study.

We have changed the title to '*The scaffolding function of LSD1 controls DNA methylation in mouse ESCs*'.

3) In the limitations section, while I appreciate the authors stating the findings are confined to mouse ES cells, it appears to diminish the current study too much. I would recommend the authors instead state that LSD1 may have similar non-catalytic functions in human ESC methylation, but that needs to be determined in future studies.

The limitations section has been edited according to the reviewer's suggestion.

New text: *LSD1 might exhibit similar non-catalytic functions in the methylation of human ESCs, but this requires validation in future studies.*